# Classical center-surround receptive fields facilitate novel object detection in retinal bipolar cells

John A. Gaynes[1], Samuel A. Budoff ®[1], Michael J. Grybko[1], Joshua B. Hunt ®[1] & Alon Poleg-Polsky ®[1] ✉

Antagonistic interactions between center and surround receptive field (RF) components lie at the heart of the computations performed in the visual system. Circularly symmetric center-surround RFs are thought to enhance responses to spatial contrasts (i.e., edges), but how visual edges affect motion processing is unclear. Here, we addressed this question in retinal bipolar cells, the first visual neuron with classic center-surround interactions. We found that bipolar glutamate release emphasizes objects that emerge in the RF; their responses to continuous motion are smaller, slower, and cannot be predicted by signals elicited by stationary stimuli. In our hands, the alteration in signal dynamics induced by novel objects was more pronounced than edge enhancement and could be explained by priming of RF surround during continuous motion. These findings echo the salience of human visual perception and demonstrate an unappreciated capacity of the center-surround architecture to facilitate novel object detection and dynamic signal representation.

The ability to detect motion begins in the retina, which contains retinal ganglion cells (RGCs) tuned to the presence of local motion[1–3], approaching objects[4,5], acceleration[6,7], and the direction of movement (for reviews, see[8,9]). In addition, some cells are capable of predictive encoding of moving object position[10]. The highly specialized computations in RGCs are driven and shaped by glutamate release from axonal terminals of bipolar cells (BCs), which in mice are divided into about 14–15 functional types that are tuned to different visual features[11–15]. The topographic stratification of BC axons in the inner plexiform layer (IPL) establishes some of the functional organization of visual processing in the retina: BCs that carry ON signals (depolarization to light) are found closer to the ganglion cell layer, and cells with sustained responses are segregated towards IPL borders[11,14–16]. The difference in visual processing between BCs reflects their circularly symmetric center-surround architecture, comprised of two separate concentric regions sampling the visual signal[12,17]. This RF structure is formed by direct innervation of BC dendrites by photoreceptors in their excitatory center and a combination of horizontal cell (HC) and amacrine cell (AC) inhibition in the antagonistic surround[12,15,18,19].

Historically, motion signals in BCs have been understood as a linear combination of static responses, much like how the perception of motion is produced in movies by a rapid presentation of discrete images[20–22]. However, computations in cells with center-surround RFs can be nonlinear[23–26] and critically depend on the spatiotemporal RF activation pattern[3,4,24,27], which differs between moving and static stimuli. Previous studies revealed that amacrine and ganglion cells with pronounced surrounds detect motion discrepancies between the center and surround RF components, which enables preferential tuning to localized motion[1,2,28–31]. However, despite the abundance of the classic center-surround RFs in the early visual system, it is presently unknown if this RF architecture supports other motion computations or how center-surround interactions affect BC activity.

To examine the properties of visual processing of moving objects in the bipolar population, we recorded the change in glutamate levels across different depths of the IPL and captured the release dynamics of

[1]Department of Physiology and Biophysics, University of Colorado School of Medicine, Aurora, CO, USA. ✉e-mail: alon.poleg-polsky@cuanschutz.edu

glutamate releasing cells to moving or stationary bars. We reveal significant alteration in the peak and the temporal characteristics of the glutamate responses following object motion. Additionally, our results indicate that cells in the retina can signal the appearance of novel objects that enter the visual scene. Flashed stationary objects or stimuli that emerge from static occluders provoke intense glutamate discharge, whereas continuous motion and disappearing stimuli suppress BC activation. These observations depended on HC feedback and were not affected by the pharmacological blockage of AC inhibition. Accordingly, a circuit-based model of visual signaling in the outer retina replicates the diversity of motion responses found experimentally and reveals how motion computations can be carried out at the first retinal synapse by a horizontal cell-derived inhibitory signal and influence the representation of a realistic visual input. Our results describe a fundamental property of signal integration in circularly symmetric center-surround RFs to identify newly appearing visual stimuli and diversify the representation of static and moving shapes.

## Results

### Glutamate responses in BCs to full-field motion are diverse and do not follow the response dynamics for stationary signals

To study the representation of moving stimuli in glutamate releasing cells in the retina, we used two-photon microscopy to collect light-driven glutamatergic signals in whole-mount mouse retinas expressing iGluSnFR, ether in all neurons or under the ChAT promoter[14,15,22,32–34]. We systematically surveyed all layers of the inner plexiform layer (IPL) with multiple scan fields; pixels with similar responses responding to static flashes and full-field moving bars were then grouped into regions of interest (ROIs, Figs. 1a, s1, s2). The spatial extent of most ROIs was smaller than 50 μm, indicating sampling from a single cell or at most two functionally similar BCs (Figs. 1a, s1)[15,22]. Responses to stationary flashes were used to combine ROIs from different experiments into functional clusters[11,15,22]. The optimal separation was obtained with 6 OFF and 8 ON clusters (Fig. 1b, d); comparable to previous classifications of glutamate signals in the IPL[11,15].

BCs are the main glutamate releasing cells in the IPL[12,14,15,35]. To shed light on the identities of the BCs contributing to the functional glutamate clusters, we decided to focus on the cluster distribution of the glutamatergic drive onto starburst amacrine cells (SACs) (Fig. 1c). The connectivity between BCs to SACs is known; OFF-SACs are innervated by most OFF-BCs (types 1, 2, 3a, 3b, and 4)[20,36], inputs from three subpopulations of the type 5 BCs and the type 7 BC dominate the drive to ON-SACs[36,37]. In our dataset, flex-iGluSnFR signals expressed under the control of the ChAT promoter were significantly enriched in OFF-polarity clusters C3, C5, and C6 and ON-polarity clusters C7, C8, and C10 (Fig. 1c). Based on the stratification profiles and response dynamics[15], C3 may represent a release from BC types 1 or 2 and C5 and C6 from BC types 3 or 4. Type 7 BCs were suggested to have slower dynamics[15] (but see ref. 22); thus a functional correspondence between C10 and type 7 BC is likely (Fig. 1c, d). Type 5 BCs have relatively similar release properties[15,22], they are probably represented by our C7 and C8 clusters (Fig. 1c, d).

Irrespective of the precise correlation between functional clusters and anatomical cell types, our dataset consists of multiple BC types, allowing us to examine responses in the BC population to moving visual stimuli. As expected from a slower RF engagement by moving bars, we observed prolonged response kinetics with this stimulus (Fig. 1d–f). Surprisingly, there was no correlation between the static flash- and motion-driven rise-time dynamics of the functional clusters (Fig. 1f, left, Pearson coefficient of determination, $r^2 = 0.09$). In some clusters, we noted a shift in the speed of the response. For example, C9 had faster mean(±SD) rise times to flashed stimuli than C10 ($53 \pm 21$ vs. $81 \pm 35$ ms; $p < 10^{-16}$, two-tailed $t$ test), but tended to respond slower to moving bars (rise time = $344 \pm 135$ vs. $282 \pm 141$ ms; $p < 10^{-7}$, two-tailed $t$ test, Fig. 1d–f). Similarly, several clusters with comparable rise/decay

response dynamics to static flashes had significantly different shape kinetics in response to moving bars (Fig. 1d–f).

It is possible that the difference in motion processing we describe reflects the topographic stratification of BC axons with sustained responses approximating the IPL borders[11–15]. To assess this, we analyzed signal parameters relative to recording depth (Fig. 1g) or signal transiency index (TI, calculated from stationary response kinetics, Fig. 1g, h). We identified a clear relationship between cluster transiency to the change in the amplitude ($r^2 = 0.85$) and the decay time ($r^2 = 0.53$) of motion responses relative to the stationary signals (Fig. 1h). Similarly, the effect of motion on the peak amplitude and decay time was greatest in the central regions of the IPL, reflecting the stratification level of the transient BCs (Fig. 1g). In contrast, the change in the rise-time did not follow the transient-sustained division (Fig. 1h). Instead, we observed a gradual decrease in the motion/stationary ratio for the rise-time kinetics with increasing depth in the retina (Fig. 1g). Overall, the observed low correlation in key aspects of response shape and the distinct pattern of signal dependency on IPL depth between static and moving objects indicate different temporal filters for the representation of motion and stationary information in the BC population.

Notably, these observations are not an artifact of our clustering approach, as our algorithm was agnostic to motion information. We conducted several tests to rule out the possibility that the results we describe here are due to the grouping of pixels with different recruitment times during motion responses. First, at odds with the predicted effects of such pixel averaging, the degree to which motion impacted signal dynamics varied systematically between clusters, and the inter-cluster variability of responses was higher during motion (Fig. 1d). Second, the mean responses recorded for each group closely mirrored the signals recorded in individual pixels (Fig. s2). Last, neighboring regions of the retina respond sequentially to motion, and for this reason, the influence of pixel averaging should be most evident in groups with wide spatial pixel distribution. In contrast to this prediction, however, we found that the spread of each group's pixels along the axis of motion was not correlated with the response dynamics (Fig. s3).

### Representation of moving stimuli by glutamate release in the IPL is highly sensitive to the presence of static occluders

Previous work demonstrated that neurons could employ a simple strategy of comparing the spatial extent of center-surround recruitment to detect local spatial contrasts[23,38,39] and diversify the representation of flashed objects[15,25]. According to the classic description of the center-surround interactions, occluders masking part of the surround enhance RF output (Fig. 2a "Edge"). We reasoned that responses to moving stimuli could also be sensitive to stationary edges in the RF. To explore this possibility, we presented horizontally moving bars and masked the stimulus on the left or the right halves of the display.

To quantify the edge effects, we analyzed responses from ROIs whose RF center was located near (less than 50 μm away from) the visual edge (Fig. s4). We first used the peak response to masked and full-field static flashes to compute the classical notion of local edge detection. In our experimental conditions, edge enhancement of static flashes was low, statistically insignificant for most functional release clusters but C10 and C13 (Figs. 2b, c, s4). By contrast, the kinetics and the amplitude of the glutamate release were significantly faster/higher for bars emerging from the mask than for motion in the opposite direction (Fig. 2a, b "Emergence" vs. "Exit", s4). The enhancement of emerging stimuli (analyzed from the comparison of the responses to bar motion from/to the mask) was highly prominent in most ON and OFF glutamate clusters (Fig. 2b, d). The degree of edge sensitivity varied significantly between ROIs in our dataset; the strongest novel object enhancement—with more than 30% difference between responses to motion from an edge than motion towards an edge—was seen in glutamate clusters near the middle of the IPL (and to some

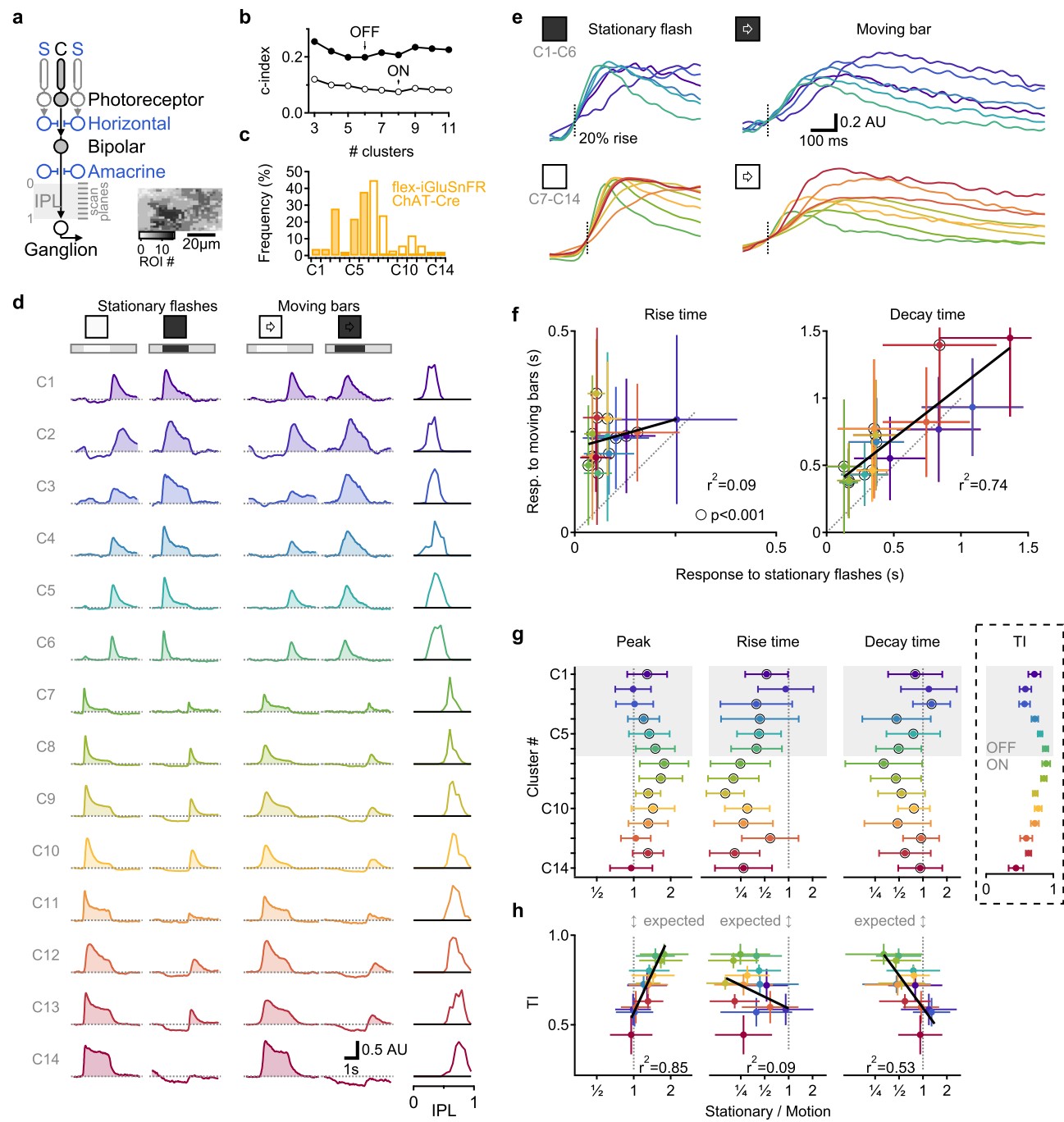

**Fig. 1 | Multiplexed representation of static and moving objects in BCs.**
**a** Schematic of the investigated retinal circuits. Glutamate release in the IPL is predominantly mediated by bipolar cells (BCs). BCs integrate photoreceptor drive in their RF center (C) with an antagonistic surround (S) formed by horizontal and amacrine cells. Inset, exemplar ROIs identified from iGluSnFR fluorescence in a single scan plane. **b** Diversity of responses to stationary flashes from 1828 ROIs (278 scan fields in 67 animals, 34 females; ages p43-359), arrows indicate the optimal number of functional clusters of glutamate release within the shown range. **c** Distribution of the clusters imaged from floxed-iGluSnFR expressed under the ChAT-Cre promotor. **d** Mean clusters' responses, sorted by pixel depth distribution in the IPL (right). **e** Focus on the rising phase of the signals. **f** Mean (±SD) clusters' kinetics, linear fits in black. Clusters with a significant ($p < 0.001$; paired $t$ test) difference between motion vs. stationary response are indicated by black circles. **g**, **h** IPL depth (**g**) or transiency index (TI; **h**, inset in **g**) vs. the mean (±SD) ratio between the stationary and motion responses for each of the glutamate clusters. Color coding in **e**–**h** by cluster identity. Black circles, clusters with stationary/motion response ratio with significant ($p < 0.001$, $t$ test) difference from unity.

degree in superficial ROIs; Fig. 2d, left). Correspondingly emerging motion enhancement was higher in transient glutamate clusters (Fig. 2b, d, right, s4). Therefore, although the visual stimuli we used were not optimized to produce pronounced edge effects[14,15], we found that BC responses to motion are nonetheless highly sensitive to the presence of occluders[40].

### Enhanced representation of novel stimuli

To explore the functional consequence of the stronger glutamatergic drive observed during emerging motion, we quantified the change in amplitude of the glutamate waveforms in these experiments. To compare different ROIs, we normalized the signals in each ROI by its peak response recorded during full-field motion (Fig. 2e). Since

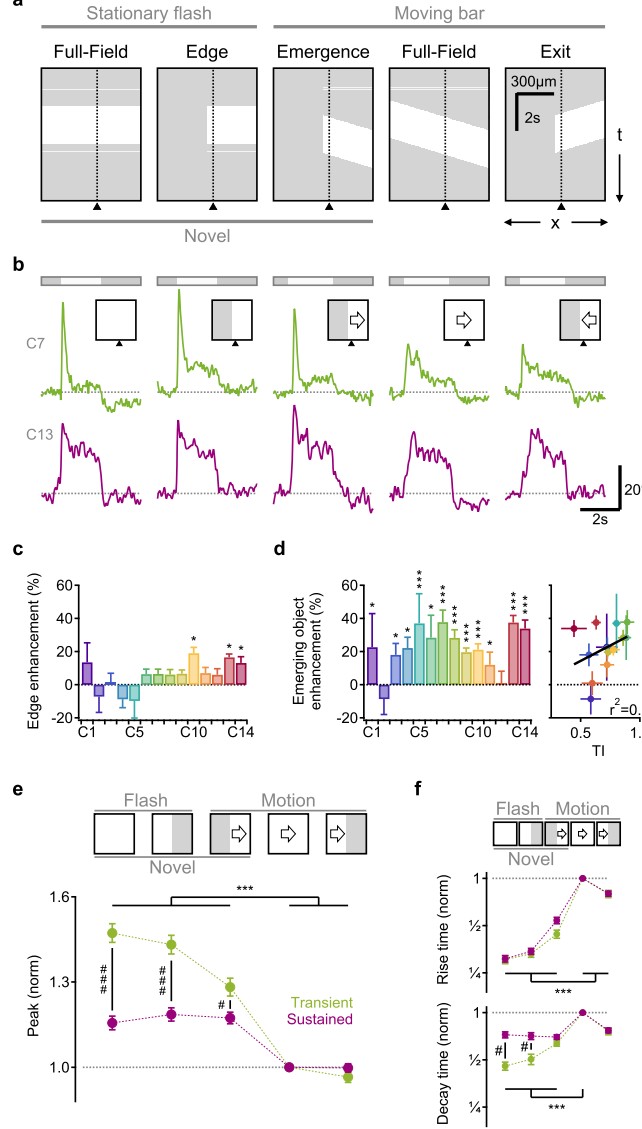

**Fig. 2 | Glutamate release in the IPL is sensitive to novel object appearance.** **a** Time-space plot indicating the position of the stimulus (white), presented either over the full extent of the display (Full-field) or masked by an occluder. Dotted line and triangle illustrate exemplar spatial receptive field (RF) center close to a mask-stimulus boundary. **b** Responses from two ROIs (green, transient; red, sustained) located near (<50 μm) a visual edge. Note the pronounced responses to emerging motion. **c** The mean (±SEM) static edge enhancement for the glutamate clusters, measured from the ratio of the peak amplitude of the responses to static edge and full-field stationary stimuli. **d** Left, the mean (±SEM) enhancement of emerging motion for the glutamate clusters, measured from the ratio between the peak amplitude of the responses to moving bar emergence and exit. Right, emerging object sensitivity vs. transiency index of the glutamate clusters, linear fit in black. *$p < 0.05$; ***$p < 0.001$ change from 0%; $n = 105$ scan fields in 55 animals, 30 females, ages p43-359. **e** The mean (±SEM) peak glutamate fluorescence for all ROIs with above (transient, green) and below (sustained, red) median transiency indexes, normalized by the peak dF/F recorded during full-field motion stimulation. ***$p = 0.001$ between different visual stimuli for transient/sustained BCs ($n = 307$ ROIs in each group). #$p < 0.001$; #$p < 0.001$; transient vs. sustained ROIs. **f** Similar to (**e**) for the rise and decay times. All statistical tests were one way ANOVA followed by Tukey test with Bonferroni's correction. See Supplementary Tables 1, 2 for detailed statistics.

sensitivity to emerging objects was higher in clusters with transient dynamics (Figs. 2d, s4), we divided our dataset into two halves based on the value of the TI. We assigned ROIs whose TI was above/below the median TI value to the transient/sustained groups respectively ($n = 307$ ROIs of both ON and OFF polarities in each) and analyzed the responses from the two populations separately (Fig. 2e, f). In both groups, the peak response amplitude during emerging object motion was significantly higher than the signal observed during continuous motion (128 ± 3% and 117 ± 2%, mean ± SEM for transient and sustained ROIs, $p < 10^{-5}$ for both, ANOVA followed by Tukey test with Bonferroni's correction for multiple comparisons (Fig. 2e)). In comparison, full-field static flashes were represented by waveforms that were 147 ± 4% and 115 ± 2% of the full-field motion responses in the two populations; this difference in the ratio between responses to stationary flashes and full-field motion was highly significant ($p < 10^{-16}$, Fig. 2e). Together with larger amplitudes, static flashes and emerging stimuli were encoded with significantly faster temporal kinetics (Figs. 2f, s4). In contrast, the dynamics of motion exit were indistinguishable from continuous motion (Figs. 2e, s4).

Based on these findings, we conclude that in terms of shape peak and temporal dynamics, the representation of emerging motion more closely resembles static flashes than continuous motion (Fig. 2). Because both object emergence behind a mask and flashed static stimuli represent novel visual items in the visual scene, whereas full-field motion and exiting objects correspond to known, preexisting stimuli by the time they reach the receptive field of the investigated cell, the difference in glutamate release we describe could correspond to the ability of BCs to identify the appearance of new visual objects[40].

## Horizontal cell, but not amacrine cell inhibition is required for novel object sensitivity

We hypothesized that the observed motion dynamics depend on interaction between the center and surround components of the receptive field. Previous work suggested that surround inhibition from ACs plays a major role in establishing BC responses to static stimuli[14,15,22]. To examine whether ACs affect motion processing in glutamate releasing cells in the IPL, we analyzed the change in the glutamate responses to the battery of visual stimuli following application of 50 μM SR95531, 100 μM TPMPA, and 1 μM Strychnine, used to block GABA_A, GABA_C and glycine receptors (Fig. 3a)[14,15,22]. We first confirmed the efficacy of the blockers on IPL synapses by noting the dramatic effect of the cocktail on calcium transients in RGCs, recorded in a separate experiment (Fig. s5). In agreement with previous findings, in most instances, the cocktail affected the shape of iGluSnFR waveforms at the stratification level of BC axon terminals in response to stationary flashes (Fig. 3b). Yet, the amplitude and the dynamics of motion signals in the IPL were largely unaffected by this perturbation (Fig. 3c-e, $n = 9$ animals, 83 ROIs)[41].

Because HCs can control photoreceptor output by mechanisms that do not require the release of neurotransmitters[19,42-44], we reasoned that our pharmacological manipulation did not fully disrupt the horizontal feedback on the photoreceptors. To explore the involvement of HCs in mediating motion computations, we again used a pharmacological approach. We perfused the tissue with CNQX (50 μM) to antagonize the photoreceptor signal to horizontal cells[45-47] and HEPES (10 mM) to reduce their feedback onto the photoreceptors[46,48-50]. We analyzed the effects of the cocktail on ON-BCs only because, as expected, this manipulation blocked the activation of OFF-BCs (data not shown). Unlike the mild effect we observed for the inhibitory blockers described above, disruption of the HC-mediated surround had a profound impact on glutamate signals (Fig. 3b, $n = 8$ animals, 105 ROIs). In the presence of CNQX and HEPES, the peaks of the responses to full-field stimulation increased relative to control conditions, both for static flashes and

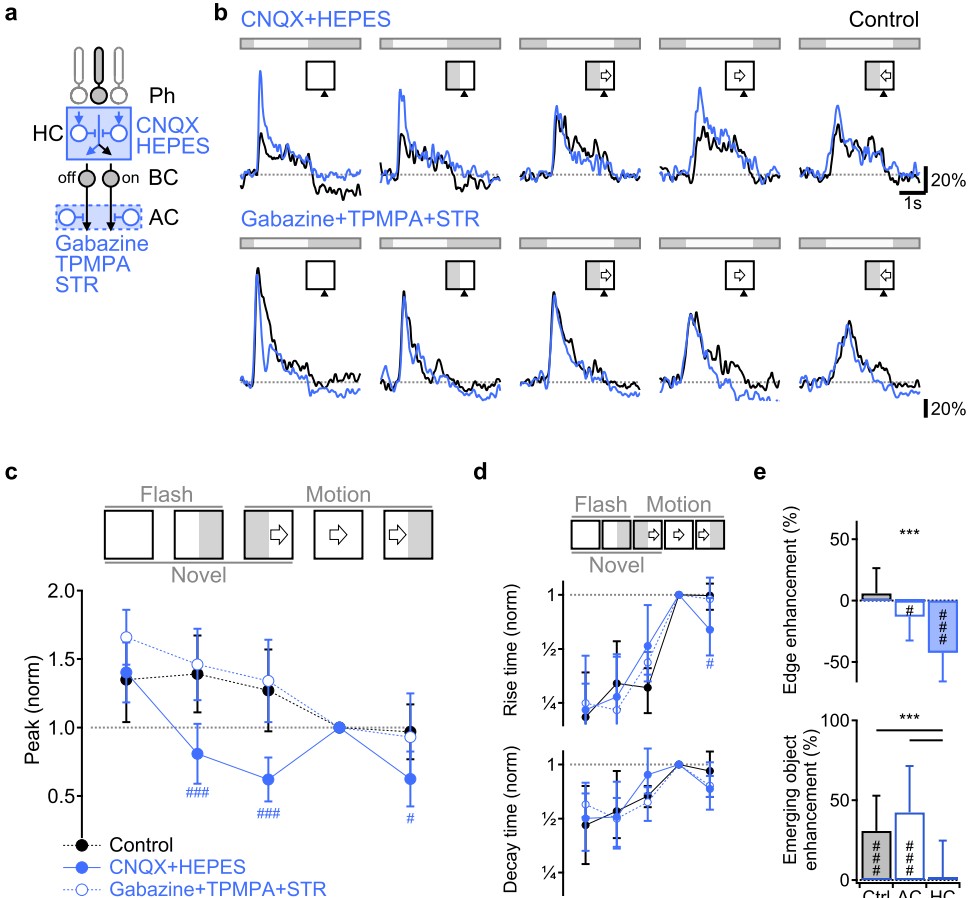

**Fig. 3 | Pharmacological blockage of inhibition from horizontal cells, but not amacrine cells, eliminates emerging object sensitivity and static edge enhancement. a** Illustration of the pharmacological manipulations designed to probe the contribution of the two circuit elements mediating the surround in BCs. CNQX (50 μM) and HEPES (10 mM) block photoreceptor input to Off-BCs and horizontal cells and part of the horizontal cells' feedback, required for the inhibitory surround. SR95531 (Gabazine, 50 μM), TPMPA (100 μM) and Strychnine (1 μM) primarily negate amacrine inhibition onto BC axons. **b** Representative glutamate responses before (black) and after (blue) blockage of horizontal (top) or amacrine (bottom) inhibition. **c, d** The mean peak glutamate fluorescence (**c**), as well as rise

and decay times (**d**) normalized by full-field motion. Control responses for both cocktails were combined. $^{\#}p < 0.05$; $^{\#\#\#}p < 0.001$ CNQX + HEPES vs. other conditions. **e** Mean static edge enhancement (top) and novel motion detection (bottom) calculated from the peak of the responses in control, with amacrine ($n = 9$ animals, 7 females, ages p43-166, 83 ROIs) and horizontal ($n = 8$ animals, 4 females, ages p44-102, 105 ROIs) cell blockers. $^{\#}p < 0.05$; $^{\#\#\#}p < 0.001$ change from 0% (two-tailed $t$ test with Bonferroni's correction). $^{***}p < 0.001$ difference between pharmacological manipulations. Unless specified, statistical tests were one way ANOVA followed by Tukey test with Bonferroni's correction. Error bars-SD. See Supplementary Tables 3, 4 for detailed statistics.

moving bars (Fig. 3b); this is the expected result of a full surround blockage (see below, Fig. s6). A further effect of the HC blockers was seen as diminished amplitude of glutamate release near mask-stimulus boundaries (Fig. 3b, c, e). The blockers had a minor effect on the temporal kinetics of the signals, reaching statistical significance only for the rise time of responses to motion towards the edge (Fig. 3d). Importantly, emerging and exiting stimuli elicited responses with similar amplitudes and dynamics, in sharp contrast to the emerging motion enhancement observed in control conditions (Fig. 3c–e). These results indicate that the critical step of motion processing and novel object enhancement relies on HC feedback and is therefore performed in the first retinal synapse.

**Simulations confirm that horizontal cell-mediated surround can drive diverse responses to emerging motion**

Our pharmacological manipulations suggest that the fundamental properties of the RF surround that govern responses to motion are likely to be mediated by horizontal but not amacrine cell inhibition. Given the marked differences in responses to moving stimuli and novel object enhancement properties between glutamate clusters (Figs. 2, 3), we wanted to understand whether a single source of surround

inhibition, shared across all BCs, can lead to a diverse representation of motion in the bipolar population. To address this question, we constructed a computational model of visual processing in the outer plexiform layer (Fig. 4a–d). We activated the simulation with stationary and moving bars and recorded the resulting signals in photoreceptors, horizontal and bipolar cells.

Intriguingly, the simulation revealed a possibility for a pronounced representation of emerging objects already in photoreceptors over a wide range of simulation parameters (Fig. 4b–d), but only when HC inhibition was intact (Figs. 4c, d, s7). This outcome relied on the lag between activation times of photoreceptors and horizontal cells; at the location of object emergence, HC engagement coincided with photoreceptor activation (Fig. 4b). Elsewhere, the initiation of HC signal preceded direct light-induced photoreceptor activation by as much as ~100 ms and correlated with diminished photoreceptor output (Fig. 4b).

While our model incorporated nonlinear interactions between cells and synaptic inputs, a similar temporal relationship in RF activation was readily observed in a linear center-surround architecture (Fig. s8). In both models, established (continuous, exiting) motion recruited the RF consecutively because moving objects encounter the surround

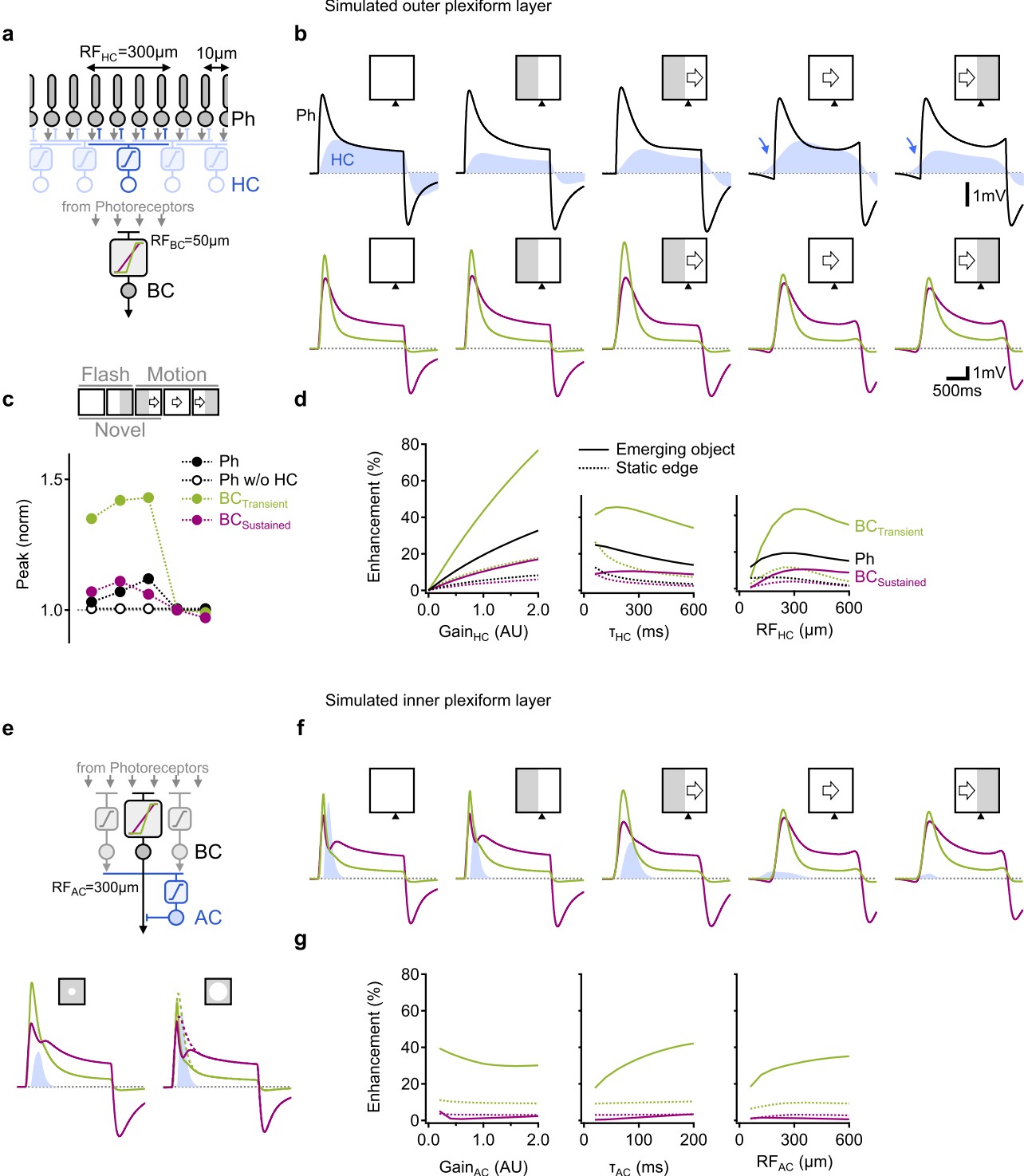

**Fig. 4 | Simulated visual processing in outer and inner plexiform layers capture empirically observed differences in the effects of horizontal and amacrine cell inhibition on novel object enhancement and population dynamics. a** The circuit modeled in **b**–**d**. Photoreceptors (black) combined the visual input with feedback inhibition from horizontal cells (blue). Transient (green) and sustained (red) BCs differed by the formulation of their dependency on the photoreceptor input. **b** Top, photoreceptor (black) and HC (blue) voltage responses (inverted for presentation purposes) elicited by the visual stimuli. Note the effect of preceding inhibition (blue arrows) that is present during established motion. Bottom, the shapes of the BC potentials for the same stimuli. **c** The peak amplitude of the simulated visual responses, normalized by the response to the full-field moving bar. In the absence of horizontal feedback, photoreceptor responses were similar across all probed stimuli (open circles and dotted lines). **d** Enhancement of the emerging object (solid) and the static edge (dotted) as a function of different model parameters. **e** Top, ACs were included in the simulated circuit. Amacrine cells were stimulated by a separate subpopulation of BCs (gray). Bottom, BC and AC (blue) activation by narrow (diameter = 100 μm, top) and wide (diameter = 500 μm, bottom) spots, used to calibrate the strength of AC surround. Dotted, responses in the absence of inhibition in the inner plexiform layer. **f**, **g** as in **b**, **d**, in the presence of ACs.

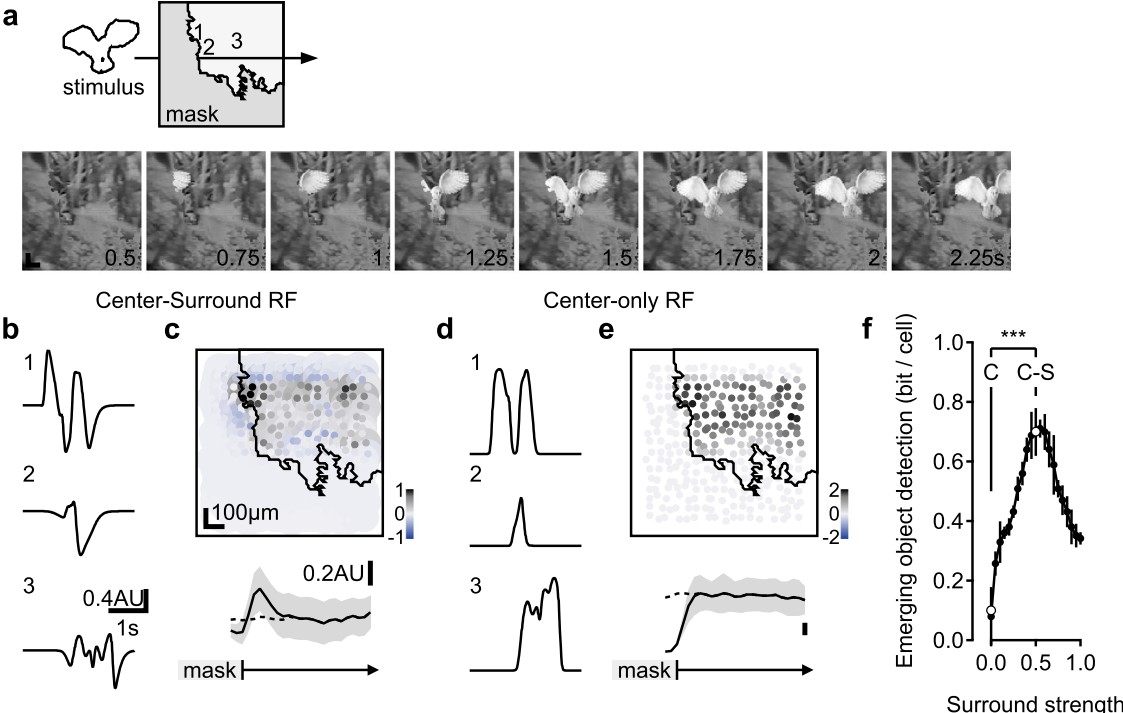

**Fig. 5 | Novel object detection by linear center-surround RFs under natural movies.** Simulated neuronal responses to a movie showing predator appearance (**a**). **b** Temporal response profile of three sample cells with a linear center-surround RF formulation at the spatial coordinates shown in **a**. **c** Top, the peak response amplitude to stimulus motion from a population of simulated neurons. Activation is maximal near stimulus emergence. Bottom, the mean (±SD, shaded) change in RF activation vs. distance from the mask (*n* = 1000 permutations of the background, the horizontal scale is preserved for both plots). Dashed trace, responses in the absence of the mask. **d**, **e** As in **b**, **c**, with the surround component removed from the RF description. **f** The mean(±SD) mutual information computed from the differences in responses of individual neurons located near (<100 μm) the location of stimulus emergence to simulations in the presence or the absence of the mask. The bottom axis indicates the strength of the surround relative to the center. C Center-only RF, C-S Center-Surround RF properties depicted in **b**, **c** ****p* = 10⁻¹⁰ between the two RF architectures (two-tailed *t* test).

first[51]. Receptive field components were engaged more synchronously by emerging stimuli and activated simultaneously by stationary flashes (Fig. 4b). In general, the encoding of existing objects was accompanied by a longer temporal delay between the initial activation of the surround and subsequent center stimulation. Due to this delay, surround inhibition is more developed by the time the center is engaged by the stimulus and is, therefore, more likely to suppress responses to continuous motion.

Further validation for this model was seen in experiments where we presented either a narrow (25 μm wide) moving bar or shuffled the stimulus frames and showed them out of order (Fig. s9)[3,24]. All ROIs responded to bar presentation over the center of their RF, but the amplitude of the response was dependent on the activation history of the surround. If stimulation of adjacent positions (presumably in RF surround) preceded center stimulation, response amplitude was similar to responses recorded in apparent motion trials. Otherwise, the responses were on average(±SD) 42 ± 27% larger (Fig. s9).

Focusing on the factors that influence BC response dynamics, we note that transient kinetics in BCs were mediated by faster neurotransmission, but also, unexpectedly, by elevated sensitivities to photoreceptor release (Fig. s10). The higher threshold required for effectual activation increased the sensitivity of transient BCs to small fluctuations around the peak photoreceptor activity. In agreement with recent findings[26], our model suggests that transient BCs receive a more rectified, nonlinear copy of the photoreceptor signal and predicts that such nonlinearity creates a substrate for more distinct responses to motion vs. stationary stimuli and promotes the enhancement of novel object emergence (Figs. 4c, d, s10).

## Simulations replicate the experimentally observed minor role of amacrine cell inhibition on generating emerging object enhancement in BCs

The simulations presented above confirm the ability of signal processing in the outer plexiform layer in mediating the sensitivity of BCs to novel objects. Could the model also provide insights as to why amacrine cell inhibition, which is an integral part of bipolar cells' surround[12,15,52–54], does not have a significant impact on motion processing? A complicating factor in addressing this question is the fact that in contrast to a single type of HC, the murine retina contains ~50 different types of ACs, whose interactions with BCs are largely unknown[55]. Therefore, we decided to implement a generic model of a single homogenous AC population to examine the general principles of visual processing in the IPL, where BC-AC interactions take place. Amacrine cells were modeled as a component of a feed-forward circuit; they were stimulated by a separate BC subpopulation (Fig. 4e, Methods) and, in turn, inhibited the transient and sustained BCs described above. To ease the comparisons between horizontal and amacrine cell-mediated surrounds, we initially matched the size of horizontal and amacrine cells' receptive fields (Figs. 4e, s11). The strength of the AC inhibition was set to recreate the reported effect of AC surround on altering the shape of the response to a wide-field but not narrow-diameter spot stimulation (Fig. 4e)[15].

As we explored the parameter space of AC dynamics, we could observe diverse and sometimes opposite effects on simulated motion processing in BCs. Over a wide range of examined parameters, motion processing was not substantially altered by the amacrine drive (Fig. 4f, g), despite the ability of the latter to modify bipolar cells' kinetics (specifically to full-field stimulation, Fig. 4e, f). We also found that inhibition in the INL could enhance or diminish emerging object

representation. For example, models with faster amacrine cells were more likely to reduce the edge effects (Figs. 4g, s11). The main factor accounting for the impact of ACs was the amplitude of the AC drive during motion (Fig. 4f). As AC signals are dependent on computations in presynaptic BCs, they were likely to inherit bipolar cells' motion sensitivity (Fig. s11)[40]. Perhaps counterintuitively, enhanced motion effects were observed in models with sustained BC populations stimulating the AC circuit (Fig. s11). In these simulations, the degree of AC activation was similar for all phases of motion, and their effect was comparable to the one observed in HCs (Fig. s11).

### Enhanced representation of novel stimuli under natural movies requires center-surround organization

Next, we asked whether the fundamental properties of the center-surround RF architecture are sufficient to identify novel objects under realistic visual conditions. To address this question, we simulated responses from a population of linear center-surround neurons (Fig. s8) to movies showing the appearance of predators in a natural mouse habitat (Fig. 5a). The strength of the surround was set to 50% relative to the center to reproduce the intensity of the center-surround interactions we identified in the full retinal model and reported in the literature[15,23,40]. Although the simulated cells lacked nonlinear signal processing mechanisms, we found these cells capable of generating a rich representation of dynamically changing scenes. Cells responding to established motion encoded the local contrast differences between the stimulus and the background (Fig. 5b, c). Comparable to our findings presented above, stimulus emergence correlated with robust responses (Figs. 5b, c, s12). Interestingly, novel motion enhancement was evident mainly at the initial site of stimulus appearance (the wing in the example shown in Fig. 5), implying a spatial focus for novel object detection spanning about $100\,\mu m$ of retinal space, comparably to empirical observations (Fig. s4).

Using the simulation, we were able to test the contribution of the surround to this computation. We reformulated the receptive field description for the tested population to exclude the surround. We found that the outputs of the cells in this simulation were still tuned to the local contrast (Fig. 5d, e). However, the response amplitudes were similar for continuously moving and emerging stimuli, indicating that similar to our findings in the simulated retinal circuit, novel object detection required surround participation (Figs. 5e, s12).

What is the benefit of utilizing the center-surround architecture to compute novel object appearance in a realistic environment? Stronger activation near the mask-stimulus boundary can be beneficial for detecting stimuli in downstream neurons. To quantify the information that is encoded by individual neurons in our simulation, we measured the mutual information from responses of cells at the location of stimulus emergence. Analysis of signal entropies calculated from the peak responses to continuous and novel motion revealed that each cell is capable of transmitting $0.7\pm0.06$ bits in each trial (Fig. 5f). Comparable information levels were found for responses in cells near vs. far ($>200\,\mu m$) from the stimulus emergence region within the same simulation trial (data not shown).

A similar analysis in center-only neurons failed to find evidence of information transfer about novel object appearance (mutual information $= 0.08\pm0.1$ bits/cell, $p > 0.6$ vs. 0, Fig. 5f), suggesting that in this scenario, postsynaptic circuits have to employ different processing schemes to detect the presence of new objects.

To examine the intensity of the surround required for novel object detection, we varied the strength of the surround between simulation trials. As before, all simulated cells had identical RF parameters. As expected, novel object information conveyed by the RF activation grew proportionally with the intensity of the surround, up to a peak at surround/center ratio of about 0.4–0.6 (Fig. 5f). More intense surround produced a reduction in the computed mutual information values due to a pronounced

hyperactivation of the modeled cells upon stimulus entrance to their RF (Fig. 5f).

### Edge effects influence the analysis of motion processing in the retina

Given the participation of BCs in novel motion detection, we asked whether the dependence of BC signals on the direction of motion near mask-stimulus boundaries impacts the computation of direction selectivity (DS). Starburst amacrine cells are a class of retinal interneurons that have a prominent role in the processing of directional information in their dendrites, which are tuned to detect stimulus motion towards dendritic tips (Fig. 6a)[36,41,56–58]. Despite intense effort, explaining the biological implementation of this computation remains elusive[20,36,59–64]. A common strategy to probe direction tuning in SAC is by isolating dendritic computations[8,65] with visual protocols structured to stimulate a part of the SAC[36,61,64,66]—effectively masking part of the stimulus (Fig. 6b). To explore whether the glutamatergic drive to SACs is affected by the mask-stimulus boundary, we reexamined iGluSnFR fluorescence in SACs in response to the battery of visual stimuli used above (Fig. 6b) in a field of view set to match the span of BC innervation of a single SAC dendrite (Fig. 6b, $\sim80\,\mu m$)[36,64].

As expected from the dynamics of the functional release clusters mediating the bipolar drive to SACs (Figs. 1, 2), glutamate responses were more pronounced for emerging stimuli (Fig. 6c, d). To measure the impact of novel object preference on the estimation of DS in SACs, we decided to quantify the difference between responses to motion from an edge to motion towards the edge with the direction selectivity index (DSI). Direction selectivity index is ubiquitously used to analyze directional tuning in SACs and other direction-selective cells. We found that the mean ($\pm$SD) direction selectivity index computed from responses to moving bars with the direction of motion towards/away from the boundary at the center of the display was $31\pm19\%$ ($p < 0.001$ vs. 0, $t$ test, $n = 141$ ROIs, 16 animals, Fig. 6e), while full-field moving stimuli evoked comparable glutamatergic responses in all directions (Fig. 6d, e).

Could the edge effect observed in the presence of a mask-stimulus boundary contribute to DS computations in SACs? A simple thought experiment suggests that the answer is no. Figure 6b left, illustrates a morphology of a hypothetical SAC (recorded in a separate experiment) whose cell body happens to lie near the mask boundary. The enhancement of glutamatergic drive aligns with the preferred dendritic axis in this cell and can contribute to the DS computed in SAC dendrites (Fig. 6b, e "Edge near soma"). However, signals to SACs in less optimal configurations are in the "wrong" direction. The gray-colored SAC morphology shown in Fig. 6b serves as an example of a cell whose soma is located deeper in the stimulated region yet proximal enough to extend its dendrites over the mask ("Edge near tips"). Assuming that both SACs receive the same copy of the BC drive (prior work had shown that SACs indiscriminately sample available BC terminals near their dendrites[20,36,37], but see ref. 67), it is possible to compute a new DSI—from the perspective of the "edge near tips" cell. With the direction of motion away from the mask-stimulus boundary and towards the soma of this cell, higher response amplitude to emerging motion lead to a reversed directional tuning (Fig. 6e "Edge near tips"), in contrast to what is expected of a proper directional mechanism[40].

## Discussion

Using the murine retina as a model system, we were able to investigate the properties of motion processing in spatially symmetric center-surround RFs. We found that the representation of continuous motion was associated with reduced peak amplitudes and prolonged temporal dynamics of glutamate signals compared with sudden object appearance in most recorded ROIs. Motion responses could not be reliably predicted from the dynamics of responses to stationary flashes,

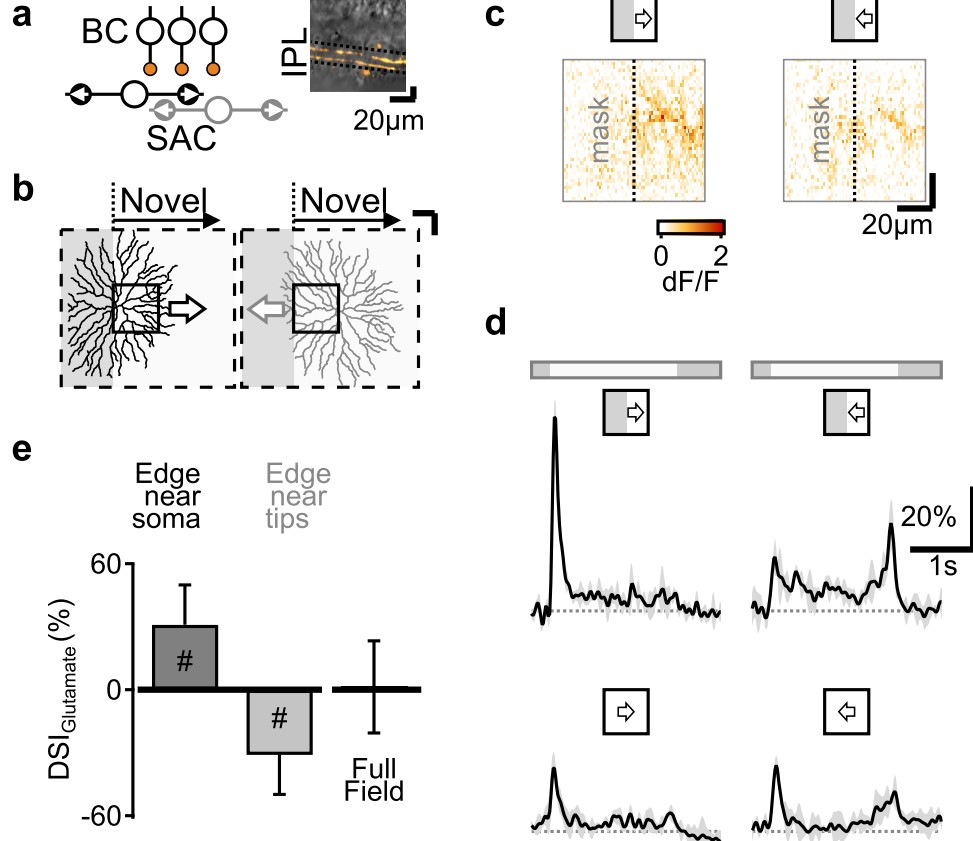

**Fig. 6 | Sensitivity to novel objects in BCs could influence the analysis of motion processing in postsynaptic circuits. a** Starburst amacrine cells (SACs) integrate BC signals to detect motion towards dendritic tips. Inset, a vertical projection of a 2-photon z-stack showing floxed-iGluSnFR expression (orange), restricted to SACs in a ChAT-Cre mouse. **b** Schematic of novel emergence enhancement vs. dendritic direction preference (arrows) in two exemplar SACs (reconstructed in a separate experiment). Solid rectangle, the imaging window, whose size and the position were set to approximate the span of BC innervation of a SAC dendrite. **c** Example peak iGluSnFR fluorescence in the imaging window evoked by moving bars in the presence of an occluding mask. **d** Top, the temporal evolution of the glutamate signals in same trials as in **c**. Bottom, responses to full-field moving bars from the same imaging window. Data are presented as mean values ± SD, shaded. **e** The peaks of the glutamate signals recorded as in **c**, **d** in 28 different imaging windows were used to estimate the mean (±SD) direction selectivity index (DSI) of the excitatory drive for SACs whose soma is located near the mask-stimulus boundary ("Edge near soma", black) and cells with their cell body away from the visual edge as illustrated in **b** ("Edge near tips", gray). DSI of the responses for full-field stimulation is shown for comparison. #$p = 2 \times 10^{-7}$ vs. zero, two-tailed $t$ test. $n = 16$ animals, 10 females, ages p57-359, 141 ROIs.

indicating that subpopulations of glutamate releasing cells convey different temporal features of the stimulus contingent on the presence of motion (Fig. 1). In the retina, visual processing is thought to be facilitated by parsing the sensory input into parallel information channels at the level of the BCs[12]. According to the literature, these communication channels represent different transformations of the photoreceptor signals and typically emerge from the underlying neuronal infrastructure. For example, luminance and chromatic selectivity arise from specific targeting of bipolar dendrites to distinct photoreceptors, and response polarity depends on the composition of glutamate receptors. The processing dynamics of static and moving stimuli are distinct, despite occurring in the same neuronal circuits, representing an additional layer of complexity present in the retina. The multi-layered decomposition of the visual scene could potentially reduce the number of cells required for effective visual processing but complicates the analysis of motion responses from stationary stimuli, as is discussed in more detail below.

Based on the examination of glutamate release in the presence of static masks, we propose a functional role of circularly symmetric center-surround RFs as detectors of novel objects. This property is a logical but previously underappreciated consequence of the classic center-surround RF formulation. The mechanistic explanation for this function is straightforward and relies on the sequence of RF activation

by the stimulus. Continuously moving stimuli always enter the surround RF region first, driving the surround towards a more effective inhibition by the time the center is engaged. The priming effect on the surround is weaker or absent in emerging and suddenly appearing objects. According to numerical simulations, the resulting enhanced representation of emerging objects is expected over a wide parameter range and is present even in a linear RF formulation (Figs. 4, 5).

Our experiments and models show that photoreceptors and horizontal cells are the only circuit elements required to generate motion responses in BCs (Figs. 3, 4). The major step in the computation of object motion already occurs at the first synapse in the retina, and the diversity of motion responses in BCs depends on their signal transformation. The most pronounced difference in the representation of established vs. emerging motion was seen in transiently releasing glutamate clusters (Fig. 2). A recent study had shown that signal transformation from the photoreceptors to BCs could be nonlinear and that the degree of nonlinearity is larger for transient BCs[26]. Why is processing linearity correlated with the shape of the response? Our model of signal integration in the outer plexiform layer suggests a possible answer. Nonlinear signal transformation at the photoreceptor-BC synapse could impose a threshold on the amplitude of the photoreceptor output that is required for effective activation of the postsynaptic cell (Fig. s10). As photoreceptors typically respond to

light onset and light offset with a rapid membrane potential fluctuation[68,69], nonlinear BCs are more likely to be disproportionally sensitive to these phases of photoreceptor release; their fast temporal dynamics reflect the transient shape of the filtered photoreceptor output they sample. Meanwhile, more linear signal processing mirrors the original shape of the photoreceptor light response (Figs. 4, s10). The exact biological implementation of the nonlinear photoreceptor-BC synapse dynamics is currently unclear but could plausibly be mediated by a differential affinity of BC dendrites to photoreceptor release[70]. In the end, the nonlinear nature of the transient BC population is known to contribute to a rudimentary feature detector-like behavior that is tuned to certain visual conditions, such as signal polarity and spatial inhomogeneity[26]. We can now add novel object appearance to this list.

We used stimuli that were explicitly designed to compare responses to moving and static objects and representation of novel vs. existing visual items. Recent studies proposed a role for gap-junction mediated priming of BC center and threshold nonlinearity of BC release in promoting responses to expanding motion (Fig. 7)[4,24]. These specialized mechanisms allow the center RF region to perform a motion computation. In a parallel project, Strauss et al. proposed that sensitivity to looming (approaching) stimuli can be conferred by center-surround interactions[40], showing that multiple mechanisms can contribute to processing of established motion in BCs.

Several studies suggested that responses to motion are affected by gain-control mechanisms in the retina (Fig. 7)[7,10]. In addition to inhibitory surrounds, likely biophysical substrates for gain control are synaptic depression in photoreceptors[68] and BCs[71,72], calcium channel inactivation[73], and the action of glutamate transporters[74]. The combined effects of these mechanisms are reflected in the temporal model kinetics. As we have shown, these properties affect the strength and the time scale of BC signal during full-field motion but are not sufficient to explain signal enhancement observed during object entrance. Our compact model formulation uses a small number of free parameters to capture multiple gain mechanisms; future studies are required to distill the roles of distinct biophysical structures in mediating BC gain in diverse visual conditions.

Pharmacological blockage of photoreceptor-HC drive with CNQX + HEPES readily abolished edge effects on motion responses (Fig. 3), but HEPES alone was ineffective (Fig. s6). Based on the literature, this finding could be explained by more effective impact of CNQX on HC feedback[75], or the membrane potential of HCs in our preparation, which affects the efficacy of pH buffering by HEPES[42].

In our hands, amacrine cells were not required for the enhancement of novel visual items in BCs. However, the pharmacological experiments were not designed to dissect the contribution of ACs to computations in specific BC subtypes or other motion computations. Diverse AC types make specific, vital contributions to motion computations in RGCs. In agreement with our colleagues[40], our numerical models propose that different amacrine cells may inherit distinct motion sensitivity capabilities from BCs (Figs. 4, s11). For example, we have shown that the presence of visual edges is a potent modulator of the glutamatergic drive to SACs. Because SAC-stimulating BC types also contact direction-selective ganglion cells[31], the latter should exhibit distinct responses to motion in the presence of occluders. A sensitivity to interrupted motion was indeed observed in a recent study[76]. Whether other cells in the visual system preferentially respond to novel objects remains to be elucidated. The mechanisms promoting this computation in the early visual system are not unique to BCs, and elevated responses to emerging objects can be computed de novo in neurons with spatially symmetric center-surround RFs even in the absence of direct BC-mediated novel object enhancement.

While we emphasize the role of the surround in mediating motion computations, previous work highlighted the contribution of gap junctions to motion processing. Electrical interactions between BCs

could prime RF center activation, augmenting responses to spatio-temporally correlated stimuli, such as smooth motion, especially if BC response profile is nonlinear[3,4,24] (Fig. 7). The difference in the results in the literature and our experiments (Fig. s9) is most likely due to a weaker engagement of the surround in prior work due to lower illumination levels (photopic here, scotopic-low mesopic in ref. 24) or narrower bar widths used for the stimulation of the primate retina in ref. 3.

In the last decade, several groups found evidence for a spatial offset between presynaptic BC populations that are aligned with the directional axis in direction-selective ganglion cells and dendritic position in SACs[20,22,36,37]. This circuit organization can support directional tuning by a mechanism first described by Hassenstein and Reichardt[77] (Fig. 7)—if the response speed of the BCs follows their spatial arrangement. Conflicting results were reached in studies designed to test the predictions of this model using electrophysiological and imaging approaches[21,64,78]. Importantly, all previous work examined bipolar output in response to the presentation of stationary inputs, which, as our results indicate, do not accurately reflect the dynamics in BCs during motion[20–22]. Proposed directional computations are particularly dependent on BCs rise and decay times, which, as our data reveal, are weakly correlated between moving and static objects (Fig. 1). At the very least, the dramatic increase in the rise time dynamics we observed in our recordings suggests a shift in speed dependence of the Hassenstein-Reichardt detector to slow-moving objects. Further experiments will be required to resolve this issue and elucidate the potential impact of visual edges on directional preference (Fig. 6).

A recent study found that a small fraction of BCs that target the DS circuit exhibit direction-selective tuning[67]. Neither this nor previous studies that recorded motion responses from BCs[79,80] have noted tuned axonal terminals. We agree with ref. 67 that the observation of directional tuning in axonal microsegments requires sparse labeling of specific BC subtypes and specialized experimental design. These DS signals rely on SAC feedback and represent an additional complexity in BC motion processing, thus showing how AC drive can influence specific synapses in the retina, potentially in addition to motion processing abilities mediated by HCs (this study) or interactions with other BCs[3,4,24].

Similar to the effect of the center-surround antagonism on forming illusory enhancement of perception around edges[81], our findings of motion processing in the early processing stages in the retina have intriguing psychophysical implications to the perception of novel stimuli over continuing motion and echo the salience of visual perception in humans[79,80]: the sudden appearance of new objects grabs attention reflexively; motion onset is less salient—but more noticeable than continuous motion. Our data propose that these computations are hard-wired in the retina and reflect the information content conveyed by the respective visual items. From an ecological perspective, the utility of continuous retinal motion is diminished as it may be self-generated by locomotion through the environment and because the trajectory of continuously moving objects could be predicted by past sensory input. Conversely, novel stimuli can alert to a predator or prey; their fast processing is vital to survival. All the necessary machinery for motion processing in BCs we describe in the mouse are conserved in primates, providing strong evidence that enhanced representation of newly flashed and emerging moving objects are consequences of a bottom-up process fundamental to how visual stimuli are computed in the retina.

Taken together, our work complements previous studies revealing decorrelation of signals by surround inhibition[15,25,82] and shows how simple operational concepts give rise to complex visual computations. Diverse representation of different features of the visual space in a single neuronal population and early detection of salient environmental cues are powerful strategies that reduce the computational

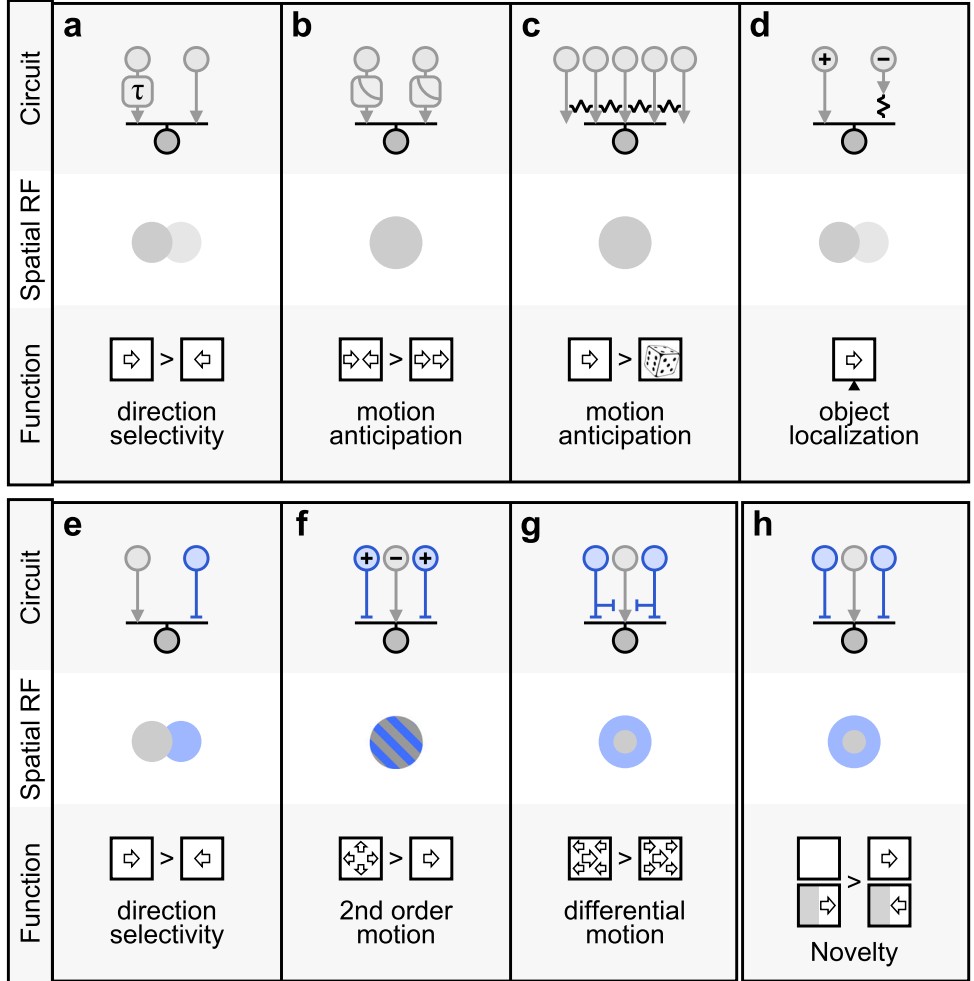

**Fig. 7 | Circuits for motion computations.** Illustration of some of the known motion processing circuits in the vertebrate retina. **a** The computation of direction selectivity in a Hassenstein-Reichardt correlator[77] is mediated by the integration of spatially separated inputs with distinct temporal dynamics (slower arm denoted by τ)[20–22,86,87]. **b** Contrast adaptation (gain-control) supports predictive encoding of moving objects[10], increased responsiveness to motion onset[7] and detection of motion reversal[88]. Electrical coupling among BCs and RGCs was shown to compensate for the lag in visual processing[89], increase sensitivity to correlated input vs. randomly shuffled stimuli[3,4,24,27] (**c**), and assist in object localization[90] (**d**). **e–h** Diverse contributions of inhibitory circuits (depicted in blue) to motion responses.

**e** A spatiotemporal offset between excitation and inhibition is the basis of the Barlow-Levick model of direction selectivity[91] found in many DSGCs[92–94]. **f** Integration of presynaptic excitatory/inhibitory units with similar spatial receptive fields but reversed polarities (marked by "+" and "−" signs) facilitates detection of second-order[95] and approaching[5] motion. Previous studies identified a role for the center-surround organization in the segregation of local vs. global motion[1,2,28–31] (**g**). **h** We and colleagues[40] reveal differential responsiveness to emerging and exiting objects in cells with a center-surround receptive field architecture, which could facilitate identification of novel items in the visual scene.

burden of the visual system. The surprising finding that the circularly symmetric center-surround RF architecture is sufficiently versatile to take part in seemingly unrelated tasks is critical to the understanding of visual computations in multiple brain regions and the design of future studies of visual perception.

## Methods

### Virus expression

All animal procedures were conducted in accordance with U.S. National Institutes of Health guidelines, as approved by the University of Colorado Institutional Animal Care and Use Committee (IACUC). Mice were housed in a 12 light /12 dark cycle at room temperature (~22 °C), 40–60% relative humidity. For intravitreal virus injections, mice of ages p40–360 of either sex were anaesthetized with isoflurane; ophthalmic proparacaine and phenylephrine were applied for pupil dilation and analgesia. A small incision at the border between the sclera and the cornea was made with a 30 gauge needle. 1 μL of AAV solution was injected with a blunt tip (30

gauge) modified Hamilton syringe (www.borghuisinstruments.com). AAV9.hsyn.iGluSnFR. WPRE.SV40, (a gift from Loren Looger, Addgene plasmid # 98929; RRID:Addgene_98929; 10¹³ vg/mL in water) was injected into the vitreous humor of wild type mice (C57BL/6J, Jackson laboratory, www.jax.org). To express iGluSnFR in SACs only, AAV9.hsyn.FLEX.iGluSnFR.WPRE.SV40 (a gift from Loren Looger, Addgene plasmid # 98931; RRID:Addgene_98931, similar concentration) was used in ChAT-Cre transgenic mice. AAV9-pGP-AAV-syn-jGCaMP7f-WPRE (Addgene plasmid # 104488; RRID:Addgene_104488) was used to measure intracellular calcium levels. Experiments on retinas from all animal groups were performed 2–6 weeks following virus injection.

### Imaging procedures

Mice were not dark-adapted to reduce rod-pathway activation. Two hours after enucleation, retina sections were whole mounted on a platinum harp with their photoreceptors facing down, suspended ~1 mm above the glass bottom of the recording chamber. The retina

was kept ~32 °C and continuously superfused with Ames media (Sigma-Aldrich, www.sigmaaldrich.com) equilibrated with 95%$O_2$/5%$CO_2$.

## Light-stimulation

Light stimuli were generated in Igor Pro 8 (Wavemetrics, www.wavemetrics.com) PC and displayed with a 415 nM LED collimated and masked by an LCD display (3.5 Inch, 480 × 320 pixels, refresh rate of 50 Hz) controlled by a custom-written python script running on raspberry pi 3 computer. Display luminosity was gamma corrected with a powermeter (Thorlabs, www.thorlabs.com); the stimulus was set to either 60% or −60% Michelson contrast. Frame timing was controlled by a clock signal from Sutter IPA patch-clamp amplifier (Sutter Instruments, www.sutter.com) driven by Igor Pro and read from one of the digital I/O ports of the raspberry pi. Light from the visual stimulus was focused by the condenser to illuminate the tissue at the focal plane of the photoreceptors (resolution = 2.5 μm/pixel, background light intensity = 30,000–60,000 R* rod$^{-1}$). Both vertical and horizontal light stimulus positions were checked and centered daily before the start of the experiments. The following light stimulus patterns were used: static bar covering the entire display (800 × 800 μm) presented for 2 s. A 1 mm-long bar moving either to the left or the right directions (speed = 0.5 mm/s; dwell time over each pixel = 2 s). These stimuli were repeated with masks (at background light levels), spanning the full height of the display, occluding different portions of the stimulus. In some experiments, we flashed 25 × 350 μm bars for 50 ms. The bars were presented in 14 possible spatial positions in a 350 × 350 μm square. The presentation was either in sequential positions (creating apparent bar motion in the left or right directions) or out of order, in seven precomputed pseudorandom patterns. Each visual stimulation protocol was repeated at least 3 times.

## Imaging

Glutamate and calcium imaging was performed with Throlabs Bergamo galvo-galvo two-photon microscope using the Thorimage 4.1 acquisition software (Throlabs). A pulsed laser light (920 nm, ~1 μW output at the objective; Chameleon Ultra II, Coherent, www.coherent.com) was used for two-photon excitation projected from an Olympus 20X (1 NA) objective. A descanned (confocal) photomultiplier tube (PMT) was used to acquire fluorescence between 500 and 550 nm. The confocal pinhole (diameter = 1 mm) largely prevented stimulus light (focused on a different focal plane), from reaching the PMT, allowing us to present the visual stimulus during two-photon imaging. A photodiode mounted under the condenser sampled transmitted laser light to generate a reference image of the tissue. Fluorescence signals were collected in a rapid bidirectional frame scan mode (128x64 pixels; ~50 Hz, Thorimage). The line spacing on the vertical axis was doubled to produce a rectangular imaging window (typically ~82 × 82 μm size, in some experiments, the window was set to ~164 × 164 μm; the corresponding pixel sizes were 0.64 μm or 1.28 μm). To reduce shot noise, images were subsampled by averaging 2 × 2 neighboring pixels and filtered by a 20 Hz low pass filter offline. Horizontal and vertical image drifts were corrected online using a reference z-stack acquired before time-series recordings.

For pharmacological manipulations, we used SR95531 (50 μM, Abcam, www.abcam.com) to block GABA$_A$ receptors, TPMPA (50 μM, Tocris, www.tocris.com) to block GABAC receptors and strychnine (1 μM, Abcam) to block glycine receptors. CNQX, a blocker of AMPA and Kainate receptors was purchased from Alomone labs (www.alomone.com) and HEPES, a pH buffer was purchased from GoldBio (www.goldbio.com). All drugs were mixed with the bath Ames medium. In some instances, GABA/Glycine blockers application elicited high level of background activity that was not synchronized with the visual stimuli.

Because these aberrant responses could corrupt the measurement of light responses, we excluded ROIs with a coefficient of variation >1 from the analysis.

## Analysis

Unless specified otherwise, analysis was done in Igor Pro 8. Fluorescence signals were averaged across repeated visual protocol presentations. Pixels with dF/F values >20% were selected for clustering analysis. For the initial clustering of ROIs with similar response kinetics, we combined 1 s recordings of the response shapes around the time of stimulus entrance to the imaging window from each of the tested visual protocols across all imaged planes. A similarity matrix was constructed from a pairwise pixel comparison measured with Igor build-in farthest-point clustering algorithm. The shapes of the resulting ROIs were fitted with a sigmoid for the rising phase of the response and with a single exponential for the decay phase. ROIs were manually curated and removed from analysis if pixel variability, measured with a coefficient of variation, exceeded 1.

We computed the horizontal RF position from responses to motion over the entire display. We first determined the timing of 50% rise-time from trials with leftward and rightward motion. ROIs with their RF center in the middle of the display should respond to both stimuli at the same time following stimulus presentation. In an ROI where the center of the RF is located to the left/right of the display center, a rightward moving stimulus elicits a response that comes earlier/later compared to a trial with a leftward moving stimulus. RF position was computed as half the time difference between the diametrically opposed trials, multiplied by stimulus speed. Trial responses were considered to be to full-field stimulation if the RF center was at least 100 μm away from the nearest visual edge formed either by masks or the boundaries of the display. Similarly, responses were considered to be near an edge if at least one of the visual edges was closer than 50 μm to the RF center.

To detect similarly shaped groups between different experiments, we conducted a secondary hierarchical clustering in R v3.6 (R foundation for statistical computing). Our initial clustering incorporated responses from trials with moving stimuli and responses near visual edges. Motion responses shift in time as the stimulus progresses over the retina, making comparisons between ROIs difficult. Edge effects may also affect the shape of the responses. For these reasons, as an input to the similarity matrix, we performed a pairwise comparison between 1-s long responses to full-field static stimulation only, for positive contrast stimuli presentation for ON groups and negative stimuli for the OFF groups. The optimal cluster number was determined with the c-index analysis. For ON groups, the clustering revealed local minima at 2, 8, and 16 clusters. From visual inspection, it was clear that the dataset is composed of more than two response clusters, and that at 16 clusters, the algorithm produced several groups with virtually identical dynamics. In addition, because both 2 and 16 cluster values fall outside of the biologically plausible number of ON-polarity glutamate releasing cell types, we did not consider these minima as reliable indicators of functional BC subtypes.

Transiency index (TI) was calculated as the ratio between the peak and the mean of the response within the stimulation window. TI = 1 indicates a sharp and transient response, TI close to zero is produced by sustained plateaus.

Static edge enhancement was computed as

$$Edge\ enhancement = \frac{R_{Edge}}{R_{FF}} - 1 \tag{1}$$

where $R_{Edge}$ and $R_{FF}$ are the peak dF/F responses to masked and full-field static flashes. Similarly, emerging object enhancement was

computed as

$$Emerging\ object\ enhancement = \frac{R_{From\ edge}}{R_{To\ edge}} - 1 \qquad (2)$$

where $R_{from\ edge}$ and $R_{To\ edge}$ are the peak dF/F responses to motion from and towards the mask, respectively.

Direction Selectivity Index (DSI) was calculated as a vector sum of vectors $V_i$ pointing in the direction of the stimulus and having the length $R_i$ = peak dF/F of the response to that stimulus:

$$DSI = \frac{\sum_{i=1}^{n} V_i}{\sum_{i=1}^{n} R_i} \qquad (3)$$

Where $n$ is the number of probed directions. DSI can range from 0 to 1, with zero indicating no directional preference and 1 indicating responses to only one direction of stimulation.

Bonferroni correction was used for multiple comparisons. Whenever ratios between parameters were compared, statistics were computed on a logarithmic transformation of the data.

IPL depth was measured from the transmitted light channel extracted from the z-stack taken of the entire width of the retina that accompanied all functional recordings. The curvature of the retina was corrected by measuring the height of the inner limiting membrane at the four corners and the center of the image stack and fitting a curved plane that crossed these 5 points. The total width of the IPL was measured at the center of the z-stack from the transmitted light channel for each animal and recording region. A similar approach was used to measure the curvature of the retina in ChAT-Cre/tdTomato mice. In these cases, the fluorescence on the red channel was used to mark the location of the ChAT bands. Both approaches provided similar anatomical estimates and were used interchangeably to measure the depth of the recorded ROI in the IPL.

## Modeling
All simulations were conducted in Igor Pro 8.

## Linear receptive field model
We simulated a simple spatiotemporal RF structure to examine the engagement of a cell with a center-surround RF organization by visual motion. The spatial extent of the center and surround RF components were defined by a two-dimensional Gaussian function with half widths of 50 and 200 μm, respectively. The responses for the RF components were modeled as a single exponential with a time constant (τ) of 20 ms for the center and 100 ms for the surround[47,75]. The simulation ran for 4500 ms with a time step of 1 ms. In each step, the total illuminated RF area was computed from the convolution of the center/surround RF components with the stimulus. RF activation at time step $t$ was changed by the difference between the sum of the newly illuminated RF area and the signal from the previous time step:

$$RF_t = (RF_{illumination,t} - RF_{t-1})/\tau + RF_{t-1} \qquad (4)$$

The full RF was computed according to the following equation:

$$RF_{Full} = RF_{center} - RFactor_{Surround} \times F_{Surround} \qquad (5)$$

Where Factor$_{Surround}$ indicated the intensity of the surround activation. In each simulation run, Factor$_{Surround}$ was set to the same value in all modeled cells. In center-only simulations, Factor$_{Surround}$ was zero.

Simulated neurons were distributed on a 1000 × 1000 μm square grid stimulated either by moving/stationary bars with similar parameters (speed, contrast, size) as in the experiments or by natural images.

## Natural movies
The natural movies were composed of background/mask chosen from individual frames of the "catcam" database[26,83] and stimuli depicting birds of prey (https://zenodo.org/record/46481#.YstBpnbML9Y). The images were cropped to 100 × 100 pixels and presented as an input to the simulated network. The intensity of the background/mask was scaled to be at the mean pixel level (i.e., 128 pixel luminance value) with an SD of 30. The mean intensity of the stimuli was set to be 2 SD higher than the background mean. In some simulations, the stimulus was not presented. Instead, the background translated horizontally at 0.5 mm/s as measured over the artificial retina. The shape of the mask was chosen by foreground objects in separate movie frames. The mask was absent for simulations of continuous motion. Response amplitudes were measured in a time window spanning 500 ms starting at the time of object appearance over the location of the simulated cell.

Mutual information was measured as the entropy of peak response of cells close (<100 μm) to the initial appearance of the stimulus near the mask/stimulus boundary in the presence/absence of the mask, minus the average entropy of the peak responses to the individual conditions. Mutual information (I) is given by:

$$I(VS;R) = H(VS) - H(VS|R) \qquad (6)$$

Where I(VS;R) is the mutual information between the visual stimulus type VS (presence/absence of mask) to the responses of the simulated RFs I. H(VS) denotes the overall entropy of the dataset and H(VS|R) is the conditional entropy, indicating the uncertainty about the visual stimulus after observing the response.

The entropy was defined as the sum over the probabilities (p) of occurrences of the peak response amplitudes (x), computed for the cells that met the inclusion criteria (number of cells = $n$ = 68):

$$H(X) = -\sum_{i=1}^{n} P(x_i)\log_2 P(x_i) \qquad (7)$$

The range of the peak response values was between 0 to about 200 (AU). To convert from continuous to discrete values of the peak amplitudes, we rounded the peak values to the nearest integer.

## Detailed retinal simulation
The simulated retina consisted of a one-dimensional array (length = 1500 μm) of photoreceptors, horizontal, bipolar, and amacrine cells, spaced 10 μm apart. Stimuli were provided by a bright bar that was either flashed for 2 s or moved over the retina (speed = 0.5 mm/s). Visual edges were created by masking visual presentation 300 μm from the borders of the array. The simulation time step was 1 ms.

Photoreceptor response to light was modeled as a difference between two activation functions (Ph$_A$, Ph$_B$) with instantaneous rise time and decay times of 60 and 400 ms[47,75], respectively.

$$Ph = Ph_A - 0.8Ph_B \qquad (8)$$

Time step computations for the activation functions were given by:

$$Ph_{A,t} = (RF_t - Ph_{A,(t-1)})/60 + Ph_{A,(t-1)} \qquad (9)$$

$$Ph_{B,t} = (RF_t - Ph_{B,(t-1)})/400 + Ph_{B,(t-1)} \qquad (10)$$

Where RF was computed from the value of the stimulus at the position of the photoreceptor and horizontal cell feedback (see below) and Ph$_{t-1}$ represents the value of the activation function on a previous time step.

Interactions between cells were guided by the spatial extent of their RFs, given by:

$$RF_x = e^{-x^2/\frac{FWHM^2}{11.09}} \tag{11}$$

Where x is the distance from RF center and FWHM is the full width at half maximum of the RF, all units are in μm.

Horizontal cells integrated all photoreceptor signals in their RF (FWHM = 300 μm[45]). The spatial RF signal in horizontal cell$_i$ (HC$_{\infty,i}$) was described according to the following:

$$HC_{\infty,i} = Gain_{Ph \to HC} \sum_{j=1}^{n} d_{i,j} Ph_j / \sum_{j=1}^{n} d_{i,j} \tag{12}$$

Where the photoreceptor−horizontal cell gain was set to 1; $n = 150$ is the number of photoreceptors reflecting the spatial length of the model, $d_{i,j}Ph_j$ represents signal from photoreceptor$_j$ on horizontal cell$_i$ and the last term used to correct responses by RF size, allowing us to compare the effect of changing horizontal cells' RF widths without affecting their gain.

The total activation of the horizontal cells at a time step $t$ was given by the following equation:

$$HC_{i,t} = \frac{E_{Ph}\left(\frac{HC_{\infty,i}}{HC_{\infty,i} + E_{Ph}}\right) - HC_{i,(t-1)}}{\tau_{HC}} + HC_{i,(t-1)} \tag{13}$$

In which $\tau_{HC}$ is the horizontal cell activation time constant = 300 ms[47,75] and E$_{ph}$ is the driving force of the photoreceptor signal = 10 mV.

Each photoreceptor combined horizontal cell signals (normalized by the same distance function) with visual illumination as follows:

$$Ph_{i,t} = Gain_{HC \to Ph}\left(\sum_{j=1}^{n} E_{HC} d_{i,j} HC_{j,t} + E_{VS} VS_{i,t}\right) / \left(\sum_{j=1}^{n} |d_{i,j} HC_{j,t}| + VS_{i,t} + 1\right) \tag{14}$$

Where the photoreceptor−horizontal cell gain was set to 0.5 unless specified otherwise, $VS_{i,t}$ represents the value of the visual stimulus over photoreceptor $i$ at time t, $E_{HC}$ and $E_{VS}$ are the driving forces of the inhibitory input from horizontal cells = 2 mV and the light-induced hyperpolarization = −10 mV, and HC$_{j,t}$ is the feedback from horizontal cell $j$.

Similar to horizontal cells, bipolar cells sampled photoreceptors' signals in their RF. The steady-state input-output transformation at the photoreceptor-BC synapse was given by the following relationship:

$$BC_{\infty,i} = -\sum_{j=1}^{n} d_{i,j}\left[\frac{1}{1 + e^{V_{slope}(V_{1/2} - Ph_j)}} - \frac{1}{1 + e^{V_{slope}V_{1/2}}}\right] \tag{15}$$

Where d$_{i,j}$ was the distance function computed from the spatial spread of the RF (FWHM = 50 μm), $V_{slope}$ and $V_{1/2}$ defined the slope and the 50% point of the Ph-BC transformation function, and the last term provided a subtraction of the baseline photoreceptor signal.

The temporal evolution of the photoreceptor input at time step $t$ was computed using the following:

$$BC_t = \left(BC_{\infty,t} - BC_{t-1}\right) / \tau_{BC} + BC_{t-1} \tag{16}$$

In which $\tau_{BC}$ indicate the activation time constant = 50 ms.

The synaptic input was converted to membrane potential (BC$_{Vm}$) by the following equation:

$$BC_{Vm} = E_{Ph} \frac{E_{Ph} BC + E_{AC} AC}{|BC| + |AC| + E_{Ph}} \tag{17}$$

Where E$_{Ph}$ = 10 mV, BC is the excitatory drive from the photoreceptors described above and E$_{AC}$ = 0 mV is the reversal potential of the amacrine drive (AC).

The following parameters were used to simulate the transient and sustained BCs: V$_{slope(transient)}$ = 1.1 mV$^{-1}$ and V$_{slope(sustained)}$ = 0.1 mV$^{-1}$ and V$_{1/2(transient)}$ = 4 mV and V$_{1/2(sustained)}$ = 1 mV.

The amacrine cell circuit was simulated as follows. First, we computed the stimulus-induced depolarization of a separate population of BCs. The parameters for these BCs used to create the data shown in Fig. 4 were V$_{slope}$ = 2 mV$^{-1}$ and V$_{1/2}$ = 4 mV. An array of amacrine cells sampled the signals from all BCs in their RF, as described above for photoreceptor-BC synapse. The signal transformation in the BC-AC synapse had the following values: Gain = 1, V$_{slope}$ = 1.5 mV$^{-1}$ and V$_{1/2}$ = 7 mV. The activation time constant of the AC was slightly longer (80 ms) than the corresponding value for BCs[15]. The AC drive to each BCs was formed by the combined output of all the amacrine cells, normalized by their RF distance. In simulations where the AC drive was absent, AC gain was set to zero.

### Reporting summary

Further information on research design is available in the Nature Research Reporting Summary linked to this article.

## Data availability

The full datatset, including the code used for analysis, is available on https://github.com/PolegPolskyLab/BipolarClustering [84] and on https://doi.org/10.5281/zenodo.6814536 [85] Source data are provided with this paper.

## Code availability

The code for the visual stimulation and simulations is available on https://github.com/PolegPolskyLab/BipolarClustering [84].

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

## Acknowledgements

We thank Shai Sabbah, Dan Denman, and Jeffrey Diamond for providing helpful comments on this paper. This work was supported by NIH grant (R01 EY030841-02) to A.P.P.

## Author contributions

J.G.: Investigation, writing-review and editing, supervision. S.B.: Investigation, formal analysis, software, writing-review and editing. M.G.: Investigation, writing-review and editing. J.H.: Formal analysis, writing-review and editing. A.P.P.: Conceptualization, methodology, software,

formal analysis, resources, data curation, writing-original draft, supervision, project administration, and funding acquisition.

## Competing interests

The authors declare no competing interests.
