## [Peer Review File · Nature Communications]

Classical Center-Surround Receptive Fields Facilitate Novel Object Detection in Retinal Bipolar CellsREVIEWER COMMENTS

Reviewer #1 (Remarks to the Author):

The article by Gaynes et al. entitled "Novel object detection and multiplexed motion representation in retinal bipolar cells" provided the novel finding of the motion-emergence sensitivity in retinal bipolar cells. The authors performed two-photon glutamate imaging from the inner layers of the mice retina. The analysis of response kinetics to the static flash, continuous motion, and motion emergence revealed that bipolar cells showed that emphasized response to the emerging motion. The motion sensitivity was generated by the center-surround RF antagonism that is mediated by the horizontal cell's feedback regulation of photoreceptor outputs. The model simulation supported the contribution of the center-surround antagonism. Further, the authors performed glutamate imaging from starburst cell processes and revealed that the motion sensitivity in bipolar cells actually had an impact on the postsynaptic cells.

The body of studies of retinal physiology and human visual psychophysics has shown that the center-surround antagonism in the retinal neurons mediates the enhancement and decorrelation of retinal outputs and that such modulated retinal signal would contribute to visual perception as shown by the studies of visual illusion at the luminance or contrast edges. Intriguingly, the findings by Gaynes et al. are providing new functional role of the center-surround antagonism in the retinal motion processing. The authors performed well-considered experiments and analyses, however, some results have not yet fully investigated.

Major comments:

- 1) There are some concerns regarding the relationship between identified BC groups by clustering and IPL depth.
 - 1a) The depth of IPL is different from area to area in the retina; central retina has thicker IPL, whereas peripheral retina has thinner IPL. The authors should clarify how they normalized IPL depth difference across different experiments and how they defined depth in relation to INL and GCL. As the authors might be aware, researchers often use ChAT band depth as a ruler for IPL depth.
 - 1b) The authors should discuss how much the identified 12 groups (Figure 1d) have similar response dynamics compared to previously identified BC functional groups (e.g. Franke et al., Nature 2017; Baden et al., Curr Biol 2013).
 - 1c) How glutamatergic amacrine cells can be excluded from the identified BC groups?
 - 1d) Line 242-244, "We demonstrated that individual BC types convey different temporal features of the stimulus contingent on the presence of motion"). The authors should not use the word "BC types" unless they show more convincing data demonstrating that the identified 12 BC groups functionally correspond to known bipolar cell types.
- 2) Figure 2, even though the authors show in Figure 1 that 12 BC groups have similar kinetics in motion responses, it would be still worth showing how the 12 BC groups responded to emerging vs exit motion (like Figure 2c), and clarify if there are any group-related differences or not. Otherwise, the significance of clustering analyses performed in Figure 1 looks very negligible for the main point of this paper.
- 3) The authors concluded that the center-surround antagonism establishes motion sensitivity. Since Figure 2c includes the data where RF positions vary within 50 μm from the mask-stimulus boundary, it is not clear the detail of the relationship between center-surround components of RF and the location of motion appearance. For example, the author can change the location of motion emergence among surround RF, center RF, and out of RF, and analyze the differences in responses. This data will provide more direct evidence for the motion sensitivity.
- 4) The authors concluded that horizontal cells, not amacrine cells, mediate the motion sensitivity in bipolar cells by the results of pharmacology using GABA and glycine receptors (Figure 3). However, the contribution of horizontal cells has not yet been fully

addressed experimentally. The authors should block the horizontal cell activity and examine the effects of the block on bipolar cell motion sensitivity. For example, the authors can manipulate horizontal cell activity by application of carbenoxolone to block ephaptic feedback (Kamermans et al., *Science* 2001; Kemmler et al., *J Neurosci* 2014).

5) The authors used GABA receptor blockers to block amacrine cells, however, previous studies showed that GABAergic transmission in the outer retina (cone, horizontal cell, bipolar cell) modulates the feedback from horizontal cells to cone outputs (Kemmler et al., *J Neurosci* 2014; Liu et al., *J Physiol* 2019). The authors should explain why the potential effects of GABA receptor blockades on the horizontal cell feedback were not considered as a mechanism for altered motion sensitivity in bipolar cells.

6) Previous studies have shown that the RF surround in bipolar cell outputs is mediated by GABAergic amacrine cells, and blockade of GABA receptors actually reduced RF surround components (Franke et al., *Nature* 2017; Matsumoto et al., *Curr Biol* 2019). If inhibition in RF surround of bipolar cells contributes to the motion sensitivity as the authors discussed in the Discussion section, it is not clear why the impairments of RF surround by GABA blockers do not affect the motion sensitivity. The authors should explain the discrepancy with the previous results.

7) The authors performed a model simulation to reveal that the horizontal cell feedback is the key to establish the motion sensitivity in bipolar cells (Figure 4). In this model, the timing of activation of each circuit component, in particular in horizontal cell, is critical to generate the motion sensitivity. However, it is not clear if the activity timing of horizontal cells relative to that of photoreceptors in the model can properly reflect the physiological properties. The authors should show that the activity timing in horizontal cells relative to photoreceptors is physiologically reasonable from the current literature, such as anatomical evidence or physiological experiments. This is important to show if this modeling is not artificial.

8) The authors concluded that the center-surround antagonism works at the luminance edge (i.e. near the mask) for motion sensitivity. Indeed, the simulation showed that cells near the mask had higher amplitudes (Figure 6). However, the object itself has luminance edges (i.e. contour of the object to segregate it from the background), and thus it would be possible that the edge of the moving object itself can induce the motion sensitivity regardless of the location of cells since cells can experience the edge of moving object. The authors should explain why the edge of the mask, not the edge of a moving object, induces motion sensitivity.

9) For the simulation (Figure 6), the intensity of the RF surround is not clear. In the method, the authors say "Where FactorSurround indicated the intensity of the surround activation and varied between 0 to 0.5". However, it is not clear if the intensity is varied among individual cells up to 0.5 or all cells have 0.5 in the C+S condition. Since the intensity of surround can be significant in the model, the authors should clarify that. In addition, if the intensity of surround can affect the results, the authors should model how much intensity can generate the motion sensitivity at the edge, and discuss if the surround intensity is in the physiological range of actual bipolar cell surround.

10) It is not clear how the authors identify the orientation of the dendrites of starburst amacrine cells from the labeled processes and grouped them into the edge near soma and edge near tips. Since the authors used Chat-Cre mice, there should be dense overlaps of dendrites from different starburst amacrine cells, which would make it difficult to determine whether the labeled dendrites are near the tips or the soma. The authors should clarify and explain.

11) In the Discussion, the authors discussed the potential role of motion sensitivity in bipolar cells in signaling the appearance of new objects to the visual centers. Although it is an interesting possibility, however, this manuscript has not yet shown that the motion sensitive outputs from bipolar cells actually influence the activity of postsynaptic

ganglion cells and that the ganglion cells relay such motion-emergence-sensitive signaling to the visual centers. For example, it would be still possible that the bipolar cell outputs that are sensitive to motion emergence are rectified at the dendrites of ganglion cells, and thus not transmitted to visual centers. The authors should tone down the discussion or carefully explain the potential contribution to ganglion cell activity.

12) The authors discussed the contribution to human perception. In addition to the visual saliency as the authors discussed, there has been a long history about the neuronal lateral inhibition on the center-surround antagonism and the visual illusion, like Mach bands or illusory contour (Pessoa, Vis Res 1996). The authors should discuss the motion sensitivity in the context of the illusory enhancement of perception around the luminance edges.

Minor comments:

- 1) Figure 1g, it is not clear what the black circles on some groups indicate.
- 2) Figure 6e, it is not clear what the "DSIglutamate (%)" indicates (Figure 6e) indicates. From the method, DSI is ranged from 0 to 1. Although there are some texts around lines 215-216, this should be clarified in the method or figure legend with equations if needed.
- 3) Figure 5c, the heat map of response amplitudes shows that there is another hotspot with higher amplitudes around the top right (not around the edge). The authors should explain why the higher amplitudes appear nowhere near the edge.
- 4) The description of mutual information is not clear. The authors should provide an equation and clarify what the entropy was calculated from.
- 5) The description in figure legend in Figure 4 (and the corresponding Method description) is not enough to understand the detail.
 - i) What type of motion stimulus (emerging, exiting, or full-field) is used for the space-time plot in Figure 4b left and Figure 4c left?
 - ii) Please clarify the reference (i.g. anatomical or physiological) for the number of photoreceptor inputs for horizontal cell ("n=150").
 - iii) The photoreceptors and bipolar cell synapses are not clear. What is the number of photoreceptor inputs for a bipolar cell is not shown?
 - iv) How do bipolar cells integrate photoreceptors within RF (a 50x50 um square)?
 - v) What is the "dimensionality-corrected RF"?

Reviewer #2 (Remarks to the Author):

The manuscript by Gaynes and colleagues explores the responses of bipolar cells (BCs) in the mouse retina to the appearance of static objects and various kinds of motion. The authors measure glutamate release throughout the inner plexiform layer (IPL) with a ubiquitously expressed virus (Figs. 1-3) and specifically onto the neurites of starburst amacrine cells (SACs) with a SAC-specific mouse line (Fig. 6).

The main finding of the study is that objects that suddenly appear in the receptive field (RF) center of BCs (either by flashing there or emerging from an occluder) drive larger and faster responses than smoothly moving stimuli that active the BC RF surrounds before the RF center.

While the data and analyses are generally of high quality, the impact of the manuscript is substantially diminished by its presentation. In many places, the authors over-interpret their results, and they frequently fail to place their data in the context of historical results and models of center-surround RFs.

The core result here is a trivial consequence of a linear center-surround RF, as the authors point out in their linear model in Fig. 4. A stimulus that hits the inhibitory RF

surround and then the excitatory RF center will give a weaker response than one that appears suddenly in the RF center. The magnitude of this effect will depend on both the strength of the RF surround and, for moving objects, the center-surround delay and their relative kinetics. This is a form of a Reichardt detector (Von Hassenstein and Reichardt, 1956) as proposed as a mechanism for direction selectivity by Barlow in 1965 (Barlow and Levick, 1965) and investigated in great detail in the fly visual system (Borst and Euler, 2011; Borst et al., 2020). Language throughout the manuscript frames this as an unexpected discovery or over-interprets the result. Some examples from the abstract:

"Center-surround RFs are thought to enhance responses to spatial contrasts (i.e., edges), but how they contribute to motion processing is unknown"

There are many, many papers on how center-surround RFs contribute to motion processing. Lots of them are cited in (Borst and Euler, 2011).

"... their responses to continuous motion are smaller, slower, and cannot be predicted by signals elicited by stationary stimuli."

Actually, a linear model does exactly that! The spatiotemporal RF inferred by static presentations of bars will predict a smaller, slower response to smooth motion across the RF surround and then center than a flashed bar in the RF center.

"The alteration in signal dynamics induced by novel objects dwarfs the enhancement of spatial edges and can be explained by priming of RF surround during continuous motion."

This is not a quantitative comparison in the paper – seems like comparing apples to oranges in this statement – and "priming" is an unnecessarily confusing term (that implies facilitation rather than suppression) for a stimulus that simply hits the RF surround before the center, as described above.

"...demonstrate an unappreciated capacity of the center-surround architecture to facilitate novel object detection..."

This has been appreciated for over 50 years.

"...and multiplexed encoding of distinct sensory modalities."

Extremely vague. There is only one sensory modality being investigated here. Is multiplexed supposed to mean that BCs respond both to flashed bars and moving objects? That's not a novel finding for BCs or for any visual neuron.

Specific issues:

1. The BC clustering in Fig. 1 is suspect. Most of the clusters within the ON and OFF subclasses have properties within 1 s.d. of each other. The McClain-Rao index (Fig. 1c) is the ratio of the mean within-cluster distances to the mean between-cluster distance. Thus, optimal clustering is defined by the MINIMUM of this quantity, not the maximum! And why does the graph stop at 3 clusters instead of 2? The point chosen for OFF clustering has a McClain-Rao index of ~ 0.95 meaning that between between-cluster distances are only 5% larger than within-cluster distances; this indicates pretty terrible clustering, as suggested by the fact that the stratification profiles for the 5 OFF clusters are almost identical. With the exception of C6, the ON clusters are also much more similar than I would expect given previous work (Franke et al., 2017), and the authors make no effort to link each cluster to a known BC type by the published stratification profiles.

2. Fig. 3 reports that the "novelty detection" mechanism (BC surrounds) does not depend on inhibition from amacrine cells (ACs) because it remains intact in a complete GABA and glycine receptor block. This is the most surprising result in the manuscript. There has been significant controversy over the years about the relative contributions of inner vs. outer retinal mechanisms to BC surrounds, and a study using the same technique (iGluSnFR to measure mouse BC outputs) reported that the same pharmacological manipulation had dramatic effects on the temporal responses of BCs that were most apparent for full-field light stimuli, suggesting a role for the RF surround

(Franke et al., 2017). Further dissection of the circuit mechanism is possible here by using each of the receptor blockers (for glycine, GABA_A, and GABA_C) alone and in pairwise combinations. Additionally, if the authors are indeed claiming that BC surrounds are dominated by an outer retinal mechanism, they should be eliminated in by pH buffering with HEPES (Davenport et al., 2008) or using a Cx57 KO animal (Shelley et al., 2005).

3. The authors cite two other important papers that have looked at motion processing in BCs in the mouse (Kuo et al., 2016) and primate (Manookin et al., 2018) retina, but they don't really address the fact that these studies used a different definition of motion sensitivity that has also been adopted in studies of downstream areas, like superior colliculus (Gale and Murphy, 2014). Both these studies showed that BCs responses to smooth motion more strongly than to a randomized sequence of bar presentations, so this is OPPOSITE to the effect reported here. The mechanism for this form of motion sensitivity required electrical coupling between BCs (via AII ACs) and the BC output nonlinearity. An important and unresolved question in this field is whether OFF BCs (which are not coupled via AII ACs) can have stronger responses to motion order than random order bar presentations.

References

- Barlow, H.B., and Levick, W.R. (1965). The mechanism of directionally selective units in rabbit's retina. *J. Physiol.* 178, 477–504.
- Borst, A., and Euler, T. (2011). Seeing Things in Motion: Models, Circuits, and Mechanisms. *Neuron*.
- Borst, A., Haag, J., and Mauss, A.S. (2020). How fly neurons compute the direction of visual motion. *J. Comp. Physiol. A Neuroethol. Sensory, Neural, Behav. Physiol.* 206, 109–124.
- Davenport, C.M., Detwiler, P.B., and Dacey, D.M. (2008). Effects of pH Buffering on Horizontal and Ganglion Cell Light Responses in Primate Retina: Evidence for the Proton Hypothesis of Surround Formation. *J. Neurosci.* 28, 456–464.
- Franke, K., Berens, P., Schubert, T., Bethge, M., Euler, T., and Baden, T. (2017). Inhibition decorrelates visual feature representations in the inner retina. *Nature* 542, 439–444.
- Gale, S.D., and Murphy, G.J. (2014). Distinct representation and distribution of visual information by specific cell types in mouse superficial superior colliculus. *J Neurosci* 34, 13458–13471.
- Von Hassenstein, B., and Reichardt, W. (1956). Systemtheoretische Analyse der Zeit-, Reihenfolgen- und Vorzeichenauswertung bei der Bewegungsperzeption des Rüsselkäfers *Chlorophanus*. *Zeitschrift Fur Naturforsch. - Sect. B J. Chem. Sci.* 11, 513–524.
- Kuo, S.P., Schwartz, G.W., and Rieke, F. (2016). Nonlinear Spatiotemporal Integration by Electrical and Chemical Synapses in the Retina. *Neuron* 90, 320–332.
- Manookin, M.B., Patterson, S.S., and Linehan, C.M. (2018). Neural Mechanisms Mediating Motion Sensitivity in Parasol Ganglion Cells of the Primate Retina. *Neuron* 97, 1327–1340.e4.
- Shelley, J.A., Dedek, K., and Weiler, R. (2005). Effects of Connexin57 Deletion on Horizontal Cell Receptive Field Size in the Mouse Retina. *Invest. Ophthalmol. Vis. Sci.* 46, 600–600.

Reviewer #3 (Remarks to the Author):

Review of NCOMMS-21-16455, 'Novel Object Detection and Multiplexed Motion Representation in Retinal Bipolar Cells'

This study focuses on the visual encoding properties of retinal bipolar cells. Response properties of these cells are an re-emerging field of study and this work is timely. The authors use two-photon fluorescence glutamate imaging to measure stimulus-evoked

responses of functionally identified bipolar cell types and report that responses to visual stimuli that first appear in the receptive field center are signaled comparatively stronger than those that originate outside it. This phenomenon is captured by a computational model with a dependence on horizontal cell signaling. Model analysis shows that it improves novel object detection. While this response property renders bipolar cell output particularly sensitive to visual object motion emerging within the receptive field, it does not establish direction selectivity in the bipolar cells themselves, nor does it appear to enhance direction tuning at subsequent signaling stages, but see point 4 below. The main impact on visual encoding appears to be increasing salience for novel, local movement.

I have the following comments and concerns.

1. Abstract, line 15 states that the alteration in signal dynamics induced by novel objects 'dwarfs' the enhancement of spatial edges. This is a subjective statement, and also appears to be an overstatement. It is not made clear exactly which is considered the measurement of enhancement of spatial edges. If I look at Figure 2c, then I see that the response to novel motion is about 25% larger than that to a full field edge stimulus. Normalizing the responses to the full field edge stimulus necessarily puts that at 1.0 (bottom of the graph), but novel moving stimulus response is only at 1.25 df/f.

2. Title and main text refer to 'multiplexed encoding' in bipolar cells based on the presented results. While the results convincingly show that specifics of the stimulus configuration including motion and position of an occluder changes the response rise time and peak amplitude, the term multiplexed implies that these specifics can also be separately decoded. It is not at all clear to me how that would work: the bipolar cell is a single encoding channel, and the mode of encoding may not be constant – as following a change in mean luminance, or contrast. The response properties help increase salience, but they do not appear to provide independent (orthogonal) modes of visual encoding, which is the image that 'multiplexed' conjures up for me. This needs to be addressed.

3. Fig. 6, title states that [BC sensitivity to novel objects] 'influences the analysis of motion processing...'. It should be made unambiguous here and in Results what is the nature of this influence.

4. The results of glutamate imaging on SAC dendrites are murky. This is due in part to experimental limitations (imaged SAC branch orientation not controlled; stimulus edge near soma of some, distal for other BCs), and the write-up of these results. Specifically, the authors report a DSI of $32 \pm 21\%$ for bipolar cell glutamate signal impinging on SAC dendrites. But the mask-stimulus boundary is said not to contribute to DS in SACs (line 218-220). The explanation (lines 220-228) is very difficult to follow and needs to be elaborated/illustrated to be accessible for readers. It should be made crystal clear whether BC motion enhanced responses increase direction tuning as measured by DSI, or if it merely boosts the (non-DS) transmitted signal to the SAC.

5. Related to the previous point, conceptually, it is not clear how the enhanced response to novel motion can lead to an increased DSI in the SAC dendrite. The bipolar cell response is increased for motion away from the center – but that can be in any direction. Any bipolar cell connected to a SAC dendrite of particular orientation therefore will provide increased glutamatergic drive to this SAC dendrite depending on the position of the edge/occluder, independent of whether or not this is the preferred direction for that SAC dendrite.

6. Figure 1e, left panel title. I understand the need to abbreviate 'Stationary flashes' (as used in d). Instead of 'Stationary' consider using 'Flashes' instead, because it better represents what was presented.

7. Figure 2 shows responses within ROIs classified as transient and sustained. Were ON and OFF responding ROIs (presumed ON and OFF bipolar cell types) combined in this analysis? If this is stated somewhere, I could not find it; consider making this explicit in

Results section and in Fig2 legend or figure panels.

8. I find Figure 4 panel b uninterpretable. The bottom part is a space time plot, with responses from two cell types overlapping, and six different stimulus configurations indicated with arrows at the top. To make this accessible, consider splitting this out in a few panels. Similarly, pane, c, how does the continuous wave form relate to the six stimulus configurations above. This is too convolved to parse easily.

9. Many parameters went into the construction of the model presented in Fig. 4. Legend refers to one of them - BC sensitivity to photoceptive (sic) release – it would be useful to give more information about the model including the parameters and sensitivity of the main results to the parameter values used in the simulations.

10. Line 72 reports the lack of correlation between flash- and motion-driven rise time. Text here says Pearson cc of -0.04; text in Figure 1F says -0.03. Which is it?

On behalf of my co-authors and myself, I am pleased to resubmit a revised version of our manuscript to be considered for publication in Nature Communications.

We greatly appreciate the positive and critical comments that had inspired us to rework and improve the paper.

We have decided to change the title of the manuscript from “Novel object detection and multiplexed motion representation in retinal bipolar cells”. The change was prompted by reviewers noticing that the term ‘multiplicative’ is ambiguous and may not accurately describe our results. Following new pharmacological experiments suggested by the reviewers, we are now much more confident that the computation of emerging object detection is facilitated by horizontal cells. We also wanted to emphasize the general implications of our findings, i.e., that motion computations rely on simple interactions in circular-symmetric receptive fields (RFs). The revised title reads “Classical Center-Surround Receptive Fields Facilitate Novel Object Detection already in the First Retinal Synapse”.

We have performed new experiments and simulations prompted by the suggestions made by the reviewers. We have corrected our clustering approach; the new dataset was best clustered into 6 OFF and 8 ON glutamate clusters. We have directly addressed the question of the role of horizontal cell feedback on motion processing in glutamate-releasing cells in the IPL. Guided by the reviewers' suggestions, we have decided to use a combination of CNQX and HEPES, which allow for very effective inhibition of photoreceptor input to horizontal cells and their ephaptic feedback, respectively. In our hands, HEPES alone did not block edge response (Supplementary Figure 6). Unlike carbenoxolone, CNQX is a highly specific and effective blocker of AMPA and Kainite receptors found on horizontal cells. While CNQX also blocks the transmission of photoreceptor signals to OFF-BCs, ON-BCs are spared, thus allowing us to compare their activity in control conditions and after the horizontal inhibition is eliminated. We have updated Figure 3 to include the effect of these blockers as shown below:

New Fig. 3. Pharmacological blockage of inhibition from horizontal cells, but not amacrine cells, eliminates emerging object sensitivity and static edge enhancement. **a** Illustration of the pharmacological manipulations designed to probe the contribution of the two circuit elements mediating the surround in BCs. CNQX (50 μ M) and HEPES (10 mM) block photoreceptor input to Off-BCs and horizontal cells and part of the horizontal cells' feedback, required for the inhibitory surround. SR95531 (Gabazine, 50 μ M), TPMPA (100 μ M) and Strychnine (1 μ M) primarily negate amacrine inhibition onto BC axons. **b** Representative glutamate responses before (black) and after (blue) blockage of horizontal (top) or amacrine (bottom) inhibition. **c-d** Peak glutamate fluorescence (**c**), as well as rise and decay times (**d**) normalized by full-field motion. Control responses for both cocktails were combined. # $p < 0.05$, ### $p < 0.001$ CNQX+HEPES vs. other conditions. **e** Static edge enhancement (top) and novel motion detection (bottom) calculated

*from the peak of the responses in control condition, with amacrine and horizontal cell blockers. #p<0.05, ###p<0.001 change from 0%. ***p<0.001 difference between pharmacological manipulations. All statistical tests were ANOVA followed by Tukey test with Bonferroni's correction. Error bars-SD.*

A second major addition to the manuscript has been a detailed simulation of the role of amacrine cells in motion processing. We now propose an explanation for the lack of effect seen in pharmacological experiments designed to block amacrine cell inhibition onto bipolar cells. Our findings indicate that for a wide range of modeled conditions, amacrine cell drive is weak and temporally offset during established motion. These simulations agree with the notion that the impact of amacrine cells on bipolar cell signaling may be significantly different for flashed and moving objects.

new Fig. 4. Simulated visual processing in outer and inner plexiform layers capture empirically observed differences in the effects of horizontal and amacrine cell inhibition on novel object enhancement and population dynamics. **a** The circuit modeled in **b-d**. Photoreceptors (black) combined the visual input with feedback inhibition from horizontal cells (blue). Transient (green) and sustained (red) BCs differed by the formulation of their dependency on the photoreceptor input. **b** Top, photoreceptor (black) and HC (blue) voltage responses (inverted for presentation

purposes) elicited by the visual stimuli. Note the effect of preceding inhibition (blue arrows) that is present during established motion. Bottom, the shapes of the BC potentials for the same stimuli. c The peak amplitude of the simulated visual responses, normalized by the response to the full-field moving bar. In the absence of horizontal feedback, photoreceptor responses were similar across all probed stimuli (open circles). d Enhancement of the emerging object (solid) and the static edge (dotted) as a function of different model parameters. e Top, ACs were included in the simulated circuit. Amacrine cells were stimulated by a separate subpopulation of BCs (grey). Bottom, BC and AC (blue) activation by narrow (diameter = 100 μm , top) and wide (diameter = 500 μm , bottom) spots, used to calibrate the strength of AC surround. Dotted, responses in the absence of inhibition in the inner plexiform layer. f-g as in b, d in the presence of ACs.

In our revised submission, we indicated changes in response to reviewers' comments in blue. Below you will find point-by-point responses to the original reviews with references to page and line numbers in the revised manuscript.

Thank you very much for your time and consideration.

Sincerely,

Alon Poleg-Polsky

Reviewer #1 (Remarks to the Author):

The article by Gaynes et al. entitled "Novel object detection and multiplexed motion representation in retinal bipolar cells" provided the novel finding of the motion-emergence sensitivity in retinal bipolar cells. The authors performed two-photon glutamate imaging from the inner layers of the mice retina. The analysis of response kinetics to the static flash, continuous motion, and motion emergence revealed that bipolar cells showed that emphasized response to the emerging motion. The motion sensitivity was generated by the center-surround RF antagonism that is mediated by the horizontal cell's feedback regulation of photoreceptor outputs. The model simulation supported the contribution of the center-surround antagonism. Further, the authors performed glutamate imaging from starburst cell processes and revealed that the motion sensitivity in bipolar cells actually had an impact on the postsynaptic cells.

The body of studies of retinal physiology and human visual psychophysics has shown that the center-surround antagonism in the retinal neurons mediates the enhancement and decorrelation of retinal outputs and that such modulated retinal signal would contribute to

visual perception as shown by the studies of visual illusion at the luminance or contrast edges. Intriguingly, the findings by Gaynes et al. are providing new functional role of the center-surround antagonism in the retinal motion processing. The authors performed well-considered experiments and analyses, however, some results have not yet fully investigated.

Major comments:

1) There are some concerns regarding the relationship between identified BC groups by clustering and IPL depth.

1a) The depth of IPL is different from area to area in the retina; central retina has thicker IPL, whereas peripheral retina has thinner IPL. The authors should clarify how they normalized IPL depth difference across different experiments and how they defined depth in relation to INL and GCL. As the authors might be aware, researchers often use ChAT band depth as a ruler for IPL depth.

We thank the reviewer for pointing out the information we inadvertently left out in the methods section. We have added the following paragraph to explain better how we estimated the position of the recorded ROIs in the IPL:

IPL depth was measured from the transmitted light channel extracted from the z-stack taken of the entire width of the retina that accompanied all functional recordings. The curvature of the retina was corrected by measuring the height of the inner limiting membrane at the four corners and the center of the image stack and fitting a curved plane that crossed these 5 points. The total width of the IPL was measured at the center of the z-stack from the transmitted light channel for each animal and recording region. A similar approach was used to measure the curvature of the retina in ChAT-Cre/tdTomato mice. In these cases, the fluorescence on the red channel was used to mark the location of the ChAT bands. Both approaches provided similar anatomical estimates and were used interchangeably to measure the depth of the recorded ROI in the IPL.

1b) The authors should discuss how much the identified 12 groups (Figure 1d) have similar response dynamics compared to previously identified BC functional groups (e.g. Franke et al., Nature 2017; Baden et al., Curr Biol 2013).

To address this question we have decided to increase our sample size of BCs that innervate starburst amacrine cells by imaging floxed iGluSnFR expressed in ChAT-Cre line. We found 3 OFF 3 ON glutamate clusters that are enriched in these experiments. We now use this information in conjunction with previous literature to propose functional – anatomical correspondence in our dataset.

1c) How glutamatergic amacrine cells can be excluded from the identified BC groups?

The reviewer raises a difficult question that we grappled with extensively. Based on Matsumoto et al., *Curr Biol* 2019, we would expect that glutamatergic amacrine cells (their G6 cluster) should have slow dynamics (specifically, a wide delay till activation and a prolonged decay). Based on this information, our clusters 12 - 14 may represent the release from these cells. We have attempted to measure glutamate signals in glutamatergic amacrine cells directly using the vGluT3-Cre line, which labels these cells (Chen, Lee et al. 2017). However, the resulting signals do not have the expected release characteristics. As can be seen from the figure below, the recordings have relatively fast temporal dynamics, leading us to conclude that the iGluSnFR fluorescence reflects the BC input to the amacrine cells and not their glutamate release.

1d) Line 242-244, "We demonstrated that individual BC types convey different temporal features of the stimulus contingent on the presence of motion"). The authors should not use the word "BC types" unless they show more convincing data demonstrating that the identified 12 BC groups functionally correspond to known bipolar cell types.

We appreciate this point. Indeed, we cannot be sure if our functional classification reflects release properties of distinct BC types. Accordingly, we have modified the text to replace 'BC types' with a more accurate description such as 'subpopulations of glutamate releasing cells' or 'recorded ROIs'. Please also see our response to Reviewer #2, specific issue 1.

2) Figure 2, even though the authors show in Figure 1 that 12 BC groups have similar kinetics in motion responses, it would be still worth showing how the 12 BC groups responded to emerging vs exit motion (like Figure 2c), and clarify if there are any group-related differences or

not. Otherwise, the significance of clustering analyses performed in Figure 1 looks very negligible for the main point of this paper.

We thank the reviewer for this suggestion. We have changed Figure 2 to include edge enhancement properties of the different functional clusters. We now show that only 2 glutamate clusters have a statistically significant stationary edge enhancement in our experimental conditions, while 12 out of 14 clusters show significant enhancement of emerging motion.

3) The authors concluded that the center-surround antagonism establishes motion sensitivity. Since Figure 2c includes the data where RF positions vary within 50 μm from the mask-stimulus boundary, it is not clear the detail of the relationship between center-surround components of RF and the location of motion appearance. For example, the author can change the location of motion emergence among surround RF, center RF, and out of RF, and analyze the differences in responses. This data will provide more direct evidence for the motion sensitivity.

We agree. We now show this analysis in supplementary figure 4b, c. The results indicate that the center RF position should be up to $\sim 70 \mu\text{m}$ from the mask-stimulus boundary to have a noticeably larger peak amplitude during emerging motion compared to responses to full-field moving bars. This is comparable to the previously estimated extent of BC center ($\sim 50 \mu\text{m}$). Thus, we conclude that RF center needs to be close to the mask-stimulus boundary but not behind the occluding mask. We have not directly measured the spatial span of the surround, which was estimated to be much wider ($\sim 300\text{-}400 \mu\text{m}$, Franke et al., 2017). Correspondingly, our results suggest that the exact location of the surround is less critical in determining the edge effects.

4) The authors concluded that horizontal cells, not amacrine cells, mediate the motion sensitivity in bipolar cells by the results of pharmacology using GABA and glycine receptors (Figure 3). However, the contribution of horizontal cells has not yet been fully addressed experimentally. The authors should block the horizontal cell activity and examine the effects of the block on bipolar cell motion sensitivity. For example, the authors can manipulate horizontal cell activity by application of carbenoxolone to block ephaptic feedback (Kamermans et al., Science 2001; Kemmler et al., J Neurosci 2014).

The reviewer raises an excellent point that we now address directly (see introduction). Our pharmacological results indeed show that horizontal mediated inhibition underpins edge effects found in our recordings.

5) The authors used GABA receptor blockers to block amacrine cells, however, previous studies showed that GABAergic transmission in the outer retina (cone, horizontal cell, bipolar cell) modulates the feedback from horizontal cells to cone outputs (Kemmler et al., J Neurosci 2014; Liu et al., J Physiol 2019). The authors should explain why the potential effects of GABA receptor blockades on the horizontal cell feedback were not considered as a mechanism for altered motion sensitivity in bipolar cells.

6) Previous studies have shown that the RF surround in bipolar cell outputs is mediated by GABAergic amacrine cells, and blockade of GABA receptors actually reduced RF surround components (Franke et al., Nature 2017; Matsumoto et al., Curr Biol 2019). If inhibition in RF surround of bipolar cells contributes to the motion sensitivity as the authors discussed in the Discussion section, it is not clear why the impairments of RF surround by GABA blockers do not affect the motion sensitivity. The authors should explain the discrepancy with the previous results.

Both points 5 and 6 address the same finding: horizontal cell, but not amacrine cell inhibition is vital to establish the part of the surround responsible for the novel object detection. It is indeed a surprising outcome, given the role of the amacrine cell surround in changing RF responses for flashed stimuli. We now note in the manuscript that gabazine could affect horizontal cell feedback. Based on the results of the pharmacological treatments, we speculate that the effect of gabazine on horizontal cells was only partly due to the role of other mechanisms that do not require GABA_A receptors.

How can amacrine cells diversify the repertoire of flashed full responses (Franke et al., 2017) but have little effect on motion responses? We decided to address this question by adding amacrine cells to the simulated retinal circuit. As we discuss in the text, the great diversity of amacrine cell types and the largely unknown circuits they are embedded in preclude us from creating a model that could encompass all possible interactions between amacrine cells and BCs. Instead, we focused on simulating a homogenous population of amacrine cells and exploring how their presence impacts motion computations in BCs. Our first finding was that it is indeed possible for amacrine cells to have a pronounced effect on responses to static flashes and at the same time keep the motion responses relatively intact. Two factors account for this outcome: first, as amacrine cells are stimulated by the bipolar drive, they are likely to inherit the motion preference of their presynaptic partners. If these partners are of the transient variety, the amplitude of amacrine responses to established motion (i.e., full-field and object exit conditions) would be significantly diminished vs. responses to static flashes. Second, the kinetics of the amacrine cell response during motion can become quite complicated. For example, we had revealed the importance of preceding inhibition from the horizontal cells to

the formation of motion responses in photoreceptors and BCs. Our current simulations suggest that the same process occurs in amacrine cells (figure 4). However, unlike the robustness of this observation in horizontal cells over a wide parameter range, the timing and the amplitude of the preceding inhibition in amacrine cells is influenced by numerous factors (such as the speed and amplitude of the BC drive, the nonlinear signal transformation in amacrine cells and their spatiotemporal RF parameters). Consequently, in some simulations, the amacrine drive during motion was strong, but it impacted the postsynaptic BCs too early or too late in the course of the light response, thus diminishing its effectiveness.

We also now show and discuss the potential enhancing or suppressing effects of amacrine cells on BC motion responses (Supplementary figure 9). We find it quite likely that a detailed experimental investigation would reveal similar findings, perhaps depending on the BC population or a more precise method to block individual amacrine cell subtypes. We have not attempted to pursue this line of inquiry further, as the required number of experiments would supersede the data we have performed for this manuscript (see below). Further, with our current direct demonstration of the role of horizontal feedback in enhancing responses to novel objects, we do not think these experiments would significantly change the conclusion that motion computations begin at the photoreceptor-horizontal cell synapse.

7) The authors performed a model simulation to reveal that the horizontal cell feedback is the key to establish the motion sensitivity in bipolar cells (Figure 4). In this model, the timing of activation of each circuit component, in particular in horizontal cell, is critical to generate the motion sensitivity. However, it is not clear if the activity timing of horizontal cells relative to that of photoreceptors in the model can properly reflect the physiological properties. The authors should show that the activity timing in horizontal cells relative to photoreceptors is physiologically reasonable from the current literature, such as anatomical evidence or physiological experiments. This is important to show if this modeling is not artificial.

We now provide citations in the body of the main text and in the methods section that validate the parameters we used for the simulation. Following this comment, we have also directly explored the parameter space (Figure 4d). As we show, enhanced responses to novel object emergence could be seen across diverse horizontal cell kinetics and spatial RF properties. Together with the results from linear models of center-surround interactions, we conclude that motion sensitivity should emerge in cells with pronounced and wide surround, which is indeed the case in photoreceptors.

8) The authors concluded that the center-surround antagonism works at the luminance edge (i.e. near the mask) for motion sensitivity. Indeed, the simulation showed that cells near the

mask had higher amplitudes (Figure 6). However, the object itself has luminance edges (i.e. contour of the object to segregate it from the background), and thus it would be possible that the edge of the moving object itself can induce the motion sensitivity regardless of the location of cells since cells can experience the edge of moving object. The authors should explain why the edge of the mask, not the edge of a moving object, induces motion sensitivity.

In our experiments and simulations, the masks had the same luminance as the background. In the absence of the stimulus (which could be brighter / darker than the background or have a complex pattern as the one used to create the simulations depicted in figure 5), the edge of the mask is invisible. The reviewer is absolutely correct that in a complex visual scene we examined in figure 5, the object itself can elicit motion sensitivity. We have performed two analyses to differentiate between contrast responses and reactions to novel objects. First, we compared responses from cells near the mask-emergence boundary to emerging objects with responses to the same stimuli in the absence of the mask (Supp fig 10), which revealed that responses near the site of object emergence are much different and enhanced by the mask edge, rather than the moving object.

Second, we simulated the responses to the same movies in cells that do not have surround inhibition. We show that cells in this model still respond to spatiotemporal contrasts but do not differentiate between the novel and established motion.

9) For the simulation (Figure 6), the intensity of the RF surround is not clear. In the method, the authors say "Where FactorSurround indicated the intensity of the surround activation and varied between 0 to 0.5". However, it is not clear if the intensity is varied among individual cells up to 0.5 or all cells have 0.5 in the C+S condition. Since the intensity of surround can be significant in the model, the authors should clarify that. In addition, if the intensity of surround can affect the results, the authors should model how much intensity can generate the motion sensitivity at the edge, and discuss if the surround intensity is in the physiological range of actual bipolar cell surround.

We have changed the description in the methods to the following: *Factor_{Surround} was set to 0.5 in all simulated cells in control conditions and 0 in all cells in center-only simulations.*

We now directly address the question of the role of the intensity of the surround (Figure 5f). As expected, information about the emergence of a novel object increases with a stronger surround, but only up to a point; afterward, the intense surround inhibition silences the cell and reduces its information content. We now state that the effect of the surround in our simple linear model is comparable to the effect of the surround in the detailed model of the retinal circuits (Figure 5 and Supplementary figures 7, 8). We also cite the relevant literature to show that the levels of inhibition we use are in the physiological range.

10) It is not clear how the authors identify the orientation of the dendrites of starburst amacrine cells from the labeled processes and grouped them into the edge near soma and edge near tips. Since the authors used Chat-Cre mice, there should be dense overlaps of dendrites from different starburst amacrine cells, which would make it difficult to determine whether the labeled dendrites are near the tips or the soma. The authors should clarify and explain.

We apologize for the unclear description of the thought experiment. We have changed the description of the main text to show that we consider the possible impact on postsynaptic cells without recording their responses. As the reviewer correctly noted, all SACs are labeled in Chat-Cre mice, making it challenging to determine dendritic orientation. What we did instead was to assume that all SACs receive a similar copy of the bipolar drive (we now cite evidence for and against this assumption). Next, we considered different alignment options for the SACs and the mask-stimulus boundary and the effect of novel object enhancement on the glutamatergic drive to SAC dendrites. Using two hypothetical SAC morphologies, we argue that motion processing in the presence of a mask can enhance or diminish the observed direction selectivity, depending on the orientation of the dendrites relative to the edge. Please note that a similar conclusion was reached by our colleagues (Strauss, Korympidou et al. 2021); we were not aware of their work when we submitted this manuscript.

11) In the Discussion, the authors discussed the potential role of motion sensitivity in bipolar cells in signaling the appearance of new objects to the visual centers. Although it is an interesting possibility, however, this manuscript has not yet shown that the motion sensitive outputs from bipolar cells actually influence the activity of postsynaptic ganglion cells and that the ganglion cells relay such motion-emergence-sensitive signaling to the visual centers. For example, it would be still possible that the bipolar cell outputs that are sensitive to motion emergence are rectified at the dendrites of ganglion cells, and thus not transmitted to visual centers. The authors should tone down the discussion or carefully explain the potential contribution to ganglion cell activity.

We agree. The discussion was changed accordingly.

12) The authors discussed the contribution to human perception. In addition to the visual saliency as the authors discussed, there has been a long history about the neuronal lateral inhibition on the center-surround antagonism and the visual illusion, like Mach bands or illusory contour (Pessoa, Vis Res 1996). The authors should discuss the motion sensitivity in the context

of the illusory enhancement of perception around the luminance edges.

Thank you for this suggestion. We have added the citation and the description of the effect to the main text

Minor comments:

1) Figure 1g, it is not clear what the black circles on some groups indicate.

Thank you for noticing this omission. The legend was changed to indicate that the circles indicate statistical significance

2) Figure 6e, it is not clear what the "DSIglutamate (%)" indicates (Figure 6e) indicates. From the method, DSI is ranged from 0 to 1. Although there are some texts around lines 215-216, this should be clarified in the method or figure legend with equations if needed.

Fixed

3) Figure 5c, the heat map of response amplitudes shows that there is another hotspot with higher amplitudes around the top right (not around the edge). The authors should explain why the higher amplitudes appear nowhere near the edge.

Based of the feedback of the reviewers, we decided to simplify the figure and remove the heatmap plot

4) The description of mutual information is not clear. The authors should provide an equation and clarify what the entropy was calculated from.

We have included an equation and explained the steps we took to quantify mutual information

5) The description in figure legend in Figure 4 (and the corresponding Method description) is not enough to understand the detail.

We have greatly expanded the description of the model in the methods section

i) What type of motion stimulus (emerging, exiting, or full-field) is used for the space-time plot in Figure 4b left and Figure 4c left?

We have changed the figure and these plots were removed from the final version

ii) Please clarify the reference (i.g. anatomical or physiological) for the number of photoreceptor inputs for horizontal cell ("n=150").

The number of photoreceptors was simply reflecting the spatial extend of the simulation (1500 μm) and the spacing between cells (10 μm). We have changed the description – now this information appears in the methods.

iii) The photoreceptors and bipolar cell synapses are not clear. What is the number of photoreceptor inputs for a bipolar cell is not shown?

We now state explicitly in the methods that the size of the BC RF is a gaussian with a 50 μm full width at half maximum. The gaussian indicates the strength (weight) of the synapse, ranging from 1 from cells that are located at the same spatial coordinates to near zero for large distances. The actual implementation in the model was that all BCs sampled from all photoreceptors, but this is a bit misleading because photoreceptors located more than ~ 100 μm away had virtually no impact on BC responses.

iv) How do bipolar cells integrate photoreceptors within RF (a 50x50 μm square)?

For the two-dimensional models, cells had a two-dimensional Gaussian RF. As in the detailed retinal simulation, the shape of the RF set the strength of response. Because we wanted to create the simplest model possible with a low number of free parameters, we did not explicitly model photoreceptors. Instead, the luminance of the stimulus was convolved with the shape of the RF.

v) What is the "dimensionality-corrected RF"?

Removed and replaced with a Gaussian

Reviewer #2 (Remarks to the Author):

The manuscript by Gaynes and colleagues explores the responses of bipolar cells (BCs) in the mouse retina to the appearance of static objects and various kinds of motion. The authors

measure glutamate release throughout the inner plexiform layer (IPL) with a ubiquitously expressed virus (Figs. 1-3) and specifically onto the neurites of starburst amacrine cells (SACs) with a SAC-specific mouse line (Fig. 6).

The main finding of the study is that objects that suddenly appear in the receptive field (RF) center of BCs (either by flashing there or emerging from an occluder) drive larger and faster responses than smoothly moving stimuli that activate the BC RF surrounds before the RF center.

While the data and analyses are generally of high quality, the impact of the manuscript is substantially diminished by its presentation. In many places, the authors over-interpret their results, and they frequently fail to place their data in the context of historical results and models of center-surround RFs.

The core result here is a trivial consequence of a linear center-surround RF, as the authors point out in their linear model in Fig. 4. A stimulus that hits the inhibitory RF surround and then the excitatory RF center will give a weaker response than one that appears suddenly in the RF center. The magnitude of this effect will depend on both the strength of the RF surround and, for moving objects, the center-surround delay and their relative kinetics. This is a form of a Reichardt detector (Von Hassenstein and Reichardt, 1956) as proposed as a mechanism for direction selectivity by Barlow in 1965 (Barlow and Levick, 1965) and investigated in great detail in the fly visual system (Borst and Euler, 2011; Borst et al., 2020). Language throughout the manuscript frames this as an unexpected discovery or over-interprets the result. Some examples from the abstract:

“Center-surround RFs are thought to enhance responses to spatial contrasts (i.e., edges), but how they contribute to motion processing is unknown”

There are many, many papers on how center-surround RFs contribute to motion processing. Lots of them are cited in (Borst and Euler, 2011).

“... their responses to continuous motion are smaller, slower, and cannot be predicted by signals elicited by stationary stimuli.”

Actually, a linear model does exactly that! The spatiotemporal RF inferred by static presentations of bars will predict a smaller, slower response to smooth motion across the RF surround and then center than a flashed bar in the RF center.

“The alteration in signal dynamics induced by novel objects dwarfs the enhancement of spatial edges and can be explained by priming of RF surround during continuous motion.”

This is not a quantitative comparison in the paper – seems like comparing apples to oranges in

this statement – and “priming” is an unnecessarily confusing term (that implies facilitation rather than suppression) for a stimulus that simply hits the RF surround before the center, as described above.

“...demonstrate an unappreciated capacity of the center-surround architecture to facilitate novel object detection...”

This has been appreciated for over 50 years.

“...and multiplexed encoding of distinct sensory modalities.”

Extremely vague. There is only one sensory modality being investigated here. Is multiplexed supposed to mean that BCs respond both to flashed bars and moving objects? That’s not a novel finding for BCs or for any visual neuron.

The reviewer presents an accurate description of the knowledge in the field regarding motion processing in neurons with center-surround RFs. We appreciate the insight from the expert in the field of signal processing in neurons with center-surround RFs. We changed the language used to describe the contribution of linear center-surround RFs (to our results/findings) to be more precise and present our findings in a more neutral tone that does not diminish the significance of the rich history that already exists for this topic.

In the next paragraphs, we would like to address the reviewer's comments by highlighting the novel findings presented in this work and how they fit or negate current and historical concepts.

First, the term ‘motion processing’ is an umbrella definition that can potentially incorporate very distinct concepts. In our manuscript, we distinguish between (1) the classical notion of established motion and (2) our finding of the ability of center-surround RFs to emphasize emerging (+novel) objects. As we will discuss next, we think that our findings and the analysis we performed provide important additions to knowledge in both regards.

Processing of established motion

We agree with the reviewer that the expected signal transformation from static flashes to continuous motion should make the signal slower and smaller. This is by and large what we found (although some of the glutamate clusters seem to break the rule). We have never framed this part of our results as surprising or unexpected. For example, below is the sentence we used to describe these findings in the results section (emphasis added) “...**As expected**, slower RF engagement prolonged motion response kinetics (Fig. 1d-f)...”.

While we consider the description of response dynamics to motion valuable (note that while one would predict slower/smaller RF engagement by motion, it is impossible to quantify the

effect without empirical measurements), the main finding is different. In contrast to expectations and predictions of linear models, we revealed that responses to motion could not be predicted from the representation of static flashes. This is a highly unexpected finding that contradicts the knowledge in the field and the assumptions of influential models that investigated the role of BC dynamics in promoting direction selectivity (Kim, Greene et al. 2014, Fransen and Borghuis 2017, Matsumoto, Briggman et al. 2019). In these models, the authors relied on the time course of BC responses to static stimuli to estimate the possibility of Reichardt detector-like interactions between BC signals during established motion. Our findings indicate that the postsynaptic signals are likely to be very different from what is predicted by the shape of the responses to static flashes. Specifically, we observed little correlation between the rise time of BC signals to static and moving stimuli. Importantly, the difference in speed of activation between clusters, the key aspect of the Reichardt mechanism, was found to be reversed between signals analyzed from static and motion responses for some of the glutamate clusters, as we now discuss in the text.

The emphasis of emerging (novel) objects

In isolation, stronger responses to flashed vs. moving stimuli is not particularly interesting. However, a potential for a new view of center-surround RF function arises when considered together with the ability of center-surround computations to emphasize other signals that suddenly appear in the RF (we focus on emerging stimuli, a study by our colleagues (Strauss, Korympidou et al. 2021) report similar effects for looming stimuli).

The reviewer is correct that the detection of newly appearing objects by of center-surround RFs relies on simple interactions between RF components that can be fully captured in a linear model. We consider this a strength and not a weakness! Our experimental work is largely restricted to the analysis of responses from a single cell type in the retina. But because our conceptual findings apply to all visual neurons with a center-surround RF, the rules of signal processing we describe are not a peculiar feature of BC, but rather can be relevant in most cells the retina, LGN, and even V1.

We believe that the message the reviewer wants to convey is that our findings are not sufficiently novel because they could be deduced from the principles of the center-surround RF formulation. This is a fair criticism that we received on several occasions. Our response is that while the responses for each of the stimuli could be readily predicted from center-surround RFs (for example, in a parallel publication by Strauss and colleagues recorded emerging motion enhancement in IPL, which they explain with a similar model of center-surround RF interactions), **we propose a framework that provides a new perspective on the computational role of center-surround RFs in visual processing.**

When Kuffler, Hartline and others pioneered the use of RF mapping to describe center-surround interactions in the 1950s, they built upon a small set of experiments to propose a theory of center-surround RF function in detecting contrast gradients or edges. This work was continued by numerous researchers who refined and greatly expanded on this concept. However, there is no question that center-surround architecture is not always able to detect edges. For example, some ganglion cells, like local edge detectors, are exceptionally good in responding to spatial contrasts, but many other types do not have a clear edge preference. Furthermore, some visual stimuli break the normal RF interactions and produce strange perceptual experiences. As reviewer #1 noted, such optical illusions are a valuable tool to study the RF composition in humans and other animals.

Similarly, we used a small set of stimuli to probe signaling in BCs in the retina. Based on the responses we recorded, we propose a new role for the center-surround organization. We did not intend to convey that BCs will detect novel objects in all visual scenes and did not draw this conclusion in the manuscript (see figure 5). The conceptual advantage of our work lies in the fact that we provide a new framework in which to consider and analyze center-surround interactions. We propose a computation that is both consistent with previous understanding of center-surround structure and serves a crucial perceptual role – and is likely to influence human vision.

Please also note that many potential mechanisms can enhance novel object detection in the retina and beyond. In this manuscript, we decided to limit our numerical simulations to the basic building blocks of the RF. We reasoned that the focus of the fundamentals would highlight the ubiquitous nature of the computation of novel object detection. The fact that a substantial fraction of new objects in the visual scene can be described as visual items that suddenly appear in the RF or first engage RF center is significant. It explains how a non-trivial computation that is vital to the survival of the organism arises from a simple RF organization.

Next, we would like to comment on the specific points raised by the reviewer:

This is a form of a Reichardt detector (Von Hassenstein and Reichardt, 1956) as proposed as a mechanism for direction selectivity by Barlow in 1965 (Barlow and Levick, 1965) and investigated in great detail in the fly visual system (Borst and Euler, 2011; Borst et al., 2020).

Our colleagues did, in fact, use the Levick detector model to explain similar findings (Strauss, Korympidou et al. 2021). We think that this is inaccurate. First, we show that motion processing in BC is not direction-selective*, which both the Reichardt and Levick detectors are. Second, to make their model work, Strauss et al. considered motion processing on a single spatial dimension and flanked the center of the BC with two inhibitory surrounds that formed two Levick-like circuits. We note that this complex model could not be readily extrapolated to two

dimensions – whereas the model we propose here is both simpler and has the correct spatial dimensionality.

* a recent work revealed that a small minority of BC boutons could be direction selective (Matsumoto, Agbariah et al. 2021). Our motion effects are observed in much broader BC populations (Figure 2). We did not observe direction selectivity in full-field stimuli measured responses (Figures 2, 6), which is likely due to the factors discussed by Matsumoto et al., 2021, the primary one being that the directional tuning in BC is a property of a small number of inputs which would be challenging to observe in our experimental conditions without sparse labeling of selected BC populations.

“Center-surround RFs are thought to enhance responses to spatial contrasts (i.e., edges), but how they contribute to motion processing is unknown”

There are many, many papers on how center-surround RFs contribute to motion processing. Lots of them are cited in (Borst and Euler, 2011).

We agree, our wording was not precise. Many studies have shown how spatially **asymmetric** RFs can contribute to direction selectivity. We thank the reviewer for this comment.

Correspondingly, we have replaced this statement with the following: “**Circularly symmetric** center-surround RFs are thought to enhance responses to spatial contrasts (i.e., edges), but their contribution to motion processing is unknown” and also state throughout the manuscript that we focus on spatially symmetric RFs.

“The alteration in signal dynamics induced by novel objects dwarfs the enhancement of spatial edges and can be explained by priming of RF surround during continuous motion.”

This is not a quantitative comparison in the paper – seems like comparing apples to oranges in this statement – and “priming” is an unnecessarily confusing term (that implies facilitation rather than suppression) for a stimulus that simply hits the RF surround before the center, as described above.

We have replaced the sentence with: “*In our hands, the alteration in signal dynamics induced by novel objects was more pronounced than edge enhancement*”. We now perform a qualitative analysis of the difference between edge and emerging motion enhancements (Figure 2). The comparison is accurate - we examined the responses in the same ROIs to edges at the same spatial condition for flashed or moving objects. We have added the statement that this happened *in our hands*, as it is possible that the observations would be different for other visual stimuli.

We have considered replacing the term 'priming' with 'conditioning' or 'enhancing', but believe that those alternatives are not significantly different from 'priming' to describe the facilitation of the surround by established motion.

{Hubert, 1976 #112}

“...demonstrate an unappreciated capacity of the center-surround architecture to facilitate novel object detection...”

This has been appreciated for over 50 years.

As far as we know, our manuscript is the first to suggest and to show the ability of the classical center-surround organization emphasize objects which enter the visual scene from behind an occluder (the term we use in the text is 'emerging objects'). As we describe above, our proposed view of the function of the classical, symmetric center-surround RF is radically different from the prevailing view, which is that these RF promote spatiotemporal contrast detection.

Specific issues:

1. The BC clustering in Fig. 1 is suspect. Most of the clusters within the ON and OFF sub-classes have properties within 1 s.d. of each other. The McClain-Rao index (Fig. 1c) is the ratio of the mean within-cluster distances to the mean between-cluster distance. Thus, optimal clustering is defined by the MINIMUM of this quantity, not the maximum! And why does the graph stop at 3 clusters instead of 2? The point chosen for OFF clustering has a McClain-Rao index of ~0.95 meaning that between between-cluster distances are only 5% larger than within-cluster distances; this indicates pretty terrible clustering, as suggested by the fact that the stratification profiles for the 5 OFF clusters are almost identical. With the exception of C6, the ON clusters are also much more similar than I would expect given previous work (Franke et al., 2017), and the authors make no effort to link each cluster to a known BC type by the published stratification profiles.

The reviewer is correct. We had a bug in our computation of the McClain-Rao index. Instead of implementing clustering in our code, we now use R to cluster the glutamate responses and calculate the best number of clusters.

We have limited the number of functional clusters to be between 3 and 11 for each response polarity to limit the number of clusters to a physiologically relevant range. We found local minima at 2 clusters and between 16-20 clusters. But as we do not think these numbers correspond to a biological number of cell types, we have decided not to show that data.

Prompted by the reviewers' suggestion, we now try to link functional cluster identity with anatomical cell types. Please see our responses to Reviewer #1, points 1b and 1c. In general,

many BCs have nearly identical stratification profiles (types 1 and 2; 3 and 4; X and 5; 8/9 and rod bipolar), which precludes exact correspondence between functional and anatomical clusters (Franke, Berens et al. 2017, Matsumoto, Briggman et al. 2019). We have used the innervation pattern on starburst amacrine cells, whose BC contacts are known (Kim, Greene et al. 2014, Ding, Smith et al. 2016, Greene, Kim et al. 2016), and light response measurements made in previous studies (Franke, Berens et al. 2017, Matsumoto, Briggman et al. 2019) to make educated estimations of the identity of our functional clusters.

2. Fig. 3 reports that the "novelty detection" mechanism (BC surrounds) does not depend on inhibition from amacrine cells (ACs) because it remains intact in a complete GABA and glycine receptor block. This is the most surprising result in the manuscript. There has been significant controversy over the years about the relative contributions of inner vs. outer retinal mechanisms to BC surrounds, and a study using the same technique (iGluSnFR to measure mouse BC outputs) reported that the same pharmacological manipulation had dramatic effects on the temporal responses of BCs that were most apparent for full-field light stimuli, suggesting a role for the RF surround (Franke et al., 2017). Further dissection of the circuit mechanism is possible here by using each of the receptor blockers (for glycine, GABA_A, and GABA_C) alone and in pairwise combinations. Additionally, if the authors are indeed claiming that BC surrounds are dominated by an outer retinal mechanism, they should be eliminated in by pH buffering with HEPES (Davenport et al., 2008) or using a Cx57 KO animal (Shelley et al., 2005).

We agree with the reviewer that this result is surprising, although we note that our other main finding of kinetics and amplitude of responses to motion that can not be readily predicted from the shape of the responses to static flashes and the presence of the computation of emerging object enhancement in retinal bipolar cells were also unexpected.

Following the suggestions from the reviewers, we have performed experiments to block horizontal cell feedback. As we describe in detail in introductions, in line with our original experimental results and models, horizontal cell blockage was highly effective in negating edge effects – both for stationary flashes and moving bars. Given this direct validation of the role of horizontal cells, we can now be much more confident that motion processing we describe begins at the outer plexiform layer.

Initially, when we planned our experiments, we assumed that amacrine cells should form the surround that guides motion computations, based on the impact of amacrine cells on BC responses reported by (Borghuis, Marvin et al. 2013, Franke, Berens et al. 2017). Consequently, we performed several experiments with a single blocker (strychnine/gabazine). In contrast to our predictions, we did not observe a noticeable effect on glutamate release in these pilot

experiments. We, therefore, decided to combine the blockers into a cocktail to see if amacrine cells impact motion processing in BCs – the negative result of these experiments appears in the manuscript.

As we have responded to Reviewer #1, we think that amacrine cells can affect BC responses to novel or established motion. However, as our empirical findings show, amacrine cells are not required to generate motion responses, at least in the BC populations we imaged in the GABAergic and glycinergic blocker experiments. It is possible (and even likely) that amacrine cells have more pronounced effects in specific BC or amacrine cell subpopulations. While the lab is interested in understanding the shaping of motion responses by BC-AC interactions in the IPL, the required number of experiments for this line of investigation is beyond the scope of the present study. Our current dataset contains full-field responses from over 1800 ROIs. Only ~30% of these ROIs were located near an edge. We estimate that each of the glutamate clusters should be represented by about 30 ROI to form a reliable comparison of the pharmacological effect. This back-of-the-envelope calculation shows that a detailed investigation of amacrine cell inhibition would require at least ~1200 ROIs for each blocker or a pairwise combination thereof (>7000 ROIs overall). Note that we assume that all the clusters are observed with equal probability; this is not the case – requiring even more experiments to account for the rare BCs. Further, pharmacological experiments are notoriously messy and difficult to interpret as they affect numerous cells and circuits. Given the direct demonstration of motion processing being dependent on horizontal feedback, we do not think these experiments would significantly change the conclusion that the motion computation begins at the photoreceptor-horizontal cell synapse

3. The authors cite two other important papers that have looked at motion processing in BCs in the mouse (Kuo et al., 2016) and primate (Manookin et al., 2018) retina, but they don't really address the fact that these studies used a different definition of motion sensitivity that has also been adopted in studies of downstream areas, like superior colliculus (Gale and Murphy, 2014). Both these studies showed that BCs responses to smooth motion more strongly than to a randomized sequence of bar presentations, so this is OPPOSITE to the effect reported here. The mechanism for this form of motion sensitivity required electrical coupling between BCs (via All ACs) and the BC output nonlinearity. An important and unresolved question in this field is whether OFF BCs (which are not coupled via All ACs) can have stronger responses to motion order than random order bar presentations.

As the reviewer correctly points out, previous studies have compared motion responses to randomized sequences of their stimulus presentation. In essence, they shuffled the frames used to create the original stimulus movies and showed them out of order. We disagree that our

study contradicts these findings as we do not know how BCs would respond to similar shuffling, and (Manookin, Patterson et al. 2018, Liu, Hong et al. 2021) did not present full-field/masked stationary flashes.

We have decided not to use the analysis performed in these studies for several reasons. First, 'motion sensitivity' is a somewhat subjective term that has a different meaning in our study. The question our colleagues asked was how is the representation of continuous motion different from the out-of-order stimulus presentation. They never tried to address the differences in the processing of other types of correlated stimuli, such as flashes and did not devise the analysis pipeline required to make such comparisons. The stimulus used by Manookin et al., 2018, (as well as conceptually similar stimuli in other studies) consisted of a presentation of a single bar at a time. The bar was shifted in sequence or randomly to create apparent motion or out-of-order videos, respectively. This approach would fail for static full-field flashes because all the bars are presented simultaneously to create the full-field flash. There is no sequence of activation for such stimuli. We could hypothetically present individual bars in separate trials. However, this analysis scheme is different than the out-of-order sequence analysis, and the two cannot be directly compared to each other.

Second, note that in our stimulation paradigm, emerging motion stimulus and motion towards an edge corresponded to the same stimulus movie played in reverse. Because both stimuli contain the same video frames, the methodology of Manookin et al., 2018 would suggest comparing the responses to these stimuli to the same shuffled video decomposition. We doubt that such analysis could be informative – how is it better than simply comparing the responses to each other?

Last, while the apparent motion paradigm was used in the past to probe direction selectivity (Fried, Munch et al. 2002, Lee and Zhou 2006), other studies of motion detection in BC or GC (for example, (Jain, Murphy-Baum et al. 2020, Ding, Chen et al. 2021, Matsumoto, Agbariah et al. 2021, Sethuramanujam, Matsumoto et al. 2021)) that come after Manookin et al 2018 have not made comparisons to out-of-sequence stimuli and therefore is not an established standard in this field, and furthermore is not relevant to our study for the reasons described above.

References

- Barlow, H.B., and Levick, W.R. (1965). The mechanism of directionally selective units in rabbit's retina. *J. Physiol.* 178, 477–504.
- Borst, A., and Euler, T. (2011). Seeing Things in Motion: Models, Circuits, and Mechanisms. *Neuron*.
- Borst, A., Haag, J., and Mauss, A.S. (2020). How fly neurons compute the direction of visual motion. *J. Comp. Physiol. A Neuroethol. Sensory, Neural, Behav. Physiol.* 206, 109–124.

Davenport, C.M., Detwiler, P.B., and Dacey, D.M. (2008). Effects of pH Buffering on Horizontal and Ganglion Cell Light Responses in Primate Retina: Evidence for the Proton Hypothesis of Surround Formation. *J. Neurosci.* 28, 456–464.

Franke, K., Berens, P., Schubert, T., Bethge, M., Euler, T., and Baden, T. (2017). Inhibition decorrelates visual feature representations in the inner retina. *Nature* 542, 439–444.

Gale, S.D., and Murphy, G.J. (2014). Distinct representation and distribution of visual information by specific cell types in mouse superficial superior colliculus. *J Neurosci* 34, 13458–13471.

Von Hassenstein, B., and Reichardt, W. (1956). Systemtheoretische Analyse der Zeit-, Reihenfolgen- und Vorzeichenbewertung bei der Bewegungserkennung des Rüsselkäfers *Chlorophanus*. *Zeitschrift Fur Naturforsch. - Sect. B J. Chem. Sci.* 11, 513–524.

Kuo, S.P., Schwartz, G.W., and Rieke, F. (2016). Nonlinear Spatiotemporal Integration by Electrical and Chemical Synapses in the Retina. *Neuron* 90, 320–332.

Manookin, M.B., Patterson, S.S., and Linehan, C.M. (2018). Neural Mechanisms Mediating Motion Sensitivity in Parasol Ganglion Cells of the Primate Retina. *Neuron* 97, 1327–1340.e4.

Shelley, J.A., Dedek, K., and Weiler, R. (2005). Effects of Connexin57 Deletion on Horizontal Cell Receptive Field Size in the Mouse Retina. *Invest. Ophthalmol. Vis. Sci.* 46, 600–600.

Reviewer #3 (Remarks to the Author):

Review of NCOMMS-21-16455, 'Novel Object Detection and Multiplexed Motion Representation in Retinal Bipolar Cells'

This study focuses on the visual encoding properties of retinal bipolar cells. Response properties of these cells are an re-emerging field of study and this work is timely. The authors use two-photon fluorescence glutamate imaging to measure stimulus-evoked responses of functionally identified bipolar cell types and report that responses to visual stimuli that first appear in the receptive field center are signaled comparatively stronger than those that originate outside it. This phenomenon is captured by a computational model with a dependence on horizontal cell signaling. Model analysis shows that it improves novel object detection. While this response property renders bipolar cell output particularly sensitive to visual object motion emerging within the receptive field, it does not establish direction selectivity in the bipolar cells themselves, nor does it appear to enhance direction tuning at subsequent signaling stages, but see point 4 below. The main impact on visual encoding appears to be increasing salience for novel, local movement.

I have the following comments and concerns.

1. Abstract, line 15 states that the alteration in signal dynamics induced by novel objects' dwarfs' the enhancement of spatial edges. This is a subjective statement, and also appears to be an overstatement. It is not made clear exactly which is considered the measurement of enhancement of spatial edges. If I look at Figure 2c, then I see that the response to novel motion is about 25% larger than that to a full field edge stimulus. Normalizing the responses to the full field edge stimulus necessarily puts that at 1.0 (bottom of the graph), but novel moving stimulus response is only at 1.25 df/f.

We have replaced the sentence with: *"In our hands, the alteration in signal dynamics induced by novel objects was more pronounced than edge enhancement"*. Following this comment, we decided to directly measure the difference in static edge enhancement for each glutamate cluster and compare it to the level of enhancement of emerging stimuli. We have decided to follow standard practice and quantify edge detection from responses to static flashes. Correspondingly, to quantify static edge enhancement, we divided the peak of the static response when the ROI was less than 50 μm away from the stimulus/mask boundary by the peak of the full-field static flash stimulation.

We reasoned that enhancement of emerging motion should be compared to other motion responses. Therefore it was quantified as the ratio between the amplitude of the emerging motion response to the response to motion towards the mask. The results of this analysis for different functional clusters are now shown in Figure 2.

2. Title and main text refer to 'multiplexed encoding' in bipolar cells based on the presented results. While the results convincingly show that specifics of the stimulus configuration including motion and position of an occluder changes the response rise time and peak amplitude, the term multiplexed implies that these specifics can also be separately decoded. It is not at all clear to me how that would work: the bipolar cell is a single encoding channel, and the mode of encoding may not be constant – as following a change in mean luminance, or contrast. The response properties help increase salience, but they do not appear to provide independent (orthogonal) modes of visual encoding, which is the image that 'multiplexed' conjures up for me. This needs to be addressed.

We agree that the word 'multiplexed' brings unintended associations. This is one of the reasons we have decided to change the title, to avoid any unambiguity. We changed the abstract to read 'dynamic signal representation'. The term 'multiplexed' was replaced with more precise wording in the main text.

3. Fig. 6, title states that [BC sensitivity to novel objects] 'influences the analysis of motion processing...'. It should be made unambiguous here and in Results what is the nature of this influence.

Prompted by the comments made by the reviewer, we have rewritten the corresponding part of the results and the figure legend.

4. The results of glutamate imaging on SAC dendrites are murky. This is due in part to experimental limitations (imaged SAC branch orientation not controlled; stimulus edge near soma of some, distal for other BCs), and the write-up of these results. Specifically, the authors report a DSI of $32 \pm 21\%$ for bipolar cell glutamate signal impinging on SAC dendrites. But the mask-stimulus boundary is said not to contribute to DS in SACs (line 218-220). The explanation (lines 220-228) is very difficult to follow and needs to be elaborated/illustrated to be accessible for readers. It should be made crystal clear whether BC motion enhanced responses increase direction tuning as measured by DSI, or if it merely boosts the (non-DS) transmitted signal to the SAC.

Both points refer to the same section of the results. We have rewritten the relevant part, specifically focusing on the remarks made by the reviewer. The data and the analysis are presented differently, hopefully in a clearer format.

5. Related to the previous point, conceptually, it is not clear how the enhanced response to novel motion can lead to an increased DSI in the SAC dendrite. The bipolar cell response is increased for motion away from the center – but that can be in any direction. Any bipolar cell connected to a SAC dendrite of particular orientation therefore will provide increased glutamatergic drive to this SAC dendrite depending on the position of the edge/occluder, independent of whether or not this is the preferred direction for that SAC dendrite.

This was indeed our point. We decided to present the experiments and the theoretical framework to illustrate a potential caveat in previous experimental investigations of direction selectivity in SACs. Our data clearly demonstrate that the experimental approach taken in many previous studies could inadvertently influence the estimation of directions selectivity. We have decided to use the DSI as it is the standard metric of direction selectivity; it allows a quick estimation of the fraction of direction selectivity observed in these studies that could be attributed to motion from edge enhancement in BCs.

6. Figure 1e, left panel title. I understand the need to abbreviate 'Stationary flashes' (as used in d). Instead of 'Stationary' consider using 'Flashes' instead, because it better represents what was presented.

Thank you for the suggestion. We have updated all titles accordingly

7. Figure 2 shows responses within ROIs classified as transient and sustained. Were ON and OFF responding ROIs (presumed ON and OFF bipolar cell types) combined in this analysis? If this is stated somewhere, I could not find it; consider making this explicit in Results section and in Fig2 legend or figure panels.

Thank you for this suggestion. We now make it clear in the text that we combine ON and OFF responses.

8. I find Figure 4 panel b uninterpretable. The bottom part is a space time plot, with responses from two cell types overlapping, and six different stimulus configurations indicated with arrows at the top. To make this accessible, consider splitting this out in a few panels. Similarly, pane, c, how does the continuous wave form relate to the six stimulus configurations above. This is too convoluted to parse easily.

Based on the feedback, we have redesigned the figure. We focused on the similarity of presentation of our experimental and modeling data and removed the space-time plot entirely.

9. Many parameters went into the construction of the model presented in Fig. 4. Legend refers to one of them - BC sensitivity to photoceptive (sic) release – it would be useful to give more information about the model including the parameters and sensitivity of the main results to the parameter values used in the simulations.

Thank you for noticing the typo. We have greatly expanded the methods section to give more insights into the model. Also, note that we now describe the effects of the different parameters in Figure 4 and Supplementary figure 8.

10. Line 72 reports the lack of correlation between flash- and motion-driven rise time. Text here says Pearson cc of -0.04; text in Figure 1F says -0.03. Which is it?

We apologize for the mistake. We have recalculated the difference between the rise times of static and motion-driven signals in our larger dataset. We found a somewhat stronger correlation $r^2=0.09$, which did not significantly change our interpretation of the data.

References

Borghuis, B. G., J. S. Marvin, L. L. Looger and J. B. Demb (2013). "Two-photon imaging of nonlinear glutamate release dynamics at bipolar cell synapses in the mouse retina." The Journal of neuroscience : the official journal of the Society for Neuroscience **33**(27): 10972-10985.

Chen, M., S. Lee and Z. J. Zhou (2017). "Local synaptic integration enables ON-OFF asymmetric and layer-specific visual information processing in vGluT3 amacrine cell dendrites." Proc Natl Acad Sci U S A **114**(43): 11518-11523.

Ding, H., R. G. Smith, A. Poleg-Polsky, J. S. Diamond and K. L. Briggman (2016). "Species-specific wiring for direction selectivity in the mammalian retina." Nature **535**(7610): 105-110.

Ding, J., A. Chen, J. Chung, H. Acaron Ledesma, M. Wu, D. M. Berson, S. E. Palmer and W. Wei (2021). "Spatially displaced excitation contributes to the encoding of interrupted motion by a retinal direction-selective circuit." Elife **10**.

Franke, K., P. Berens, T. Schubert, M. Bethge, T. Euler and T. Baden (2017). "Inhibition decorrelates visual feature representations in the inner retina." Nature **542**(7642): 439-444.

Fransen, J. W. and B. G. Borghuis (2017). "Temporally Diverse Excitation Generates Direction-Selective Responses in ON- and OFF-Type Retinal Starburst Amacrine Cells." Cell Rep **18**(6): 1356-1365.

Fried, S. I., T. A. Munch and F. S. Werblin (2002). "Mechanisms and circuitry underlying directional selectivity in the retina." Nature **420**(6914): 411-414.

Greene, M. J., J. S. Kim, H. S. Seung and EyeWriters (2016). "Analogous Convergence of Sustained and Transient Inputs in Parallel On and Off Pathways for Retinal Motion Computation." Cell Rep **14**(8): 1892-1900.

Jain, V., B. L. Murphy-Baum, G. deRosenroll, S. Sethuramanujam, M. Delsey, K. R. Delaney and G. B. Awatramani (2020). "The functional organization of excitation and inhibition in the dendrites of mouse direction-selective ganglion cells." Elife **9**.

Kim, J. S., M. J. Greene, A. Zlateski, K. Lee, M. Richardson, S. C. Turaga, M. Purcaro, M. Balkam, A. Robinson, B. F. Behabadi, M. Campos, W. Denk, H. S. Seung and EyeWriters (2014). "Space-time wiring specificity supports direction selectivity in the retina." Nature **509**(7500): 331-336.

Lee, S. and Z. J. Zhou (2006). "The synaptic mechanism of direction selectivity in distal processes of starburst amacrine cells." Neuron **51**(6): 787-799.

Liu, B., A. Hong, F. Rieke and M. B. Manookin (2021). "Predictive encoding of motion begins in the primate retina." Nat Neurosci.

Manookin, M. B., S. S. Patterson and C. M. Linehan (2018). "Neural Mechanisms Mediating Motion Sensitivity in Parasol Ganglion Cells of the Primate Retina." Neuron.

Matsumoto, A., W. Agbariah, S. S. Nolte, R. Andrawos, H. Levi, S. Sabbah and K. Yonehara (2021). "Direction selectivity in retinal bipolar cell axon terminals." Neuron.

Matsumoto, A., K. L. Briggman and K. Yonehara (2019). "Spatiotemporally Asymmetric Excitation Supports Mammalian Retinal Motion Sensitivity." Curr Biol.

Sethuramanujam, S., A. Matsumoto, G. deRosenroll, B. Murphy-Baum, J. M. McIntosh, M. Jing, Y. Li, D. Berson, K. Yonehara and G. B. Awatramani (2021). "Rapid multi-directed cholinergic transmission in the central nervous system." Nat Commun **12**(1): 1374.

Strauss, S., M. M. Korympidou, Y. Ran, K. Franke, T. Schubert, T. Baden, P. Berens, T. Euler and A. L. Vlasits (2021). "Center-surround interactions underlie bipolar cell motion sensing in the mouse retina." bioRxiv: 2021.2005.2031.446404.

REVIEWER COMMENTS

Reviewer #1 (Remarks to the Author):

The authors have done an excellent job of incorporating my suggestions. I have no further comments.

Reviewer #2 (Remarks to the Author):

The authors have made an effort to improve the manuscript, have performed some new experiments, and have tried to respond to the previous round of reviews. Unfortunately, major issues remain, and the response to the reviews was not as complete as it should have been. I cannot recommend publication of this manuscript in Nature Communications.

Major issues:

1. Imprecise and incorrect language persists:

There are numerous examples. Here are two of them

“However, computations in cells with center-surround RFs are inherently nonlinear...”

Why? This is just incorrect. I can guess what the authors meant, but this statement as written does not make any sense. A center-surround RF is not “inherently nonlinear”. They cite some recent papers about the fact that bipolar cells may actually integrate space nonlinearly, which means that a linear center-surround RF is an incomplete model. The authors may be talking about space-time separability, which is a different question than spatial nonlinearity. Indeed, BC RFs are probably not space-time separable either. This is the kind of question their experiments could actually answer. But they don’t state the question accurately or address it. Statements like this will really confuse readers.

“...continuous motion and disappearing stimuli suppress glutamate signaling...”

No. They may activate BCs less than onset motion, but that’s not the same thing as “suppressing” glutamate signaling. In a second revision of a manuscript, reviewers should not be correcting misstatements this important.

2. Authors still show a lack of understanding of previous literature

The authors still show a lack of understanding of both the historical literature about receptive fields and more recent work on motion responses in the retina. They don’t even cite probably the most relevant previous paper on motion onset responses, which include a model incorporating bipolar cells (Chen et al., 2013).

As far as the classic Reichardt detector literature, they incorrectly make the assertion that these papers only considered an asymmetric RF and never considered how inhibition and excitation could interact if they were circularly symmetric. This is false. While of course a model specifically made to explain direction selectivity requires some spatial asymmetry, the fact that the same mechanism can explain motion responses depending on the spatiotemporal integration of the RF surround and center is not some kind of revolutionary concept that had been overlooked for 60 years! Indeed, some researchers oversimplified and didn’t think about this in their models, but others did, and the core idea dates back at least to Barlow.

3. The new model (Fig. 4) appears to be all about the kinetics of the inhibition rather than its location.

The model in Fig. 4 assumes sustained HC feedback and transient (and more delayed) AC feedback with no real justification for those choices. The kinetics of the inhibition (and its spatial extent and strength which are kept fixed) will determine how the motion onset response compares to the smooth motion response. The authors seem to be manipulating the wrong things in the model and make the false HC/AC dichotomy. It's just a linear RF model. The strength and timing of the surround are what matters, not its circuit location.

And don't repeatedly refer to your own model as "detailed". This can only frustrate a reader about all the details that are oversimplified or left out.

4. Responses to the specific suggestions in my previous review were unsatisfactory

A. Clustering.

The authors had a bug in their previous clustering, so they used a different method (which is described in the Methods but only referred to by the programming language in the Response). The authors state in the response that the new method found local minima at 2 and 16-20 clusters, yet they show the data from 3-11 clusters as that is a better fit to the known number of BCs. And they don't mention in the paper that this was not the real minimum. This is intentionally misleading!

B. HC manipulation

While it's nice that the authors did a manipulation directed at HC, their manipulation has a fatal flaw in that it also eliminates ACs! CNQX blocks AMPA receptors on ACs! Myself and another reviewer suggested HC-specific manipulations (pharmacological or genetic), and while the pharmacological experiments may have represented an incomplete block or had some off-target effects, they are not as problematic as the choice to include CNQX and block all the ACs!

I suggested using the individual GABA receptor blockers one at a time and the authors simply made excuses for not performing that experiment.

C. Other forms of motion stimuli are not tested, including the one specifically requested. The arguments for not testing this stimulus are thin and don't make a lot of sense. It's not a difficult experiment, and it helps connect this work to previous work on the role of bipolar cells in motion detection. The authors claim in the introduction that bipolar cells have rarely been considered to be important in motion detection. Yet they refuse to do the experiments to connect to the few papers that have made this connection!

Reference

Chen, E.Y., Marre, O., Fisher, C., Schwartz, G., Levy, J., Silviera, R.A. da, and Berry, M.J. (2013). Alert Response to Motion Onset in the Retina. *J. Neurosci.* 33, 120–132.

Reviewer #3 (Remarks to the Author):

Review of revised manuscript NCOMMS-21-16455A.

Related to my point 4: The presentation of the BCSAC glutamate imaging experiments (Lines 73-95) is a great improvement.

Line 322 States that 'The earliest direction-selective signals are present in dendrites of starburst amacrine cells', and the authors' results show that motion processing in BCs is not direction-selective. It appears relevant to relate these results to recent work - the most detailed study of BC glutamate output during motion stimulation, Matsumoto, ..., Yonehara (Neuron, Sep2021; not cited) which showed direction tuning of individual boutons on axon terminals on some bipolar cell types, including those referenced in the

current study. This direction tuning may still originate in SACs – so the opening sentence may be factually correct, given the contentious nature of every claim DS, a concise reference of earlier/conflicting findings is warranted, at least in Discussion.

Related to my point 5: In rebuttal what is the 'potential caveat in previous experimental investigations' of direction selectivity in SACs? Please explain how the switch to a DSI measure (authors response) resolved the issue that a bipolar cell would have larger response to motion in all directions, which does not obviously aid postsynaptic direction tuning in SACs.

Abstract, Line 12: it appears false to claim that how 'center-surround RFs [contribute] to motion processing is unknown'. I strongly agree with Rev.2 that from the very first studies of mechanisms of DS on - Hassenstein/Reichardt, Barlow and Levick, Borst fly work, the role of interacting antagonistic subunits has been proposed and explored, and lots is known about it. It is unclear to me if what is shown here goes beyond the predictions of the early models. What is novel in the current study may be how a bipolar cell implements this computation, and the mechanistic data given by the pharmacological perturbations and model simulations.

Line 528 – 'kainite' is a mineral; correct to kainate, the neurotransmitter.

Reviewer #1 (Remarks to the Author):

The authors have done an excellent job of incorporating my suggestions. I have no further comments.

Thank you!

Reviewer #2 (Remarks to the Author):

The authors have made an effort to improve the manuscript, have performed some new experiments, and have tried to respond to the previous round of reviews. Unfortunately, major issues remain, and the response to the reviews was not as complete as it should have been. I cannot recommend publication of this manuscript in Nature Communications.

Major issues:

1. Imprecise and incorrect language persists:

There are numerous examples. Here are two of them

“However, computations in cells with center-surround RFs are inherently nonlinear...”

Why? This is just incorrect. I can guess what the authors meant, but this statement as written does not make any sense. A center-surround RF is not “inherently nonlinear”. They cite some recent papers about the fact that bipolar cells may actually integrate space nonlinearly, which means that a linear center-surround RF is an incomplete model. The authors may be talking about space-time separability, which is a different question than spatial nonlinearity. Indeed, BC RFs are probably not space-time separable either. This is the kind of question their experiments could actually answer. But they don’t state the question accurately or address it. Statements like this will really confuse readers.

We agree that this paragraph is too general. In the revised manuscript, we replaced it with a more precise formulation of the known motion-processing abilities of the center-surround receptive field. We changed the wording to indicate that center-surround RFs **can be** nonlinear (in spatial and temporal domains). As our results reveal, nonlinearity is not required for motion processing we describe, but it plays an important role in diversifying bipolar responses to static/moving objects.

“...continuous motion and disappearing stimuli suppress glutamate signaling...”

No. They may activate BCs less than onset motion, but that’s not the same thing as “suppressing” glutamate signaling. In a second revision of a manuscript, reviewers should not be correcting misstatements this important.

The paragraph in question presents a brief summary of our main findings. Responses in the circuit we describe to continuous motion and disappearing stimuli are primarily determined by the **inhibitory surround that suppresses bipolar activation**. Our readout of this suppression is based on glutamate release. While we do not think our language was incorrect or imprecise, we changed 'glutamate signaling' to 'BC activation' as suggested by the reviewer.

2. Authors still show a lack of understanding of previous literature

The authors still show a lack of understanding of both the historical literature about receptive fields and more recent work on motion responses in the retina. They don't even cite probably the most relevant previous paper on motion onset responses, which include a model incorporating bipolar cells (Chen et al., 2013).

Chen et al., 2013 appeared in a previous version of the manuscript, and the relevant paragraph has been reinstated. We expanded our Discussion to compare different motion processing circuits in the retina. Chen et al., 2013 suggested that gain control in bipolar cells can contribute to motion processing in ganglion cells. However, as is clear from that study and a follow-up paper (Chen, Chou, Park, Schwartz, & Berry, 2014), bipolar cells were considered to be relatively simple visual integrators and not motion-sensitive.

We disagree with the reviewer's assertion that we lack an understanding of the literature. We acknowledge that retinal motion processing literature is dense and, at times, confusing, even to experts in the field. To assist in the understanding of how our findings fit with known retinal computations, we have composed a new summary figure, which is reproduced below.

Fig. 7. Circuits for motion computations

Illustration of some of the known motion processing circuits in the vertebrate retina. **a** The computation of direction selectivity in a Hassenstein-Reichardt correlator (Hassenstein, 1956) is mediated by the integration of spatially separated inputs with distinct temporal dynamics (slower arm denoted by τ) (Borst, 2014; Clark & Demb, 2016; Fransen & Borghuis, 2017; J. S. Kim et al., 2014; Matsumoto, Briggman, & Yonehara, 2019). **b** Contrast adaptation (gain-control) supports predictive encoding of moving objects (Berry, Brivanlou, Jordan, & Meister, 1999), increased responsiveness to motion onset (Chen et al., 2013) and detection of motion reversal (Chen et al., 2014). Electrical coupling among BCs and RGCs was shown to compensate for the lag in visual processing (Trenholm, Schwab, Balasubramanian, & Awatramani, 2013), increase sensitivity to correlated input vs. randomly shuffled stimuli (Huang, Rangel, Briggman, & Wei, 2019; Kuo, Schwartz, & Rieke, 2016; Liu, Hong, Rieke, &

Manookin, 2021; Manookin, Patterson, & Linehan, 2018) (c), and assist in object localization (Cooler & Schwartz, 2021) (d). e-h Diverse contributions of inhibitory circuits (depicted in blue) to motion responses. e A spatiotemporal offset between excitation and inhibition is the basis of the Barlow-Levick model of direction selectivity (Barlow & Levick, 1965), found in many DSGCs (Fried, Munch, & Werblin, 2002; Hanson, Sethuramanujam, deRosenroll, Jain, & Awatramani, 2019; I. J. Kim, Zhang, Yamagata, Meister, & Sanes, 2008). f Integration of presynaptic excitatory/inhibitory units with similar spatial receptive fields but reversed polarities (marked by '+' and '-' signs) facilitates detection of second-order (Demb, Zaghloul, & Sterling, 2001) and approaching (Munch et al., 2009) motion. Previous studies identified a role for the center-surround organization in the segregation of local vs. global motion (Baccus, Olveczky, Manu, & Meister, 2008; T. Kim & Kerschensteiner, 2017; T. Kim, Soto, & Kerschensteiner, 2015; Olveczky, Baccus, & Meister, 2003, 2007; Zhang, Kim, Sanes, & Meister, 2012) (g). h We and colleagues (Strauss et al., 2021) reveal differential responsiveness to emerging and exiting objects in cells with a center-surround receptive field architecture, which could facilitate identification of novel items in the visual scene.

As far as the classic Reichardt detector literature, they incorrectly make the assertion that these papers only considered an asymmetric RF and never considered how inhibition and excitation could interact if they were circularly symmetric. This is false. While of course a model specifically made to explain direction selectivity requires some spatial asymmetry, the fact that the same mechanism can explain motion responses depending on the spatiotemporal integration of the RF surround and center is not some kind of revolutionary concept that had been overlooked for 60 years! Indeed, some researchers oversimplified and didn't think about this in their models, but others did, and the core idea dates back at least to Barlow.

We agree that previous literature considered motion processing in spatially symmetric center-surround receptive fields. We cite several studies published in leading journals showing that pronounced surrounds can veto center activation during correlated local and global motion in dedicated retinal circuits. This result is different from the computations we describe in our study. Further, it is unclear if bipolar cells, which do not have as strong and wide surroundings, can segregate local motion.

Our core findings are as follows: 1. The dynamics of responses to motion in bipolar cells cannot be predicted from responses to stationary flashes. As we explain in the text, this finding directly contradicts prior assumptions in the field. 2. Bipolar responses to emerging motion and flashes are significantly larger than responses to exiting stimuli and continuing motion. This effect depends on suppressive (inhibitory) surround formed by horizontal cells and could signify a general principle of tuning to novel visual objects in cells with a center-surround organization. This formulation of bipolar function is not presented in prior literature.

Please see more discussion of the novelty of our findings in our response to Reviewer #3.

3. The new model (Fig. 4) appears to be all about the kinetics of the inhibition rather than its location.

This is incorrect. Panels d and g of figure 4 show how the enhancement of emerging motion is affected by the spatial extent of the horizontal and amacrine surrounds.

The model in Fig. 4 assumes sustained HC feedback and transient (and more delayed) AC feedback with no real justification for those choices. The kinetics of the inhibition (and its spatial extent and strength which are kept fixed) will determine how the motion onset response compares to the smooth motion response. The authors seem to be manipulating the wrong things in the model and make the false HC/AC dichotomy. It's just a linear RF model. The strength and timing of the surround are what matters, not its circuit location.

The model presented in figure 4 is not a linear RF model. In addition to each cell being a rectifying integrator of signals from individual presynaptic cells, horizontal cell drive is modeled as feedback inhibition, whereas amacrine cells perform feedforward inhibition. This information is shown schematically on panels a and e of figure 4, and we provide a complete description of the model in the text.

We did not assume '...sustained HC feedback and transient (and more delayed) AC feedback...'. Panels d and g show bipolar responses over a wide range of model parameters. We varied the temporal kinetics of the horizontal feedback from 60-600 ms, in line with known horizontal cell dynamics. Similarly, we explored the effect of amacrine cell inhibition with amacrine cell kinetics in the 20-200 ms range. The reason amacrine cell drive is transient and delayed in panels e and f is because of nonlinear synaptic integration of bipolar drive by the amacrine cells in our model.

What is the basis of the reviewer's declaration that '...The strength and timing of the surround are what matters, not its circuit location...'? We explicitly show that in a circuit that mimics the visual information flow in the retina, amacrine and horizontal cells' surrounds are qualitatively and quantitatively different.

And don't repeatedly refer to your own model as "detailed". This can only frustrate a reader about all the details that are oversimplified or left out.

In the manuscript, we present two models. One is a linear RF formulation where the activation of the cell is determined by two equations. The second incorporates hundreds of cells with nonlinear RFs and several feedforward and feedback signal propagation arms. **It is more detailed.** That being said, as urged by the reviewer, we replaced the word 'detailed' with 'a circuit-based model'.

4. Responses to the specific suggestions in my previous review were unsatisfactory

A. Clustering.

The authors had a bug in their previous clustering, so they used a different method (which is described in the Methods but only referred to by the programming language in the Response). The authors state in the response that the new method found local minima at 2 and 16-20 clusters, yet

they show the data from 3-11 clusters as that is a better fit to the known number of BCs. And they don't mention in the paper that this was not the real minimum. This is intentionally misleading!

We did not try to mislead. Our clustering follows the standards in the field, where the biologically relevant range of clusters is considered. We have included the minima information in the new version of the manuscript. Additionally, we would like to stress that our dataset used for clustering is available in full online.

B. HC manipulation

While it's nice that the authors did a manipulation directed at HC, their manipulation has a fatal flaw in that it also eliminates ACs! CNQX blocks AMPA receptors on ACs! Myself and another reviewer suggested HC-specific manipulations (pharmacological or genetic), and while the pharmacological experiments may have represented an incomplete block or had some off-target effects, they are not as problematic as the choice to include CNQX and block all the ACs!

We disagree with this criticism.

1. In our initial submission, we have shown that specific and effective blockage of amacrine cell inhibition did not have a pronounced effect on motion responses in putative bipolar cells.
2. Based on point (1), the remaining possible mechanisms of motion sensitivity in bipolar cells are horizontal cell-mediated surround, or signal interactions in bipolar cell center (perhaps akin to (Kuo et al., 2016; Liu et al., 2021; Manookin et al., 2018)).
In the revised manuscript, we performed pharmacological blockage of horizontal cell inhibition. The blockers were highly effective in influencing the signals in bipolar cells, blocking the enhanced representation of emerging motion in 100% of our experiments. This outcome is a clear indicator of the role of horizontal inhibition in the computations we studied here.
3. We agree that if we performed the blockage of horizontal cells only, a valid objection to our interpretation could be an indirect effect on amacrine cells. However, because we conclusively ruled out amacrine cells as significant mediators to the computation in (1) we are not concerned that our pharmacology blocks amacrine cells.
4. Perhaps the reviewer did not consider this point when they made the suggestion of affecting horizontal cell feedback, but any effective manipulation of horizontal cell inhibition by definition affects bipolar cells and thus their input to amacrine cells. Previous work had shown that horizontal cell blockage has numerous, unpredictable effects on light responses in downstream cells (Drinneberg et al., 2018). Therefore, perturbation of amacrine cell function following manipulation of horizontal cells is unavoidable. For this reason, our strategy, which is based on published pharmacological manipulation of horizontal cells (Barnes, Grove, McHugh, Hirano, & Brecha, 2020), is a clean method to avoid any unknown effects of amacrine cells on bipolar responses.

I suggested using the individual GABA receptor blockers one at a time and the authors simply made excused for not performing that experiment.

In our previous response, we provided a comprehensive explanation of why these experiments are not informative. In brief, the pharmacological manipulations proposed by the reviewer are not specific to known circuits. There are multiple possible mechanisms by which blockage of some amacrine cells could affect bipolar cells. Some of them are direct; others may be indirect, for example, involving inhibition of other amacrine cells. The number of possible circuits that may be affected is astronomical. **We already know that amacrine cells are not required for the motion computations we describe.** It would be highly likely that any potential effect of individual blockers could be explained as indirect due to elevated hyperexcitability due to an imbalance in amacrine cell inhibition.

As discussed in our previous response, the number of experiments required to test individual blockers with a decent sample size across bipolar cell types greatly exceeds our current dataset. Furthermore, we expect the possible results to be largely uninterpretable due to unknown cells involved and off-target effects. As we had mentioned before, we used individual GABA/glycine blockers in our preliminary experiments but did not observe a notable effect on motion processing. We humbly ask the reviewer to provide a compelling reason to perform a time-consuming investigation lacking clear hypotheses or objectives of circuits that we had already ruled out as the principal mediators of the studied phenomenon.

C. Other forms of motion stimuli are not tested, including the one specifically requested. The arguments for not testing this stimulus are thin and don't make a lot of sense. It's not a difficult experiment, and it helps connect this work to previous work on the role of bipolar cells in motion detection. The authors claim in the introduction that bipolar cells have rarely been considered to be important in motion detection. Yet they refuse to do the experiments to connect to the few papers that have made this connection!

We have performed the experiment as in (Kuo et al., 2016; Manookin et al., 2018) as asked by the reviewer. Previous studies indicated that bipolar responses to smooth motion are stronger than to a randomized sequence of bar presentations. Our findings are different (Fig. S9) but in line with our other experimental results.

We presented multiple trials consisting of movies of pseudorandomly shuffled bars. We found that responses to shuffled bar presentation critically depend on the activation of the surround. In movies where the **surround was stimulated first**, the amplitude of the responses was similar to smooth motion. When the stimulus engaged the **center first**, the responses were much more pronounced and comparable in amplitude to responses to flashed and emerging motion.

The results of this experiment are in accordance with our model of center-surround interactions in the bipolar receptive field. We used comparable parameters to (Kuo et al., 2016), but our results are dissimilar. The most plausible explanation for the apparent discrepancy is different surround

strengths due to the distinct light levels (photopic in our case, scotopic-low mesopic in Kuo et al. 2016). The stronger surrounds, known to be present in the photopic light regime, probably dominate over gap junction mediated amplification of bipolar responses during correlated motion, as shown in previous work. We discuss this result further in the Discussion.

Reference

Chen, E.Y., Marre, O., Fisher, C., Schwartz, G., Levy, J., Silveira, R.A. da, and Berry, M.J. (2013). Alert Response to Motion Onset in the Retina. *J. Neurosci.* 33, 120–132.

Reviewer #3 (Remarks to the Author):

Review of revised manuscript NCOMMS-21-16455A.

Related to my point 4: The presentation of the BCSAC glutamate imaging experiments (Lines 73-95) is a great improvement.

Line 322 States that ‘The earliest direction-selective signals are present in dendrites of starburst amacrine cells’, and the authors’ results show that motion processing in BCs is not direction-selective. It appears relevant to relate these results to recent work - the most detailed study of BC glutamate output during motion stimulation, Matsumoto, ..., Yonehara (*Neuron*, Sep2021; not cited) which showed direction tuning of individual boutons on axon terminals on some bipolar cell types, including those referenced in the current study. This direction tuning may still originate in SACs – so the opening sentence may be factually correct, given the contentious nature of every claim DS, a concise reference of earlier/conflicting findings is warranted, at least in Discussion.

We completely agree. We have corrected that statement. We have also included a paragraph discussing the findings of (Matsumoto et al., 2021) and how they compare to our work.

Related to my point 5: In rebuttal what is the ‘potential caveat in previous experimental investigations’ of direction selectivity in SACs? Please explain how the switch to a DSI measure (authors response) resolved the issue that a bipolar cell would have larger response to motion in all directions, which does not obviously aid postsynaptic direction tuning in SACs.

We agree with the reviewer that enhanced responses to emerging motion in bipolar cells do not aid directional tuning in SACs. However, it can influence the interpretation of experimental results. We and others used visual stimuli that were focused on a small part of SAC dendritic fields (Ding, Smith, Poleg-Polsky, Diamond, & Briggman, 2016; Lee & Zhou, 2006; Oesch & Taylor, 2010; Vlasits et al.,

2016)). The resulting visual stimulation produced stimuli that we refer to here as emerging motion and exiting motion. Our present work clearly indicates an asymmetry in the excitatory drive reaching the SAC during such stimulation paradigm, the presence of which was unfortunately not appreciated in previous studies. By computing the DSI of the asymmetric BC drive, we aim to use a 'common currency' to quantify the extent of the potential bias in previous SAC DS estimations. Specifically, in our hands, the difference in the peak glutamate release had a DSI of 30%. We would anticipate that recordings from SACs that found similar DSI levels would have a disproportionately large 'contamination' of SAC DS abilities by the underlying differences in the bipolar drive.

Abstract, Line 12: it appears false to claim that how 'center-surround RFs [contribute] to motion processing is unknown'. I strongly agree with Rev.2 that from the very first studies of mechanisms of DS on - Hassenstein/Reichardt, Barlow and Levick, Borst fly work, the role of interacting antagonistic subunits has been proposed and explored, and lots is known about it. It is unclear to me if what is shown here goes beyond the predictions of the early models. What is novel in the current study may be how a bipolar cell implements this computation, and the mechanistic data given by the pharmacological perturbations and model simulations.

We apologize for the unclear wording in the abstract. We have replaced the sentence with the following:

Circularly symmetric center-surround RFs are thought to enhance responses to spatial contrasts (i.e., edges), but how visual edges affect motion processing is unclear.

As we state in response to reviewer #2, we have added a new figure and expanded the discussion section to compare and contrast our findings with previous literature.

We interpret the comments by Reviewer #2 to mean that they are concerned that our findings lack novelty. This concern is based on the well-established understanding of center-surround RF organization and several previous studies which have studied motion processing in BCs. In response, we argue that the findings of our research are novel and interesting because they directly contradict long-standing assumptions about motion response in BCs and explore a novel visual function of the retina.

In Figure 1 and accompanying paragraphs, we studied the simple case of comparison between responses to flashes and moving bars. **Our findings of diverse and unpredictable kinetics and amplitude of the responses to motion (vs. flashes) are in a direct contradiction to the assumptions in the field**, as is evident from studies that applied the concepts developed by Hassenstein/Reichardt to motion processing in bipolar cells (Fransen & Borghuis, 2017; Greene, Kim, Seung, & EyeWirers, 2016; J. S. Kim et al., 2014; Stincic, Smith, & Taylor, 2016).

Additionally, in the rest of the 5 data figures and the body of the manuscript, we present evidence for a visual computation that enhances the detection of novel objects in the visual scene. To the best of our knowledge, **this visual function has never been observed in the retina and the simple center-surround antagonism has never been considered as a potential mechanism for this ethologically critical computation**. Finally, to the best of our knowledge, this is the first study to

investigate and show evidence that motion processing begins in the first retinal synapse. We believe that these features of our research adequately demonstrate its novelty.

Another concern raised by Reviewer number 2 is that models developed for direction selectivity are conceptually similar to our findings. In response, we argue that the computation for the detection of novel objects is distinct from the computation of motion direction in the retina. The detection of an emerging visual object does not rely on spatially asymmetric RFs which is the basis of directional tuning in DSGCs (and potentially SACs), and the essence of Hassenstein/Reichardt and Barlow/Levick models of direction selectivity. We think that a potential source of misunderstanding is due to our colleagues' interpretation of the results in an alternative model that draws inspiration from the Barlow/Levick circuit for direction selectivity (Strauss et al., 2021). While our results are essentially the same, our explanation of our findings is grounded in the spatiotemporal properties of the center-surround circuit itself. Similar to the Barlow/Levick model, we stress the importance of the history of surround inhibition in the responses of the cell. We propose that the basis of novel object detection computation relies on the enhancement of stimuli that do not engage the surround before the center (flashes and emerging motion), similar to how preferred-direction stimuli are enhanced in the Barlow/Levick model. In contrast to Barlow/Levick, our model does not require spatially asymmetric RFs, and for this reason, can not differentiate between directions of stimulation. We believe our model is both simpler and applicable to a wider range of possible stimuli and therefore has greater explanatory power.

We would like to stress that while antagonistic RF subunits were studied extensively already in the 50s and the 60s, early investigators did not explore the phenomena fully. As evidence, several studies published in the last 20 years showed that strong inhibitory surrounds could facilitate preferential responses to local motion (Baccus et al., 2008; T. Kim & Kerschensteiner, 2017; T. Kim et al., 2015; Olveczky et al., 2003, 2007; Zhang et al., 2012). One could argue that these results were expected and potentially fully explained by the classic concepts of the center-surround RFs. Instead, the studies were published in top-tier journals, clearly indicating that the field found the computation and circuit mechanisms to be novel and interesting. We believe the same argument applies to our work. While in principle, enhanced representation of emerging motion could be deduced from the fundamental principles of signal integration in center-surround RFs, in practice, this functional ability was not appreciated in the large body of work describing the contribution of these RFs to visual processing. What is more, unlike the studies cited above that primarily focused on motion processing in cells in which the intensity of the surround is dialed to an extreme, our manuscript describes computations that are relevant to the majority of the cells in the retina and other visual structures where such extremity does not exist.

To summarize, **our findings are different from previous motion processing models in the literature in terms of the visual function, readout cells, and circuit implementation.** We present empirical results that challenge the prior understanding of motion processing in bipolar cells, and propose a conceptually new role for center-surround RFs in visual computations.

Line 528 – 'kainite' is a mineral; correct to kainate, the neurotransmitter.

Thank you, corrected

References:

- Barnes, S., Grove, J. C. R., McHugh, C. F., Hirano, A. A., & Brecha, N. C. (2020). Horizontal Cell Feedback to Cone Photoreceptors in Mammalian Retina: Novel Insights From the GABA-pH Hybrid Model. *Front Cell Neurosci*, *14*, 595064. doi:10.3389/fncel.2020.595064
- Berry, M. J., 2nd, Brivanlou, I. H., Jordan, T. A., & Meister, M. (1999). Anticipation of moving stimuli by the retina. *Nature*, *398*(6725), 334-338. doi:10.1038/18678
- Borst, A. (2014). Neural circuits for elementary motion detection. *J Neurogenet*, *28*(3-4), 361-373. doi:10.3109/01677063.2013.876022
- Chen, E. Y., Chou, J., Park, J., Schwartz, G., & Berry, M. J., 2nd. (2014). The neural circuit mechanisms underlying the retinal response to motion reversal. *J Neurosci*, *34*(47), 15557-15575. doi:10.1523/JNEUROSCI.1460-13.2014
- Chen, E. Y., Marre, O., Fisher, C., Schwartz, G., Levy, J., da Silveira, R. A., & Berry, M. J., 2nd. (2013). Alert response to motion onset in the retina. *J Neurosci*, *33*(1), 120-132. doi:10.1523/JNEUROSCI.3749-12.2013
- Clark, D. A., & Demb, J. B. (2016). Parallel Computations in Insect and Mammalian Visual Motion Processing. *Curr Biol*, *26*(20), R1062-R1072. doi:10.1016/j.cub.2016.08.003
- Cooler, S., & Schwartz, G. W. (2021). An offset ON-OFF receptive field is created by gap junctions between distinct types of retinal ganglion cells. *Nat Neurosci*, *24*(1), 105-115. doi:10.1038/s41593-020-00747-8
- Demb, J. B., Zaghloul, K., & Sterling, P. (2001). Cellular basis for the response to second-order motion cues in Y retinal ganglion cells. *Neuron*, *32*(4), 711-721.
- Ding, H., Smith, R. G., Polog-Polsky, A., Diamond, J. S., & Briggman, K. L. (2016). Species-specific wiring for direction selectivity in the mammalian retina. *Nature*, *535*(7610), 105-110. doi:10.1038/nature18609
- Drinnenberg, A., Franke, F., Morikawa, R. K., Jüttner, J., Hillier, D., Hantz, P., . . . Roska, B. (2018). How Diverse Retinal Functions Arise from Feedback at the First Visual Synapse. *Neuron*, *99*(1), 117-134 e111. doi:10.1016/j.neuron.2018.06.001
- Fransen, J. W., & Borghuis, B. G. (2017). Temporally Diverse Excitation Generates Direction-Selective Responses in ON- and OFF-Type Retinal Starburst Amacrine Cells. *Cell Rep*, *18*(6), 1356-1365. doi:10.1016/j.celrep.2017.01.026
- Fried, S. I., Munch, T. A., & Werblin, F. S. (2002). Mechanisms and circuitry underlying directional selectivity in the retina. *Nature*, *420*(6914), 411-414.
- Greene, M. J., Kim, J. S., Seung, H. S., & EyeWirers. (2016). Analogous Convergence of Sustained and Transient Inputs in Parallel On and Off Pathways for Retinal Motion Computation. *Cell Rep*, *14*(8), 1892-1900. doi:10.1016/j.celrep.2016.02.001
- Hanson, L., Sethuramanujam, S., deRosenroll, G., Jain, V., & Awatramani, G. B. (2019). Retinal direction selectivity in the absence of asymmetric starburst amacrine cell responses. *Elife*, *8*. doi:10.7554/eLife.42392
- Hassenstein, B., Reichardt, W. (1956). Systemtheoretische Analyse Der Zeit, Reihenfolgen Und Vorzeichenauswertung Bei Der Bewegungspereption Des Rüsselkäfers Chlorophanus. *Z. Naturforsch. B*, *11*, 513-524.

- Huang, X., Rangel, M., Briggman, K. L., & Wei, W. (2019). Neural mechanisms of contextual modulation in the retinal direction selective circuit. *Nat Commun*, *10*(1), 2431. doi:10.1038/s41467-019-10268-z
- Kim, I. J., Zhang, Y., Yamagata, M., Meister, M., & Sanes, J. R. (2008). Molecular identification of a retinal cell type that responds to upward motion. *Nature*, *452*(7186), 478-482. doi:10.1038/nature06739
- Kim, J. S., Greene, M. J., Zlateski, A., Lee, K., Richardson, M., Turaga, S. C., . . . EyeWriters. (2014). Space-time wiring specificity supports direction selectivity in the retina. *Nature*, *509*(7500), 331-336. doi:10.1038/nature13240
- Kim, T., & Kerschensteiner, D. (2017). Inhibitory Control of Feature Selectivity in an Object Motion Sensitive Circuit of the Retina. *Cell Rep*, *19*(7), 1343-1350. doi:10.1016/j.celrep.2017.04.060
- Kim, T., Soto, F., & Kerschensteiner, D. (2015). An excitatory amacrine cell detects object motion and provides feature-selective input to ganglion cells in the mouse retina. *Elife*, *4*. doi:10.7554/eLife.08025
- Kuo, S. P., Schwartz, G. W., & Rieke, F. (2016). Nonlinear Spatiotemporal Integration by Electrical and Chemical Synapses in the Retina. *Neuron*, *90*(2), 320-332. doi:10.1016/j.neuron.2016.03.012
- Lee, S., & Zhou, Z. J. (2006). The synaptic mechanism of direction selectivity in distal processes of starburst amacrine cells. *Neuron*, *51*(6), 787-799.
- Liu, B., Hong, A., Rieke, F., & Manookin, M. B. (2021). Predictive encoding of motion begins in the primate retina. *Nat Neurosci*. doi:10.1038/s41593-021-00899-1
- Manookin, M. B., Patterson, S. S., & Linehan, C. M. (2018). Neural Mechanisms Mediating Motion Sensitivity in Parasol Ganglion Cells of the Primate Retina. *Neuron*. doi:10.1016/j.neuron.2018.02.006
- Matsumoto, A., Agbariah, W., Nolte, S. S., Andrawos, R., Levi, H., Sabbah, S., & Yonehara, K. (2021). Direction selectivity in retinal bipolar cell axon terminals. *Neuron*. doi:10.1016/j.neuron.2021.07.008
- Matsumoto, A., Briggman, K. L., & Yonehara, K. (2019). Spatiotemporally Asymmetric Excitation Supports Mammalian Retinal Motion Sensitivity. *Curr Biol*. doi:10.1016/j.cub.2019.08.048
- Munch, T. A., da Silveira, R. A., Siebert, S., Viney, T. J., Awatramani, G. B., & Roska, B. (2009). Approach sensitivity in the retina processed by a multifunctional neural circuit. *Nat Neurosci*, *12*(10), 1308-1316. doi:10.1038/nn.2389
- Oesch, N. W., & Taylor, W. R. (2010). Tetrodotoxin-resistant sodium channels contribute to directional responses in starburst amacrine cells. *PLoS One*, *5*(8), e12447. doi:10.1371/journal.pone.0012447
- Olveczky, B. P., Baccus, S. A., & Meister, M. (2003). Segregation of object and background motion in the retina. *Nature*, *423*(6938), 401-408. doi:10.1038/nature01652
- Olveczky, B. P., Baccus, S. A., & Meister, M. (2007). Retinal adaptation to object motion. *Neuron*, *56*(4), 689-700.
- Stincic, T., Smith, R. G., & Taylor, W. R. (2016). Time course of EPSCs in ON-type starburst amacrine cells is independent of dendritic location. *J Physiol*, *594*(19), 5685-5694. doi:10.1113/JP272384
- Strauss, S., Korympidou, M. M., Ran, Y., Franke, K., Schubert, T., Baden, T., . . . Vlasits, A. L. (2021). Center-surround interactions underlie bipolar cell motion sensing in the mouse retina. *bioRxiv*, 2021.2005.2031.446404. doi:10.1101/2021.05.31.446404
- Trenholm, S., Schwab, D. J., Balasubramanian, V., & Awatramani, G. B. (2013). Lag normalization in an electrically coupled neural network. *Nat Neurosci*, *16*(2), 154-156. doi:10.1038/nn.3308
- Vlasits, A. L., Morrie, R. D., Tran-Van-Minh, A., Bleckert, A., Gainer, C. F., DiGregorio, D. A., & Feller, M. B. (2016). A Role for Synaptic Input Distribution in a Dendritic Computation of Motion Direction in the Retina. *Neuron*, *89*(6), 1317-1330. doi:10.1016/j.neuron.2016.02.020

Zhang, Y., Kim, I. J., Sanes, J. R., & Meister, M. (2012). The most numerous ganglion cell type of the mouse retina is a selective feature detector. *Proc Natl Acad Sci U S A*, *109*(36), E2391-2398. doi:10.1073/pnas.1211547109

REVIEWERS' COMMENTS

Reviewer #2 (Remarks to the Author):

The authors have made some improvements to the text and performed some additional experiments that strengthen the paper in several ways. However, two really major issues remain unresolved.

1. The ROI clustering is unconvincing, so it is not clear if the clusters correspond to BC types. Thus, the data does not tell us anything about whether the surrounds of different BCs are different. The Euler lab has published several papers clustering glutamate signals into particular anatomically-identified BC types, and their related work on BioRxiv (Strauss et al, 2022) shows differences in surround strength among the BC types that affect their motion processing.

2. While I appreciate that the authors did additional experiments to explore the circuit mechanism of BC surrounds, their results do not match their conclusion that the "critical step of motion processing and novel object enhancement relies on HC feedback and is therefore performed in the first retinal synapse." Figure S6 is critical here. The authors applied HEPES to block the pH-dependent mechanism of horizontal cell (HC) feedback in the outer retina (an experiment I suggested in the previous round), but the result was that this manipulation did NOT affect sensitivity to emerging motion. Their other experiment blocking ionotropic GABA and glycine receptors (Fig. 3) also did not eliminate this effect. Finally, they block all ionotropic glutamate receptors in the retina (including the input to both HCs and amacrine cells) via CNQX + HEPES (though the role of HEPES in these experiments is unclear since it did not have any effect alone) and this manipulation does eliminate the emerging motion response (Fig. 3).

The authors conclude from these pharmacology experiments that HCs are responsible for BC surrounds. But by what synaptic mechanism? HEPES blocks the pH-dependent feedback, and Gabazine blocks the GABA receptors on BC dendrites onto which HCs might release GABA. Are the authors suggesting some ephaptic mechanism is responsible for BC surrounds? This would be a pretty bold claim to be making with these indirect measurements and no positive evidence. Isn't it just as likely that the BC surrounds come from some amacrine cell mechanism that does not depend on ionotropic GABA or glycine receptors? Metabotropic receptors? Gap junctions? Neuromodulators? I agree that the pharmacology experiments suggest that something non-canonical and interesting might be responsible for BC surrounds, but instead of doing more experiments to track it down, they just conclude it is HCs without even acknowledging that their pharmacology results do not match the known mechanisms of that synapse.

The revised manuscript has some improvements in its treatment of the past literature on motion processing and its precision in wording about nonlinearity versus space-time separability, but this issue of the pharmacology and the mechanism of BC surrounds is central to the paper. "First Retinal Synapse" is in the title, and the authors have not proven that is where BC surrounds originate.

Reviewer #3 (Remarks to the Author):

I appreciate the authors' responses to my comments. I am satisfied with the response to my first point, origin of DS statement, and my second point, use of DSI. I think the authors have done well in the response to my third point, reports on influence of center-surround on motion processing goes back >6 decades. This is why lines 46-47 triggered in me a 'wait a minute...' response. This point – novelty – appears to be at the heart of Reviewer 2's comments, also. The authors nail this issue in one of their comments: the finding they report is at the same time 'obvious' when considered from what is presumed known in the field (center-surround organization

and its impact on visual encoding), but at the same time it has not been demonstrated and probed in a focused and comprehensive manner. This is what the authors have done, and I think this is a contribution for the field. I think the main claims are sufficiently well-supported by the data, and the experiments and modeling – which are both state-of-the-art – performed to a high standard. My remaining comments pertain to presentation, as follows.

Lines 425-426: 'Several studies suggested that responses to motion are suppressed by gain control mechanisms in the retina (Fig. 7) 7,71'. It was my impression that the Meister paper (ref. 71) shows that gain control DRIVES responses to motion. The follow-up, Leonardo and Meister (2013) certainly does.

Line 512: This URL is outdated, consider using the shorter link www.borghuisinstruments.com

Line 756: Figure legend states that 'Diversity of responses [...] suggests 14 functional clusters'. This is an overstatement, as several other numbers could have been used – low and high based on reported minima (see also Rev2, point A). The statistically most valid number would be determined using a Bayesian Information Criterion calculation, but this was not done here. It is fine to go with expected outcome based on final anatomical reconstructions. Best not to suggest that the cluster analysis of these recorded responses leads to the number of 14.

REVIEWERS' COMMENTS

Reviewer #2 (Remarks to the Author):

1. The ROI clustering is unconvincing, so it is not clear if the clusters correspond to BC types. Thus, the data does not tell us anything about whether the surrounds of different BCs are different. The Euler lab has published several papers clustering glutamate signals into particular anatomically-identified BC types, and their related work on BioRxiv (Strauss et al, 2022) shows differences in surround strength among the BC types that affect their motion processing.

Please see the response to reviewer#3 regarding clustering. We must stress that emerging object enhancement was evident in multiple layers of the IPL and for both On- and Off-signals (Fig. 2). Thus, this is a general property of multiple BC types, even with imprecise functional cluster-BC type correlation.

We have not tried to correlate the strength of the surround with motion processing in the current manuscript. We are actively investigating the question of how spatiotemporal properties of the surround influence motion processing. Our preliminary data suggest a complex relationship, which we hope to delineate in ganglion cells – where the surround could be much more precisely measured.

2. While I appreciate that the authors did additional experiments to explore the circuit mechanism of BC surrounds, their results do not match their conclusion that the “critical step of motion processing and novel object enhancement relies on HC feedback and is therefore performed in the first retinal synapse.” Figure S6 is critical here. The authors applied HEPES to block the pH-dependent mechanism of horizontal cell (HC) feedback in the outer retina (an experiment I suggested in the previous round), but the result was that this manipulation did NOT affect sensitivity to emerging motion. Their other experiment blocking ionotropic GABA and glycine receptors (Fig. 3) also did not eliminate this affect. Finally, they block all ionotropic glutamate receptors in the retina (including the input to both HCs and amacrine cells) via CNQX + HEPES (though the role of HEPES in these experiments is unclear since it did not have any affect alone) and this manipulation does eliminate the emerging motion response (Fig. 3).

The authors conclude from these pharmacology experiments that HCs are responsible for BC surrounds. But by what synaptic mechanism? HEPES blocks the pH-dependent feedback, and Gabazine blocks the GABA receptors on BC dendrites onto which HCs might release GABA. Are the authors suggesting some ephaptic mechanism is responsible for BC surrounds?

This would be a pretty bold claim to be making with these indirect measurements and no positive evidence. Isn't it just as likely that the BC surrounds come from some amacrine cell mechanism that does not depend on ionotropic GABA or glycine receptors? Metabotropic receptors? Gap junctions? Neuromodulators? I agree that the pharmacology experiments suggest that something non-canonical and interesting might be responsible for BC surrounds, but instead of doing more experiments to track it down, they just conclude it is HCs without even acknowledging that their pharmacology results do not match the known mechanisms of that synapse.

The revised manuscript has some improvements in its treatment of the past literature on motion processing and its precision in wording about nonlinearity versus space-time separability, but this issue of the pharmacology and the mechanism of BC surrounds is central to the paper. "First Retinal Synapse" is in the title, and the authors have not proven that is where BC surrounds originate.

In general, we agree with the reviewer that pharmacological interventions should be interpreted carefully due to off-target effects. However, our experimental results strongly support our hypothesis that the critical step in emerging motion processing occurs in the photoreceptor-HC synapse. First, ionotropic glutamate receptor blockers are commonly used in the field and are extremely potent in eliminating HC feedback (Hirasawa and Kaneko 2003, Vessey, Stratis et al. 2005, Szikra, Trenholm et al. 2014, Chapot, Behrens et al. 2017, Grove, Hirano et al. 2019, Hirano, Vuong et al. 2020).

Why did not pH buffering with HEPES or GABAergic blockers affect the motion processing properties of BCs? We note that CNQX blocks the input to HC entirely, whereas pH buffering or GABA blockers affect only one of several mechanisms mediating feedback inhibition from HCs. Thus, it is possible that the remaining HC feedback is sufficient to affect motion responses (for example of the difference in efficacy see (Szikra, Trenholm et al. 2014)).

HC action on photoreceptors is strongly influenced by light levels (Szikra, Trenholm et al. 2014). Most previous studies measured ephaptic feedback in the dark, while the current study was performed in photopic light levels. The precise mechanisms mediating HC feedback remain under debate. Most of the evidence for ephaptic HC feedback to photoreceptors is from studies in non-mammalian vertebrates, where it occurs through Connexin hemichannels, however, these channels are not found in mouse retina (Barnes, Grove et al. 2020). Pannexin channels are found diffusely and away from the invaginating HC tips in the rodent retina. Further, it is possible that distinct processes underlie HC-photoreceptor cleft alkalinization and acidification (Wang, Holzhausen et al. 2014).

Another tantalizing possibility is that the HCs in our preparation were relatively hyperpolarized.

Ephaptic feedback to photoreceptors requires HC depolarization; the effect of HEPES and GABA blockers on HC feedback is significantly impaired in hyperpolarized HCs. We have modified the manuscript to include a discussion of the mechanisms that can be responsible for the lack of effect of HEPES in our hands.

We base our conclusions on three key findings. 1. Amacrine cell inhibition is not required for the motion processing we describe. 2. Computational model supports the feedback of HCs as a mediator of motion sensitivity. 3. Direct blockage of HCs eliminates edge dependence of motion responses. While the last finding is a direct validation of our hypothesis, the first experiment and the modeling were already reliable (but indirect) indicators that HCs are the most plausible mediators of motion effects we describe, as we noted in the first version of the manuscript. The reason is that BC surround is established by amacrine and horizontal cells. It is possible that some BC types receive an atypical input that affects their surround, such a scheme is highly unlikely to explain our results, which we observed for most BC types with both On and Off response polarities.

HCs form an electrically coupled network and can also provide feedback to photoreceptors on an individual cell-by-cell basis. As we have mentioned, HC have multiple distinct feedback mechanisms. As a result, the required experiments to determine the specific feedback mechanism will be extensive and that study is outside of the scope of this manuscript.

Given the strong evidence from the CNQX experiments, combined with the computational modeling, we feel confident that novel motion enhancement seen in BPC glutamate release arises from HC-mediated feedback. However, given the imprecise nature of the pharmacological interventions, unclear specific HC feedback mechanism, and since we do not want to divert readers' attention from other main findings, we have decided to change the title. The new title now reads 'Classical Center-Surround Receptive Fields Facilitate Novel Object Detection in Retinal Bipolar Cells'.

We appreciate the previous feedback regarding the treatment of the literature and are glad that this version is more acceptable in this regard.

Last, concerning the minor comment of the reviewer dealing with the reasons we applied CNQX and HEPES. For the first revision of our manuscript, we performed the CNQX+HEPES experiment first, which has been used by other groups to block HC activity as we describe above. After seeing that this manipulation did in fact, inhibit emerging motion enhancement, we performed the experiment with HEPES alone, as suggested by the reviewers.

Reviewer #3 (Remarks to the Author):

I appreciate the authors' responses to my comments. I am satisfied with the response to my first point, origin of DS statement, and my second point, use of DSI. I think the authors have done well in the response to my third point, reports on influence of center-surround on motion processing goes back >6 decades. This is why lines 46-47 triggered in me a 'wait a minute...' response. This point – novelty – appears to be at the heart of Reviewer 2's comments, also.

The authors nail this issue in one of their comments: the finding they report is at the same time 'obvious' when considered from what is presumed known in the field (center-surround organization and its impact on visual encoding), but at the same time it has not been demonstrated and probed in a focused and comprehensive manner. This is what the authors have done, and I think this is a contribution for the field. I think the main claims are sufficiently well-supported by the data, and the experiments and modeling – which are both state-of-the-art – performed to a high standard. My remaining comments pertain to presentation, as follows.

Lines 425-426: 'Several studies suggested that responses to motion are suppressed by gain control mechanisms in the retina (Fig. 7) 7,71'. It was my impression that the Meister paper (ref. 71) shows that gain control DRIVES responses to motion. The follow-up, Leonardo and Meister (2013) certainly does.

Corrected, thank you

Line 512: This URL is outdated, consider using the shorter link www.borghuisinstruments.com

Fixed

Line 756: Figure legend states that 'Diversity of responses [...] suggests 14 functional clusters'. This is an overstatement, as several other numbers could have been used – low and high based on reported minima (see also Rev2, point A). The statistically most valid number would be determined using a Bayesian Information Criterion calculation, but this was not done here. It is fine to go with expected

outcome based on final anatomical reconstructions. Best not to suggest that the cluster analysis of these recorded responses leads to the number of 14.

The reviewer's point is well taken that finding a 'statistically most valid number' of clusters may be, and often is, done using BIC or AIC. However, for the present cluster analysis, we chose to take advantage of the R's NbClust library when addressing reviewer 2's concern regarding the fidelity of our original clustering implementation in Igor Pro. While NbClust is very widely used (cited at least 2117 times), it, unfortunately, does not implement these indices. Thus, we deferred to the C-index which we had originally devised using at the outset of this project. Moreover, after exploring a non-exhaustive set of common clustering methods, we opted to use Ward's method because it did the best job minimizing intra-cluster variance; this of course necessitates an increase in bias. Overall, we believe that given the experience and findings of other groups attempting similar clustering based on iGluSnFR responses (we provide multiple citations in the manuscript), the biological variability inherent to these experiments precludes 1:1 mapping of response clusters to BC types. We changed the legend accordingly.

References:

- Barnes, S., J. C. R. Grove, C. F. McHugh, A. A. Hirano and N. C. Brecha (2020). "Horizontal Cell Feedback to Cone Photoreceptors in Mammalian Retina: Novel Insights From the GABA-pH Hybrid Model." Front Cell Neurosci **14**: 595064.
- Chapot, C. A., C. Behrens, L. E. Rogerson, T. Baden, S. Pop, P. Berens, T. Euler and T. Schubert (2017). "Local Signals in Mouse Horizontal Cell Dendrites." Curr Biol **27**(23): 3603-3615 e3605.
- Grove, J. C. R., A. A. Hirano, J. de Los Santos, C. F. McHugh, S. Purohit, G. D. Field, N. C. Brecha and S. Barnes (2019). "Novel hybrid action of GABA mediates inhibitory feedback in the mammalian retina." PLoS Biol **17**(4): e3000200.
- Hirano, A. A., H. E. Vuong, H. L. Kornmann, C. Schietroma, S. L. Stella, Jr., S. Barnes and N. C. Brecha (2020). "Vesicular Release of GABA by Mammalian Horizontal Cells Mediates Inhibitory Output to Photoreceptors." Front Cell Neurosci **14**: 600777.
- Hirasawa, H. and A. Kaneko (2003). "pH changes in the invaginating synaptic cleft mediate feedback from horizontal cells to cone photoreceptors by modulating Ca²⁺ channels." J Gen Physiol **122**(6): 657-671.
- Szikra, T., S. Trenholm, A. Drinnenberg, J. Juttner, Z. Raics, K. Farrow, M. Biel, G. Awatramani, D. A. Clark, J. A. Sahel, R. A. da Silveira and B. Roska (2014). "Rods in daylight act as relay cells for cone-driven horizontal cell-mediated surround inhibition." Nat Neurosci **17**(12): 1728-1735.
- Vessey, J. P., A. K. Stratis, B. A. Daniels, N. Da Silva, M. G. Jonz, M. R. Lalonde, W. H. Baldrige and S. Barnes (2005). "Proton-mediated feedback inhibition of presynaptic calcium channels at the cone photoreceptor synapse." J Neurosci **25**(16): 4108-4117.
- Wang, T. M., L. C. Holzhausen and R. H. Kramer (2014). "Imaging an optogenetic pH sensor reveals that protons mediate lateral inhibition in the retina." Nat Neurosci **17**(2): 262-268.